
**Elevated levels of OH observed in haze events during wintertime in central Beijing**

Eloise J. Slater[1], Lisa K. Whalley[1,2], Robert Woodward-Massey[1,a], Chunxiang Ye[1,a], James D Lee[3,4], Freya Squires[4], James R. Hopkins[3,4], Rachel E Dunmore[4], Marvin Shaw[3,4], Jacqueline F. Hamilton[4], Alastair C Lewis[3,4], Leigh R. Crilley[5,b], Louisa Kramer[5], William Bloss[5], Tuan Vu[5], Yele Sun[6], Weiqi Xu[6], Siyao Yue[6], Lujie Ren[6], W. Joe F. Acton[7], C. Nicholas Hewitt[7], Xinming Wang[8], Pingqing Fu[9] and Dwayne E. Heard[1]

[1]School of Chemistry, University of Leeds, Leeds, LS2 9JT, UK

[2]National Centre for Atmospheric Science, University of Leeds, Leeds, LS2 9JT, UK

[3]National Centre for Atmospheric Science, University of York, Heslington, York, YO10 5DD, UK

[4]Wolfson Atmospheric Chemistry Laboratories, Department of Chemistry, University of York, Heslington, York, YO10 5DD, UK

[5]School of Geography, Earth and Environmental Sciences, University of Birmingham, B15 2TT, Birmingham, UK

[6]State Key Laboratory of Atmospheric Boundary Layer Physics and Atmospheric Chemistry, Institute for Atmospheric Physics, Chinese Academy of Sciences, 40 Huayanli, Chaoyang District, Beijing 100029, China

[7]Lancaster Environment Centre, Lancaster University, Lancaster, LA1 4YW, UK

[8]State Key Laboratory of Organic Geochemistry, Guangzhou Institute of Geochemistry, Chinese Academy of Sciences, 511 Kehua Street, Wushan, Tianhe District, Guangzhou, GD 510640 , China

[9] Institute of Surface-Earth System Science, Tianjin University, Tianjin 300072, China

[a]Now at: College of Environmental Sciences and Engineering, Peking University, Beijing, 100871, China

[b]Now at Department of Chemistry, Faculty of Science, York University, 4700 Keele Street, Toronto ON, M3J 1P3, Canada

*Correspondence to*: Dwayne Heard (d.e.heard@leeds.ac.uk) and Lisa Whalley (l.k.whalley@leeds.ac.uk)

**Abstract**

Wintertime *in situ* measurements of OH, HO$_2$ and RO$_2$ radicals and OH reactivity were made in central Beijing during November and December 2016. Exceptionally elevated NO was observed on occasions, up to ~250 ppbv, believed to be the highest mole fraction for which there have then co-located radical observations. The daily maximum mixing ratios for radical species varied significantly day-to-day over the range 1 - 8 x 10$^6$ cm$^{-3}$ (OH), 0.2 - 1.5 x 10$^8$ cm$^{-3}$ (HO$_2$) and 0.3 - 2.5 x 10$^8$ cm$^{-3}$ (RO$_2$). Averaged over the full observation period, the mean daytime peak in radicals was 2.7 x 10$^6$ cm$^{-3}$, 0.39 x 10$^8$ cm$^{-3}$ and 0.88 x 10$^8$ cm$^{-3}$ for OH, HO$_2$ and total RO$_2$, respectively. The main daytime source of new radicals *via* initiation processes (primary production) was the photolysis of HONO (~83 %), and the dominant termination pathways were the reactions of OH with NO and NO$_2$, particularly under polluted, haze conditions. The Master Chemical Mechanism (MCM) v3.3.1 operating within a box model was used to simulate the concentrations of OH, HO$_2$ and RO$_2$. The model underpredicted OH, HO$_2$ and RO$_2$, especially when NO mixing ratios were high (above 6 ppbv). The observation-to-model ratio of OH, HO$_2$ and RO$_2$ increased from ~ 1 (for all radicals) at 3 ppbv of NO to a factor of ~3, ~20 and ~91 for OH, HO$_2$ and RO$_2$, respectively, at ~200 ppbv of NO. The significant underprediction of radical concentrations by the MCM suggests a deficiency in the representation of gas-phase chemistry at high



$NO_x$. The OH concentrations were surprisingly similar (within 20 % during the day) inside and outside of haze events, despite $j(O^1D)$ decreasing by 50% during haze periods. These observations provide strong evidence that gas-phase oxidation by OH can continue to generate secondary pollutants even under high pollution episodes, despite the reduction in photolysis rates within haze.

## 1. Introduction

In China, especially its capital city, Beijing, air pollution and air quality are serious concerns (Tang et al., 2017). Beijing can experience severe haze episodes (Hu et al., 2014; Lang et al., 2017) with high particulate matter loadings during winter months, and high ozone episodes during the summer (Cheng et al., 2016; Wang et al., 2015). China has one of the world's fastest expanding economies and has rapidly increased its urban population to form numerous megacities. From 1980 to 2005, the fraction of the population living in urban areas of China increased from 19.6 to 40.5 %. China's economic growth has led to an increase in energy consumption, with 50% of the global demand for coal accounted for by China in 2016 (Qi et al., 2016). The Chinese government have been implementing air quality controls in China (Zhang et al., 2016a) and emission and concentrations of primary pollutants have been decreasing nationwide, however, secondary pollutants still remain a major concern (Huang et al., 2014).

The OH radical mediates virtually all oxidative chemistry during the daytime, and converts primary pollutants into secondary pollutants, as shown in Figure 1. The reaction of OH with primary pollutant emissions (particularly $NO_x$, $SO_2$ and VOCs) can form secondary pollutants such as $HNO_3$, $H_2SO_4$ and secondary oxygenated organic compounds (OVOCs). These secondary pollutants can lead to the formation of secondary aerosol and contribute to the mass of $PM_{2.5}$. During the photochemical cycle initiated by OH, NO can be oxidised to form $NO_2$ via reaction with $HO_2$ and organic peroxy radicals, $RO_2$, and the subsequent photolysis of $NO_2$ can lead to the net formation of ozone. It has been shown in previous field campaigns that measured mixing ratios of radicals have a strong dependence with $j(O^1D)$ (Ehhalt and Rohrer, 2000; Ma et al., 2019; Stone et al., 2012; Tan et al., 2018). Hence, the radical concentrations measured during wintertime are typically expected to be lower than in the summertime due to lower photolysis rates of primary radical sources such as $O_3$, HONO and HCHO. Here we define primary production as any process which initiates the formation of radicals and hence the photochemical chain reaction. Also, the lower temperatures experienced in the winter lead to lower water vapour concentrations and this is expected to further limit primary OH formation via $(O^1D) + H_2O$ (Heard and Pilling, 2003).

In contrast to the expectation of limited photochemistry in winter, particularly during haze episodes when light levels are reduced, aerosol composition analysis has highlighted that the contribution of



secondary aerosol to the total particulate mass increases during pollution events in the North China

Plain (NCP) (Huang et al., 2014), suggesting that chemical oxidation still plays an important role in

aerosol formation in winter.  To fully understand the role of the OH radical during haze events

experienced in central Beijing, direct *in situ* measurements of ambient OH concentration are required.

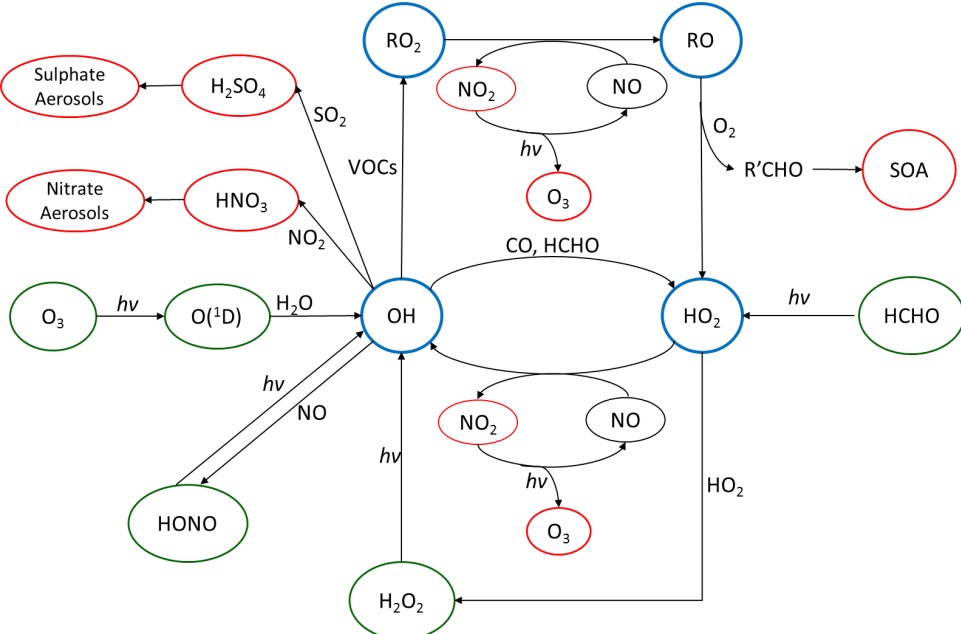

**Figure 1.** The tropospheric photochemical cycle, with the green circles representing species acting as
primary routes for radical formation via initiation reactions, the blue circles representing the radical
species themselves and the red circles representing the formation of secondary pollutants. The cycle
does not show any heterogeneous source or loss processes for the radical species.

Measurements of OH and $HO_2$ in northern China during the wintertime have only recently been made.

The first measurements were made during the BEST-ONE campaign (Tan et al., 2017) that took place

in January 2016 in Huairou, which is a suburban site 60 km northeast from Beijing. The average

daytime maximum concentrations observed during the BEST-ONE campaign for OH, $HO_2$ and $RO_2$ were

$2.5 \times 10^6$ $cm^{-3}$, $0.8 \times 10^8$ $cm^{-3}$ (3.2 pptv) and $0.6 \times 10^8$ $cm^{-3}$ (2.4 pptv) respectively. The concentration of

OH during the BEST-ONE campaign was an order of magnitude higher than predicted by global models

over the North China Plain region (Lelieveld et al., 2016), and is consistent with the increase in

secondary aerosol contribution to $PM_{2.5}$ observed during haze events (Huang et al., 2014). The radical

measurements during the BEST-ONE campaign were separated into clean and polluted periods (OH

reactivity ($k_{OH}$) > 15 $s^{-1}$) with an average daily maximum OH concentration for these periods of

$4 \times 10^6$ $cm^{-3}$ and  $2.3 \times 10^6$ $cm^{-3}$, respectively. The RACM2-LIM1 (Regional Atmospheric Chemistry

Model coupled with Leuven Isoprene Mechanism 1) box model was used to simulate the radical



concentrations measured during BEST-ONE (Tan et al., 2018) but these could not reproduce the OH concentration observed when NO was above 1 ppbv or below 0.6 ppbv; consistent with previous campaigns when OH was measured and modelled under NO concentrations > 1 ppbv (Emmerson et al., 2005; Kanaya et al., 2007; Lu et al., 2013; Tan et al., 2017; Zhou et al., 2003). More recently, OH and $HO_2$ were measured in central Beijing during winter-time at the Peking University (PKU) campus

in November/December 2017 (Ma et al., 2019). The radical measurements were simulated using the RACM2-LIM1 box model which highlighted an under-prediction of the OH concentration when NO exceeded 1 ppbv (Ma et al., 2019). Two further campaigns have taken place in northern China during the summertime. The first took place in 2006 at a suburban site in Yufa (Lu et al., 2013), which is 40 km south of Beijing. The second took place in 2014 at the rural site in Wangdu (Tan et al., 2017). In

both the Wangdu and Yufa field campaigns, the box model calculations underestimated the OH concentration when NO was below 0.5 ppbv. When NO exceeded 2 ppbv, a missing peroxy radical source was found, leading to a large underestimation of local ozone production by the model.

To try to understand the link between radical chemistry and the extremely high air pollution that is seen during Beijing in the wintertime, a field campaign "Air Pollution and Human Health in Chinese

Megacities" (APHH) took place in central Beijing from November to December in 2016. Simultaneous measurements of OH, $HO_2$, and $RO_2$ concentrations were performed during the APHH campaign. OH reactivity ($k$(OH)), which is the sum of the concentration of species ($X_i$) that react with OH multiplied by the corresponding bimolecular rate coefficient, $k_{OH+Xi}$, along with other trace gas and aerosol measurements were made alongside the radicals.

In this paper we present the measurements of OH, $HO_2$, $RO_2$ and OH reactivity from the winter campaign. The concentrations of the radical species are compared to model results from the Master Chemical Mechanism (MCM3.3.1.) to assess if the radical concentrations can be simulated across the range of measured $NO_x$, with a particular focus under on the high $NO_x$ conditions that were experienced. The importance of OH-initiated oxidation processes on the formation of ozone and SOA

in the wintertime in Beijing are demonstrated.

**2 Experimental**

**2.1 Location of the field measurement site**

The observations took place in central Beijing at the Institute of Atmospheric Physics (IAP), which is part of the Chinese Academy of Sciences; the location of the site is shown in Figure 2, and is ~ 6.5 km

from the Forbidden City. Beijing is the capital city of China and is located on the northwest border of the North China Plain (NCP). It is surrounded by the Yanshan Mountains in the west, north and northeast (Chan and Yao, 2008). The topography of Beijing allows for the accumulation of pollutants,



especially when southerly winds carrying emissions from the industrial regions are experienced. As shown by Figure 2, the measurement site was within 100m of a major road, thus local anthropogenic

emissions likely influence the site, although no rush hour was observed from the diel variation of the trace gas measurements (see Figure 5). The site was also close to local restaurants and a petrol station. More details of the measurement site and instrumentation can be found in the APHH overview paper (Shi et al., 2018).

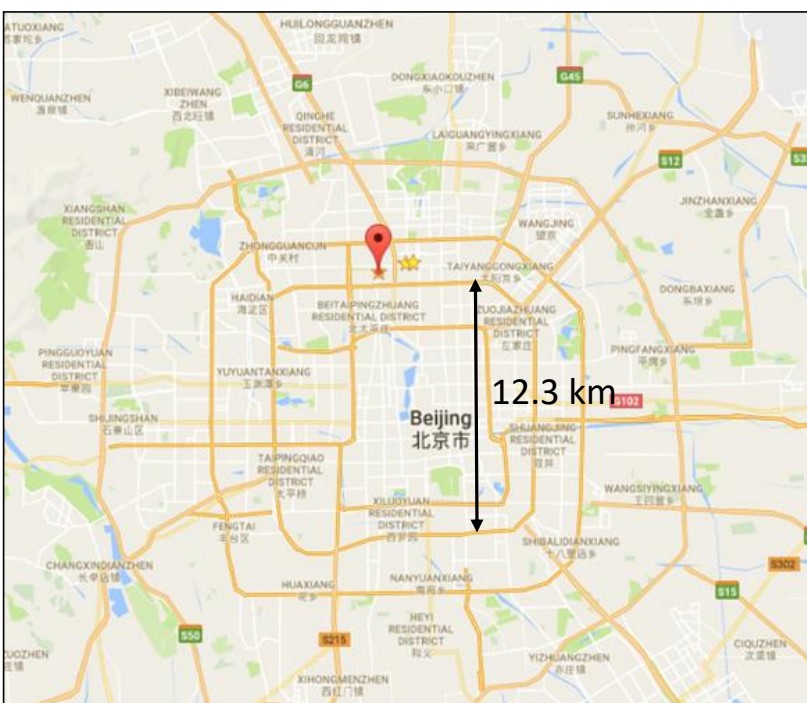

**Figure 2.** Location of the Institute of Atmospheric Physics, Chinese Academy of Sciences (source: ©Google Maps), the location (39°58'33'' N, 116°22'41'' E) of the APHH campaign.

### 2.2 Instrumental details

### 2.2.1 OH, $HO_2$ and $RO_2$ measurements

The University of Leeds ground-based FAGE (fluorescence assay by gas expansion) instrument

(Whalley et al., 2010) was deployed at the IAP site and made measurements of OH, $HO_2$ and $RO_2$ radicals, as well as OH reactivity ($k$(OH)). A general outline, specific set-up and the running conditions during APHH are described here. Further details on the methodology for sequential measurements of OH and $HO_2$ that are made in the first fluorescence cell ($HO_x$) and sequential measurements of $HO_2^*$ and $RO_2$ using the $RO_x$LIF method (described in detail below) in the second cell ($RO_x$) can be found in

Whalley et al. (2018). $HO_2^*$ refers to the measurement of $HO_2$ and complex $RO_2$ species; complex $RO_2$





are $RO_2$ species that are formed from alkene and aromatic VOCs, or VOCs that have a carbon chain greater than $C_4$ and which under certain conditions are detected together with $HO_2$ (Whalley et al., 2018). The radical measurements were made from a 6.1 m air-conditioned shipping container which has been converted into a mobile laboratory. The FAGE instrument has two detection cells which are

located on top of the shipping container (sampling height of 3.5 metres) within a weather-proof housing. A Nd:YAG pumped Ti:Sapphire laser (Photonics Industries) generated pulsed tuneable near IR radiation at a pulse repetition rate of 5 kHz, which was frequency doubled then tripled using two non-linear crystals to produce UV light at 308 nm and used to excite OH via the $Q_1(1)$ transition of the $A^2\Sigma^+$, v'=0 $\leftarrow$ $X^2\Pi_i$, v''=0 band.

During the APHH campaign the configuration of the two detection cells was the same as deployed during the ClearfLo campaign in London (Whalley et al., 2018), with the two cells coupled together via a connecting side arm, which enabled the laser light exiting the $HO_x$ cell to pass directly into the $RO_x$ cell. The channel photo-multiplier (CPM) detectors that were used to detect fluorescence previously (Whalley et al., 2018) have been replaced by gated MCPs (micro-channel plates, Photek

PMT325/Q/BI/G) and fast gating units, Photek GM10-50B) for the AIRPRO project.

### 2.2.1.3 Inlet Pre Injector

For part of the campaign, an Inlet-pre-injector (IPI) was attached to the $HO_x$ cell. The IPI removes ambient OH by the injection of propane directly above the cell inlet and facilitates a background measurement whilst the laser wavelength is still tuned to an OH transition, with this type of OH

measurement known as "OHchem". The OHchem background signal will include a signal from laser scattered light, scattered solar radiation and may potentially also include a fluorescence signal from any OH that is generated internally from an interference precursor within the LIF cell. Internally generated OH constitutes an interference, but can be readily identified by comparing the OHchem background signal to the background signal measured when the laser wavelength is tuned away from

the OH transition, with this type of OH measurement known as "OHwave". The OHwave background signal is from laser scattered light and solar scattered radiation only.

The Leeds IPI was first implemented during the ICOZA campaign in Norfolk, UK, in the summer of 2015, and is described in further detail elsewhere (Woodward-Massey et al., 2019). During the APHH winter campaign the laser online (wavelength tuned to the OH transition) period lasted 300 seconds for both

OHchem and OHwave data acquisition cycles. When the IPI was physically taken off the $HO_x$ fluorescence cell, OH and $HO_2$ were measured sequentially in this cell with 150 seconds online period each. The other ($RO_x$) fluorescence cell measured $HO_2^*$ and $RO_2$ simultaneously with OH and $HO_2$, respectively, when the IPI was removed. When the IPI was being operated during the APHH campaign

OHwave, OHchem and HO$_2$ were measured in the HO$_x$ cell sequentially for 120, 120 and 60 seconds, respectively. The RO$_x$ cell measured HO$_2^*$ and RO$_2$ for 240 and 60 seconds, respectively when the IPI was operated. The laser offline period for both data acquisition cycles lasted 30 seconds, with NO injected for the final 15 seconds of this laser offline period. From the 08/11/2016 to 24/11/2016 the HO$_x$ cell was operated without the IPI assembly in place, the IPI was then installed and run on the HO$_x$ cell from 02/12/2016 to 08/12/2016.

The correlation of OHwave and OHchem during the APHH winter campaign is shown in Figure 3. The slope of 1.05±0.07 demonstrates that within the errors in the linear fit no interference was evident during the winter campaign. OHwave data were corrected for the known interference from O$_3$ + H$_2$O, see (Woodward-Massey et al., 2019) for further details. All figures and calculation from now on have used OHwave as it is the most extensive time-series (12 days compared to 5 days).

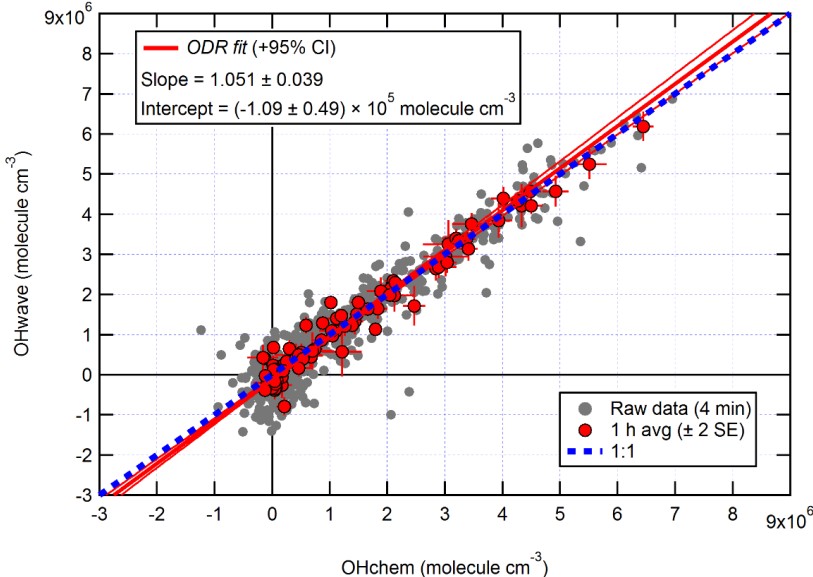

**Figure 3.** Overall intercomparison of OHwave and OHchem observations from the winter 2016 AIRPRO campaign. Grey markers represent raw data (6 min acquisition cycle), with 1 h averages (±2 standard error, SE) in red. The thick red line is the orthogonal distance regression (ODR) fit to the hourly data, with its 95% confidence interval (CI) bands given by the thin red lines; fit errors given at the 2σ level. For comparison, 1:1 agreement is denoted by the blue dashed line. OHwave data were corrected for the known interference from O$_3$ + H$_2$O. Taken from (Woodward-Massey et al., 2019) where further details can be found.

### 2.2.2 Calibration

The instrument was calibrated approximately every three days by photolysis of a known concentration of water vapour at 185 nm in synthetic air (Messer, Air Grade Zero 2) within a turbulent flow tube to




generate equal concentrations of OH and $HO_2$ as described in Whalley et al. (2018). The product of the photon flux at 185 nm and the water vapour photolysis time, which is required to calculate the concentration of OH and $HO_2$, was measured using a $N_2O \rightarrow NO$ chemical actinometer (Commane et al., 2010) both before and after the APHH campaign. For calibration of $RO_2$ concentrations, methane

(Messer, Grade 5, 99.99%) was added to the humidified air flow in sufficient quantity to rapidly titrate OH completely to $CH_3O_2$. For reporting the total concentration of $RO_2$ the calibration factor for $CH_3O_2$ was used. More details on the $RO_x$LIF and calibration, for example the sensitivity of the instrument towards various $RO_2$ species which is taken into account when comparing $RO_2$ measurements to model calculations, can be found in Whalley et al. (2018). The limit of detection (LOD) on average for the

APHH campaign was $5.5 \times 10^5$ molecule $cm^{-3}$ for OH, $3.1 \times 10^6$ molecule $cm^{-3}$ for $HO_2$ and $6.5 \times 10^6$ molecule $cm^{-3}$ for $CH_3O_2$ at a typical laser power of 11 mW for a 7 minute data acquisition cycle (SNR=2). The field measurements of all species were recorded with 1 s time-resolution, and the precision of the measurements was calculated using the standard errors in both the online and offline points. The accuracy of the measurements was ~ 26 % ($2\sigma$), and is derived from the error in the

calibration, which derives largely from that of the chemical actinometer (Commane et al., 2010).

**2.2.3 The Master Chemical Mechanism, MCM**

A constrained zero-dimensional (box) model incorporating version 3.3.1 of the Master Chemical Mechanism (MCMv3.3.1) ([http://mcm.leeds.ac.uk/MCM/home](http://mcm.leeds.ac.uk/MCM/home)) was used to predict the radical concentrations and OH reactivity and to compare with the field observations. The MCM is a detailed

mechanism that almost explicitly describes the oxidative degradation of ~ 140 VOCs ranging from methane to those containing 12 carbon atoms (C1 – C12). The complete details of the kinetic and photochemical data used in the mechanism can be found at the MCM website ([http://mcm.leeds.ac.uk/MCM/home](http://mcm.leeds.ac.uk/MCM/home)). For this work, the model was run with a sub-set of the MCM and treated the degradation of simultaneously measured non-methane VOCs, $CH_4$ and CO following

oxidation by OH, $O_3$ and $NO_3$, and included 11,532 reactions and 3,778 species. The model was constrained by measurements of NO, $NO_2$, $O_3$, CO, HCHO, $HNO_3$, HONO, water vapour, temperature, pressure and individual VOC species measured by GC-FID (gas chromatography with flame ionisation).

Table 1 shows the different species measured by the GC-FID whose degradation was included in the mechanism used. The model was constrained with the measured photolysis frequencies $j(O^1D)$, $j(NO_2)$

and $j(HONO)$), which were calculated from the measured wavelength-resolved actinic flux and published absorption cross sections and photodissociation quantum yields. For other species which photolyse at near-UV wavelengths, such as HCHO and $CH_3CHO$, the photolysis rates were calculated by scaling to the ratio of clear-sky $j(O^1D)$ to observed $j(O^1D)$ to account for clouds. For species which photolyse further into the visible the ratio of clear-sky $j(NO_2)$ to observed $j(NO_2)$ was used. The


variation of the clear-sky photolysis rates ($j$) with solar zenith angle ($\chi$) was calculated within the model using the following expression Eq. 1:

$$j = l \cos(\chi)^m \times e^{-n \sec(\chi)} \qquad \text{Eq. 1}$$

with the parameters $l$, $m$ and $n$ optimised for each photolysis frequency (see Table 2 in Saunders et al. (2003).

A constant $H_2$ concentration of 500 ppbv was assumed (Forster et al., 2012). The model inputs were
updated every 15 minutes, the species that were measured more frequently were averaged to 15 minutes whilst the measurements with lower time resolution were interpolated. The loss of all non-constrained, model generated species by deposition or mixing was represented as a first order deposition rate equivalent to 0.1/MH (MH represent the height of the boundary layer). The effect of changing the deposition rate is minor, as shown in Figure S1 of the Supplementary Information. The
model was run for the entirety of the campaign in overlapping 7 day segments. To allow all the unmeasured, model generated intermediate species time to reach steady state concentrations, the model was initialised with inputs from the first measurement day (16$^{th}$ November 2016) and spun-up for 2 days before comparison to measurements were made. The model described above is from now on called MCM-base.

An additional model was run using higher weight VOCs that were measured using a PTR-MS (Proton Transfer Mass Spectrometer) to assess the effect on modelled radical species (OH, HO$_2$ and RO$_2$) and modelled OH reactivity, with this model run showing there is <10% effect on the radical concentration and OH reactivity (see Supplementary Information, Figures S2 and S3).

| Instrument | Species | Reference |
|---|---|---|
| DC-GC-FID | Methane, Ethane, ethylene, propane, propene, isobutane, butane, $C_2H_2$, trans-but-2-ene, but-1ene, Isobutene, cis-but-2-ene, 2-Methylbutane, pentane, 1,3-butadiene, trans-2-pentene, cis-2-pentene, 2-methylpetane, 3-methypetane, hexane, isoprene, heptane, Benzene, Toluene, m-xylene, p-xylene, o-xylene, methanol, dimethyl ether. | Hopkins et al. (2011) |


**Table 1.** VOC species measured by the DC-GC-FID (dual channel gas-chromatography with flame ionisation detection) that have been constrained in the box-model utilising the Master Chemical Mechanism.





The model scenarios involved in this work are summarised in Table 2.

| Model Name | Description |
| --- | --- |
| MCM-base | The base model described above in Section 2.2.3. |
| MCM-cHO$_2$ | The same as MCM-base, but with the model constrained to the measured value of the HO$_2$ concentration. |
| MCM-PRO2 | The same as MCM-base, but including an extra primary source of RO$_2$ species to reconcile the measured total RO$_2$ with modelled RO$_2$. Details for this can be found in section 4.2. |
| MCM-PRO2-SA | The same as MCM-PRO2 but including the uptake of HO$_2$ to aerosol with an uptake coefficient of $\gamma$ = 0.2 Jacob et al.(2000). |

**Table 2.** Description of the model scenarios and how they differ from the base model, and the associated name of that model that has been used in the body of this work.


### 3 Results

### 3.1 Chemical and Meteorological conditions

During the campaign various chemical and meteorological conditions were observed, as shown in Figure 4, including several haze periods. According to the meteorological standards (QX/T113-2010,
Shi et al. (2018)), haze is defined as (i) visibility < 10 km at relative humidity (RH) < 80 % or (ii) if RH is between 80 and 95 %, visibility < 10 km and PM2.5 > 75 µg m$^{-3}$. For the purpose of this work the periods defined as haze are when PM$_{2.5}$ exceeds 75 µg m$^{-3}$. The wind rose for the winter 2016 campaign shows the dominant wind direction is from the northwest which coincides with higher wind speeds, also south westerly flows were frequent in the winter APHH campaign (see Shi et al. (2018) for more
details). The south-westerly wind direction observed in the winter 2016 campaign had the potential to bring more polluted air from the upwind Hebei province to the observations site in Beijing.

The diel variation in $j$(O$^1$D), relative humidity (RH), temperature, CO, SO$_2$, O$_2$, NO,NO$_2$, HONO and PM$_{2.5}$ is shown in Figure 4. There were several co-located measurements of HONO made during the APHH campaign, and the HONO mixing ratios shown in Figure 4 and used in the model were values taken
from a combination of all measurement at the IAP site, and recommended by Crilley et al. (2019) who provide further details for the methodology for selection of the HONO data. For a given time of day, large variations in $j$(O$^1$D) during the campaign were observed, with the reductions caused by decreasing light levels driven by enhanced PM$_{2.5}$. The temperature during the campaign varied between -10°C and +15°C. The relative humidity during the campaign varied between 20 – 80% RH;
generally with higher RH coinciding with haze events. The time-series for trace gas species showed high mole fractions for CO (1000-4000 ppbv), SO$_2$ (5 – 25 pbbv), NO (20 – 250 ppbv) but relatively low



O$_3$ (1 – 30 ppbv). HONO during the campaign was generally quite high reaching up to 10 ppbv (Crilley et al., 2019). Frequent haze events were also observed during the winter campaign, with PM$_{2.5}$ mass concentration reaching up 530 µg m$^{-3}$.

The diel variation for $j$(O$^1$D), NO, NO$_2$, O$_3$, O$_x$, HONO, SO$_2$ and CO separated into haze and non-haze periods is shown in Figure 5; the periods defined as haze are shown in Table 3. During the haze events $j$(O$^1$D) decreased by ~50% at midday, as shown in Figure 5. The photo-activity of $j$(HONO) and $j$(NO$_2$) extends further into the visible region of the solar spectrum compared with $j$(O$^1$D) and so the reductions in their photolysis rates within haze are less; ~40% for $j$(HONO) and ~35% for $j$(NO$_2$) as

discussed in (Hollaway et al., 2019). During polluted and hazy periods NO on average reached 100 ppbv at 8 am; on some days NO was close to 250 ppbv, some of the highest levels ever recorded during an urban field campaign. On clearer days, the peak NO was ~ 40 ppbv at 8 am (CST). A distinct increase in CO, NO$_2$ and SO$_2$ was also observed during haze periods, but no clear diurnal pattern in and outside of haze for these species was observed, as shown in Figure 5. The O$_3$ during the haze periods reduced

on average by a factor of 3, due to titration by reaction with the high levels of NO observed. NO and O$_3$ show an anti-correlation during the cleaner periods due to their inter-conversion. The sum of NO$_2$ and O$_3$, O$_x$, increased during pollution periods from 40 ppbv to a maximum of 53 ppbv on average. HONO in both clean and haze periods shows a distinct diel pattern, with a large decrease in the morning from loss through photolysis and a minimum in the afternoon; a large increase in HONO

concentration overnight probably originates from heterogeneous sources (i.e. NO$_2$ converting to HONO on humid surfaces) (Finlayson-Pitts et al., 2003; Lee et al., 2015; Li et al., 2012; Lu et al., 2018; Zhang et al., 2016b; Zhou et al., 2003). The HONO concentration was a factor of 3 higher on average during haze periods at midday than during the clearer periods.

| Haze Event | Local Time | PM$_{2.5}$ (µg m$^{-3}$) | Visibility (km) |
|---|---|---|---|
| Event 1 | 08/11 21:00 – 10/11 16:00 | 158 (79 – 229) | 4.1 (2.3 – 8) |
| Event 2 | 15/11 21:00 – 19/11 08:00 | 143 (56 – 244) | 4.2 (0.6- 8) |
| Event 3 | 24/11 12:00 – 27/11 02:00 | 210 (68 – 363) | 4.2 (1.5 -8) |
| Event 4 | 02/12 16:00 – 05/12 02:00 | 239 (58 – 530) | 3.9 (0.9 -8) |
| Event 5 | 06/12 09:00 – 08/12 10:00 | 144 (64 – 229) | 4.6 (2.2 – 8) |


**Table 3.** The different haze periods observed during the winter campaign. Table recreated from Shi et al. (2018), from which further details can be found.

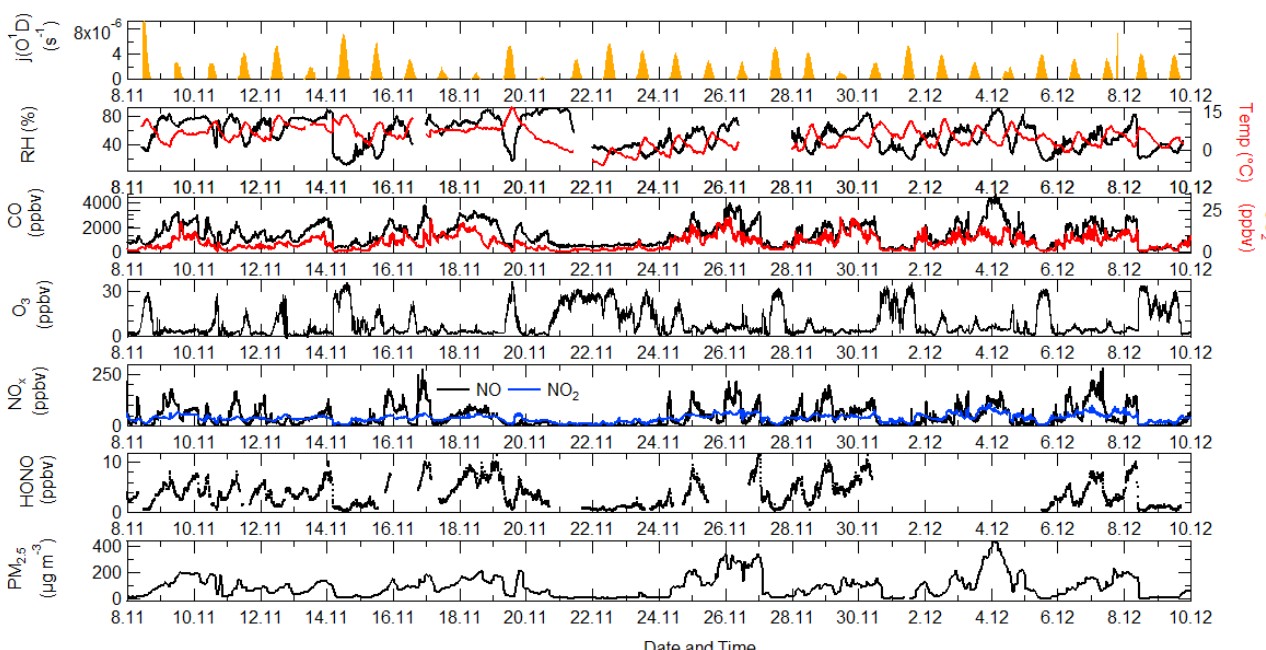

314

**Figure 4.** Time-series of $j(O^1D)$, relative humidity (RH), temperature (Temp), CO, SO$_2$, O$_3$, NO$_x$, HONO and PM$_{2.5}$ from the 8th of November to 10th December
2016 at Institute of Atmospheric Physics (IAP), Beijing.

315
316

317

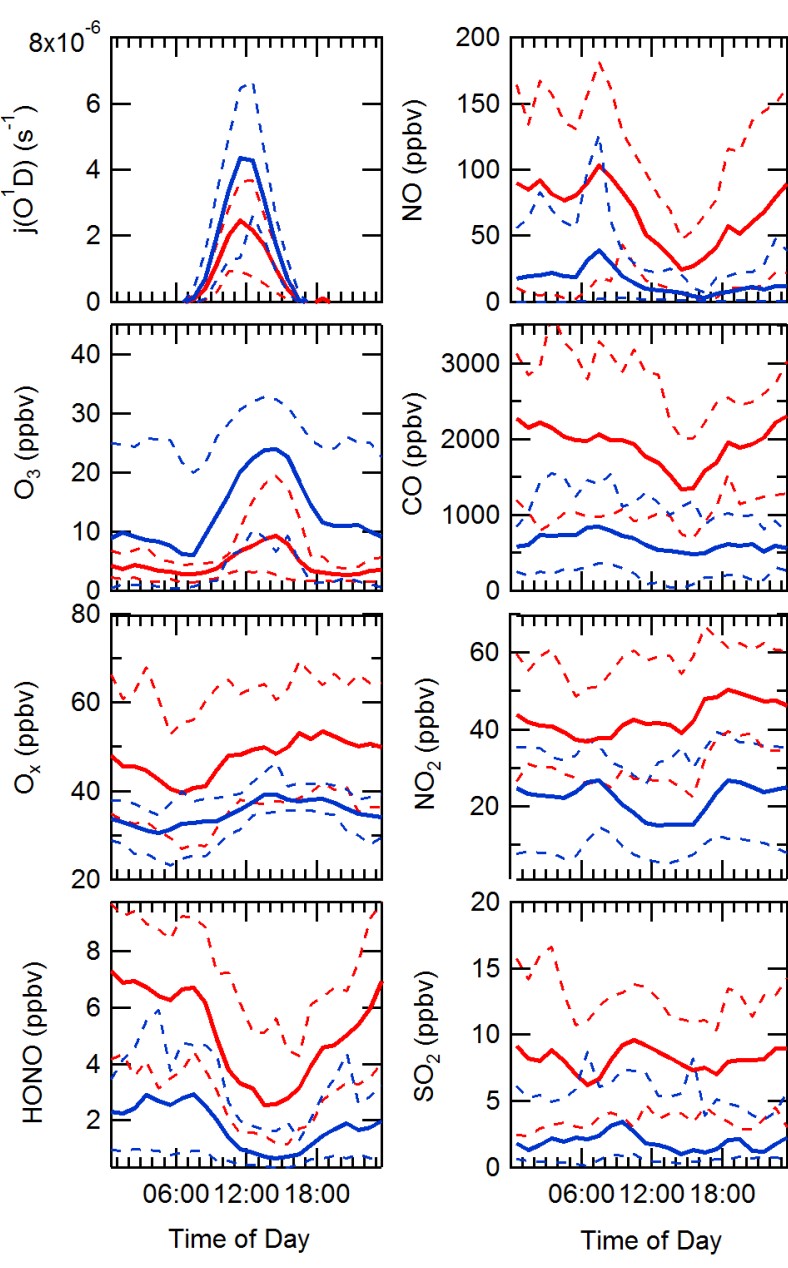

**Figure 5.** Comparison of $j$(O$^1$D) (s$^{-1}$), NO (ppbv), O$_3$ (ppbv), CO (ppbv), O$_x$ (ppbv), NO$_2$ (ppbv), HONO (ppbv) and SO$_2$ (ppbv) in and outside haze events; denoted by solid red and blue lines, respectively. The dashed lines represent the 25/75 percent confidence interval for the respective species and pollution period.

320





### 3.2 Steady State calculation of OH

Using measured quantities, a steady state approach has been used to calculate the OH concentrations for comparison with measurements, and also to determine the major sources of OH measured during the campaign. The photostationary steady state equation for OH, obtained from $d[OH]/dt = 0$, is given by a balance of the rate of production and the rate of destruction of OH:

$$[OH]_{pss} = \frac{p(OH) + j(HONO)[HONO] + k[HO_2][NO]}{k(OH)}$$
Eq. 2

where p(OH) is the measured rate of OH production from ozone photolysis and the subsequent reaction of $O(^1D)$ with water vapour, $k$ is the rate coefficient for the reaction of $HO_2$ with NO at the relevant temperature, and $k(OH)$ is the measured OH reactivity. Equation (2) is a simplification, and only takes into account the production of OH from two photolysis sources ($O_3$ and HONO) and from the reaction of $HO_2$ + NO. $O_3$+alkene and $HO_2$ + $O_3$ reactions are not included as, owing to the generally low ozone experienced, these were found to contribute < 1 % to the total OH production, as discussed in the MCM modelling section below. The pseudo-first order rate of loss of OH was constrained using the measured OH reactivity during the campaign, and hence includes all loss processes for OH. OH reactivity is discussed further in Section 2.5.

Figure 6 shows the steady state calculation for OH between 2/12/2016 to 8/12/2016 where it is compared with the measured OH concentrations. These days were chosen as full data coverage for HONO, NO, $j$ values, radical and $k(OH)$ measurements were available. The agreement between the observed OH and OH calculated by equating the rate of OH produced from $HO_2$+NO and HONO photolysis and the loss of OH by reaction with all of its sinks, Eq.2, is very good. The agreement highlights that the OH budget can be determined by field measurements of the parameters necessary to quantify its rate of production and loss, and is closed to within 10%, well within the 26% error on the OH measurements themselves. The closure of the experimental budget suggests that measured OH and $HO_2$ are internally consistent, and that just from measured quantities the rate of production and the rate of destruction are the same within uncertainties. The reaction of $HO_2$ and NO is the dominant source of OH (~80 − 90%) for Beijing during wintertime, owing to NO being so high in concentration. The photolysis of HONO is the second most important source producing ~10 − 20% of OH (and a much larger primary source of radicals in general as discussed below). Due to low concentrations of $O_3$ in winter, the photolysis of $O_3$ and the subsequent reaction of $O(^1D)$ with water vapour is not an important source, being < 1 % of the rate of production. In addition, the reaction of $O_3$ with alkenes (whose concentrations were elevated in the winter) also contributed < 1% to the rate of OH production.

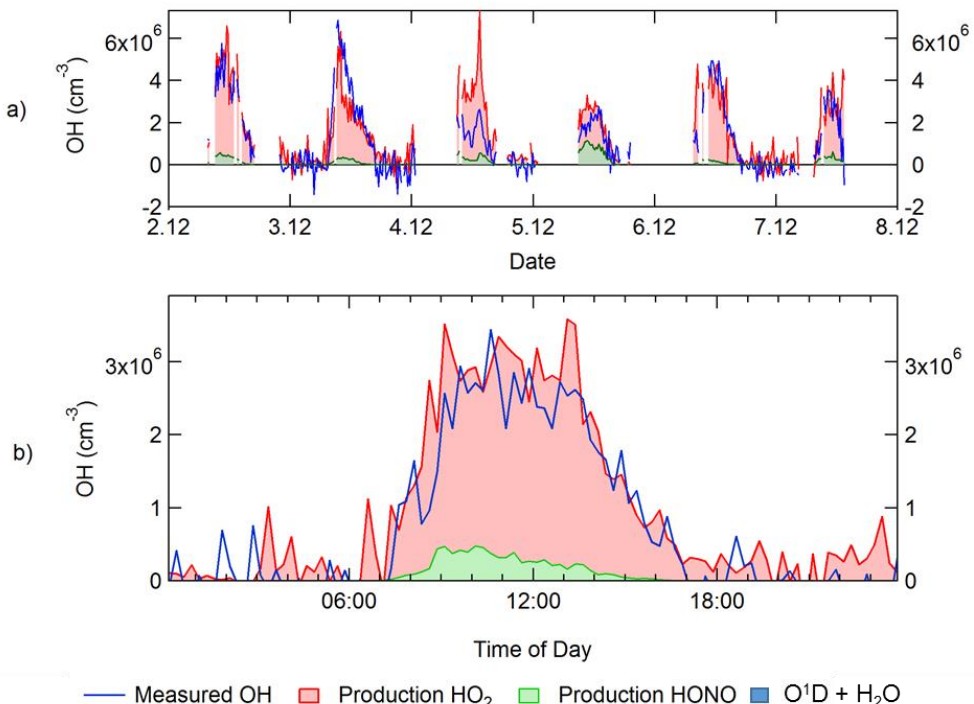

**Figure 6.** a) Time series for the steady state calculation of OH using equation (2). b) Average diel profile for observed and steady state calculated OH. Production $HO_2$ represents the recycling of $HO_2$ to OH *via* NO, Production HONO represents OH production from HONO photolysis. The OH generated by $O^1D+H_2O$, although included in the key, is too small to be visible.

### 3.3 Comparison of measured OH, $HO_2$, $RO_2$ radical concentrations and OH reactivity with calculations using a box-model and the Master Chemical Mechanism

Figure 7 shows a comparison between measured and modelled (MCM-base, defined in Table 2) OH, $HO_2$, $RO_2$ (speciated into simple and complex $RO_2$, defined in section 2.2.1) and OH reactivity. As seen in Figure 7, the measured daily maximum for the radical species varied day-to-day over the range 1 to 8 x $10^6$ $cm^{-3}$, 0.7 to 1.5 x $10^8$ $cm^{-3}$ and 1 to 2.5 x $10^8$ $cm^{-3}$ for OH, $HO_2$ and sum of $RO_2$ respectively. The daily maximum concentration for the sum of simple $RO_2$ varied between 0.2 to 1.3 x $10^8$ $cm^{-3}$, and the complex $RO_2$ daily maximum concentration varied between 0.2 and 0.6 x $10^8$ $cm^{-3}$. On average, the model underpredicts the OH, $HO_2$ and $RO_2$ concentrations by a factor of 1.7, 5.8 and 25, as shown in Figure 8. Although the underprediction by the model varies day-to-day: for OH, the underprediction varies from a factor of 5.9 to an overprediction of 1.05 (showing good agreement) between the model and measurements; for $HO_2$ the underprediction varies from a factor of 13.6 to an over prediction by a factor of 5.3 and for $RO_2$ the under prediction varies from a factor of 2.1 to an over prediction of 8.0. Figure 8 shows the diel profile of OH, $HO_2$ and $RO_2$ averaged over the campaign, with daily average



375  maximum of $2.7 \times 10^6$ $cm^{-3}$, $0.39 \times 10^8$ $cm^{-3}$ and $0.88 \times 10^8$ $cm^{-3}$ for OH, $HO_2$ and total $RO_2$, respectively. The total measured OH reactivity during the campaign was quite large and varied between 10 to 145 $s^{-1}$. Averaged over the full campaign period the contributions to reactivity came from CO (17.3%), NO (24.9%), $NO_2$ (22.1%), alkanes (3.0%), alkynes and alkenes (10.8%), carbonyls (5.7%), terpenes (3.7%) and modelled intermediates (6.77%). Unusually, the largest contribution to OH reactivity is from

380  reaction with NO. As shown in Figure 7 and Figure 8, the OH reactivity is reproduced within 10% implying that the OH reactivity budget is captured well by the model. The model OH reactivity is the sum of all measured and modelled intermediate species multiplied by the respective rate coefficient for their reaction with OH.

Consistent with the steady state calculation, and as shown also in Figure 8, when the box-model was

385  constrained to the concentrations of $HO_2$ measured using FAGE in the field (from now on this model scenario is called MCM-$cHO_2$), the measured and modelled OH concentration are in agreement within 10% which is less than the 26% error on the OH measurements. The $HO_2$ was constrained in the model by inputting the $HO_2$ concentration at every 15 minute time-step.

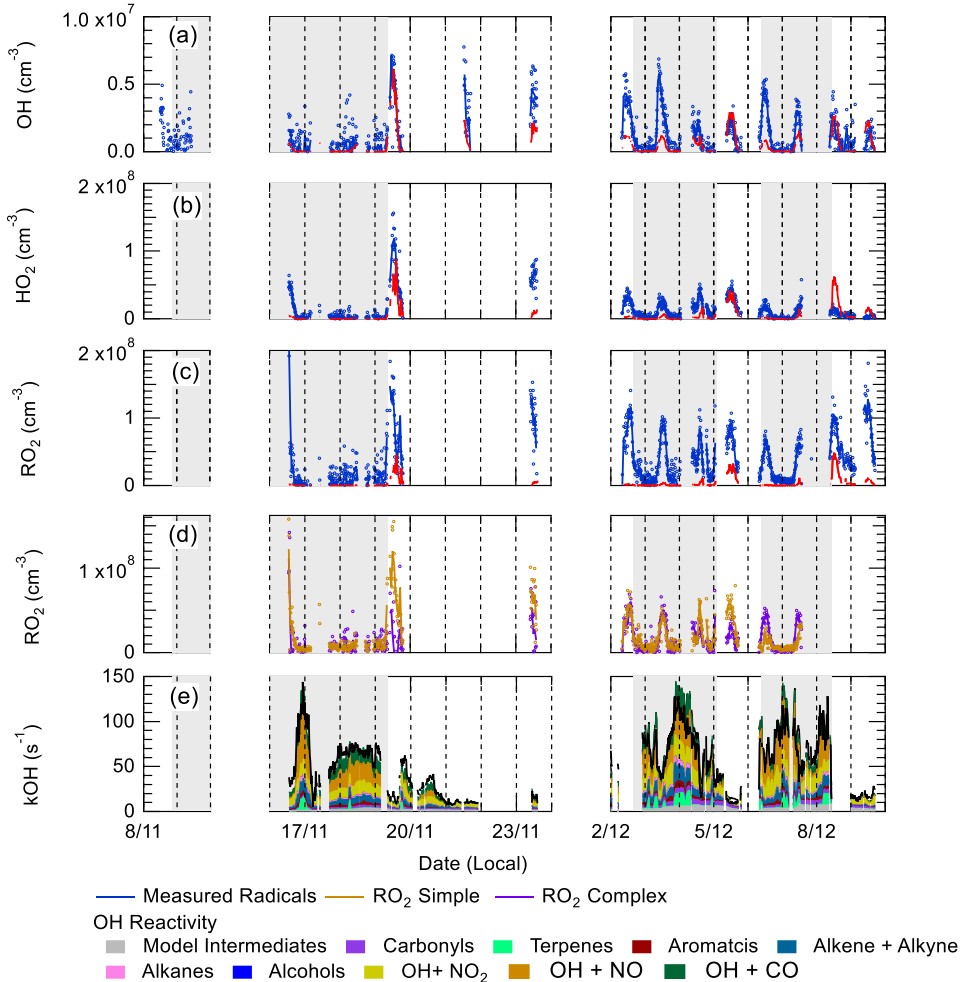

**Figure 7**. Time-series of OH, b) HO₂, c) total RO₂, d) partly-speciated RO₂ and e) OH Reactivity. For (a)-(c), the raw measurements (6-min data acquisition cycle) are blue open circles with 15 min average represented by the solid blue line. The 15 min model output in a-c is represented by the red line for OH, HO₂ and RO₂. The partly-speciated RO₂ is separated into simple (gold open circles) and complex (purple open circles), with the model in the same colour (solid line). The individual contributions of the model to the OH reactivity is given below the graph. The grey shaded areas show the haze periods when PM₂.₅ > 75 µg m⁻³.



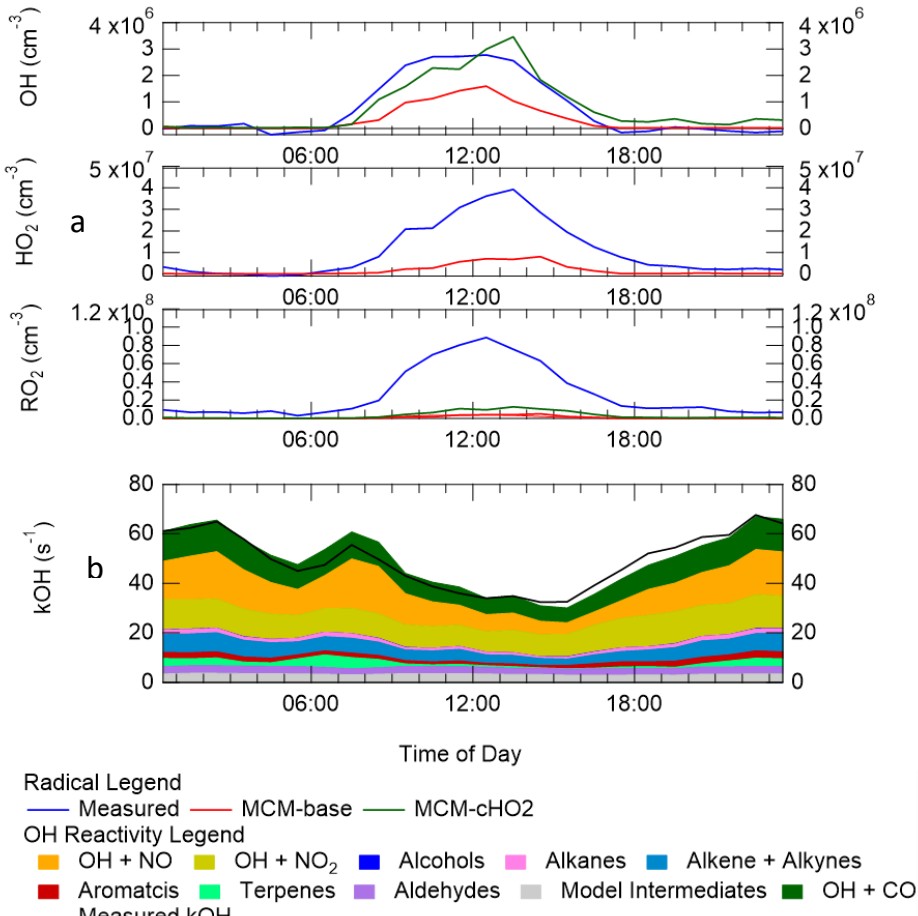

**Figure 8.** (a)– Campaign averaged diel profile of OH (top panel), HO$_2$ (middle panel) and sum of RO$_2$ (lower panel) for measurements (blue) and box-model calculations: MCM-base (red) and MCM-cHO$_2$ (green) See text for descriptions of each model scenario. (b) – OH reactivity (s$^{-1}$) for measurements (black line) and model (stacked plot) with the contribution to reactivity from different measured species and modelled intermediates shown in the key.

The ability of the model to reproduce (to within ~10%) both the OH reactivity and the OH concentration when constrained to measured HO$_2$ (in MCM-cHO$_2$), but not to reproduce RO$_2$ radicals (whether constrained or not to HO$_2$) is suggestive of an incomplete representation of the chemistry of RO$_2$ radicals in the winter Beijing environment. The significant model underprediction of RO$_2$ implies either that additional sources of RO$_2$ radicals are required, or that it is inaccuracies in the recycling chemistry within RO$_2$ species which leads to an overestimate of the loss rate of RO$_2$ under the high NO$_x$ conditions experienced in central Beijing. The cause of the model underprediction of RO$_2$ is explored further in section 4.



As summarised in Table 4, previous winter campaigns, where the environment controlling peroxy radicals is generally dominated by NO, have shown a similar underprediction of radical species at high levels of $NO_x$ (above 3 ppbv of NO) (Lu et al., 2013; Ma et al., 2019; Tan et al., 2017; Tan et al., 2018). For the BEST-ONE campaign, which took place in suburban Beijing (~60 km from the centre) it was suggested that in order to reconcile the model with the measurements, an additional source of $RO_2$

was required.

The OH concentrations measured are surprisingly high for a winter campaign where photolysis rates and RH are low; the average 12:00 OH maximum for the campaign was $3.3 \times 10^6$ molecule $cm^{-3}$. Comparisons with the level of agreement between measured and modelled radicals for other winter field campaigns are given in Table 1. The OH concentration is ~ 3, 2.3, 2, 1.65 and 1.5 times larger than

winter measurements in New York (Ren et al., 2006), Beijing (Ma et al., 2019), Tokyo (Kanaya et al., 2007), Birmingham (Emmerson et al., 2005) and the BEST-ONE (Tan et al., 2018) campaigns, respectively, and similar to the campaign in Boulder (Kim et al., 2014). However, it should be noted that the Boulder campaign took place at a time in the year (late February/March) closer to mid-summer when there are higher light levels and water vapour (see Table 5 for details). As shown in

Figure 7, the elevated OH concentrations inside haze events, for example up to $6 \times 10^6$ molecule $cm^{-3}$ of OH was observed on 03/12/2016, suggests gas-phase oxidation is still highly active (this is explored more in section 4.3 and 4.4).



| Campaign | Months, Year | NO (ppbv) | O$_3$ (ppbv) | OH | | HO$_2$ | | RO$_2$ | | Notes | References |
|---|---|---|---|---|---|---|---|---|---|---|---|
| | | | | Measured ($10^6$ cm$^{-3}$) | Obs/ Model | Measured ($10^8$ cm$^{-3}$) | Obs/ Model | Measured ($10^8$ cm$^{-3}$) | Obs/ Model | | |
| AIRPRO, Central Beijing, China | Nov – Dec, 2016 | 60 | 12 | 2.7 | 1.7 | 0.39 | 5.9 | 0.88 | 25 | Average midday. | This work. |
| BEST-ONE Suburban Beijing, China | Jan – March, 2016 | 7 | 30 | 2.2 | 2 | 0.5 | 2.5 | 0.7 | 5 | Campaign Median, midday, polluted period | Tan et al. 2018 |
| NCITT Boulder, USA | Late Feb, 2011 | 7 | 37 | 3 | 1.1 | - | - | - | - | Average midday | Kim et al. (2014) |
| PUMA, Birmingham, UK | Jan-Feb, 2000 | 10 | 13 | 2 | 0.50 | 3 | 0.49 | - | - | Average midday | Emmerson et al. (2005) |
| IMPACT Tokyo, Japan | Jan-Feb, 2004 | 8.1 | 35 | 1.5 | 0.93 | 0.27 | 0.88 | - | - | Average midday | Kanaya et al. (2007) |
| PMTACS-NY2001 New York, US | Jan–Feb, 2004 | 25 | 20 | 1 | 0.83 | 0.17 | 0.17 | - | - | Average midday | Ren et al. (2006) |
| PKU | Nov – Dec, 2017 | 30 | 10 | 1.4 | 1.4 | 0.3 | 7.5 | - | - | Average Midday, Polluted period | Ma et al. (2019) |

**Table 4.** Previous field measurements of OH, HO$_2$ and RO$_2$ that have taken place during wintertime in urban areas, together with the campaign average observed to modelled ratio. Modified from Kanaya et al. (2007).





## 4 Discussion

### 4.1 Sources and sinks of ROₓ radicals

As shown in Figure 9, primary production of new radicals (radicals defined as $RO_x = OH + HO_2 + RO + RO_2$) via initiation reactions was dominated by the photolysis of HONO (83%, averaged over the campaign), with a small contribution from the photolysis of HCHO (1.1%), photolysis of carbonyl species (4.4%) and ozonolysis of alkenes (10%). An increased rate of production of ROₓ radicals is observed during haze events, which is counterbalanced by an increase in the rate of termination.

Figure 9 shows that alkene ozonolysis does not play an important role in production of ROₓ radicals at night and is reflected by little to no OH observed during night-time as shown in Figure 8 (a). Similarly ozone photolysis does not appear to play an important role for the formation of OH, due to the low $O_3$ during the campaign, presumably a consequence of local titration via NO, as shown in Figure 4 and Figure 5. In addition, the low temperatures observed during winter caused a low water vapour

concentration (~0.5 % mixing ratio), and hence the fraction of $O^1D$ formed from the photolysis of ozone and which reacts with water vapour to form OH compared with collisional quenching (by $N_2$ and $O_2$) to form $O(^3P)$ was also low, and varied between 1% to 7% throughout the campaign.

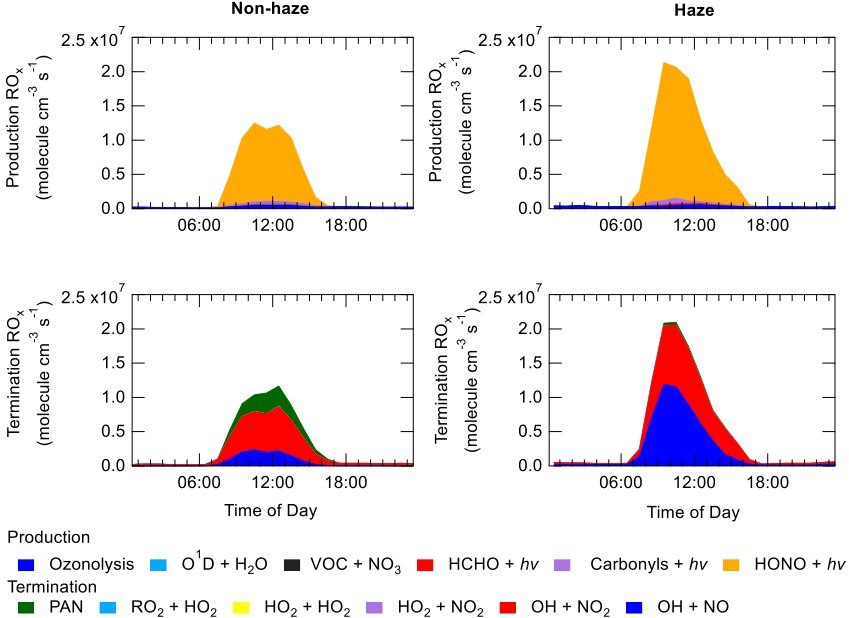

**Figure 9.** Rates of primary production (top panel) and termination (bottom panel) for ROₓ radicals
(defined as $OH + HO_2 + RO + RO_2$) separated into haze (right) and non-haze (left) periods. The definition of haze is when $PM_{2.5}$ exceeds 75 µm⁻³. The production from: $O^1D + H_2O$, $VOC + NO_3$, carbonyls + $hv$ and the termination reactions: $RO_2 + HO_2$, $HO_2 + HO_2$, $HO_2 + NO_2$, although shown in the key, are not visible and contributed <1% of the total prodcution and termination.



The importance of HONO photolysis as a source of OH has been highlighted in several previous studies
in both urban and suburban sites as summarised in Table 5.

The BEST-ONE campaign, 60 km north of Beijing, showed HONO produced ~ 46 % of the $RO_x$ during
the campaign, although in comparision to the APHH campaign, ozonolysis and carbonyl photolysis in
BEST-ONE made up a more significant portion of primary production of radicals, 28 % and 9 %,
respectively. The larger contribution to primary production from ozonolysis during BEST-ONE is
probably due to higher ozone concentrations (3 times higher at midday, Figure 9). Both the APHH and
BEST-ONE campaigns showed that ozone photolysis followed by the reaction of $O(^1D)$ atoms was not
an important source of new radicals. As summarised in Table 2, several other winter-time campaigns
have highlighted the importance of HONO, including the PUMA campaign (Emmerson et al., 2005) in
Birmingham; the IMPACT campaign in Tokyo (Kanaya et al., 2007); the NCITT campaign in Boulder (Kim
et al., 2014) and the PMTACS-NY campaign in New York (Ren et al., 2006). These campaigns showed
36.2, 19, 80.4, and 46 % contribution to primary production of ROx from HONO. However, it should
be noted that HONO was not measured during the PUMA camapign, so the percentage contribution
to the primary production of radicals should be considered a lower limit as it is based upon modelled
HONO (where only the reaction of OH + NO was considered), which is often an underestimate (Lee et
al., 2015). As shown in Table 5, the Birmingham, Tokyo, New York and Surburban Beijing campaigns
all show a high contribution towards $RO_x$ production from ozonolysis, 63, 35, 42 and 28%, respectively,
only the campaign in Boulder (5%) showed little contribution, which is similar to the observations made
during AIRPRO campaign. The Boulder campaign is the only one that showed a significant contribution
(14.9 %) to primary radical production from the reaction of $O(^1D) + H_2O$, whilst other winter campaigns
show a contribution of less than 1%. The higher contribution from photolysis of $O_3$ during the Boulder
campaign may be due to the campaign taking place in late February (closer to summer) and, as shown
in Table 5, photolysis rates, water vapour and temperature were all higher.

In both haze and non-haze conditions, the two key reactions which caused a termination of the radical
cycling chain reaction were OH + NO and OH + $NO_2$. Figure 9 shows that OH + NO contributes up to
53% and 25% of the rate of termination of radicals in haze and non-haze conditions, respectively, and
OH + $NO_2$ contributes up to 44% and 55% , respectively. Figure 9 shows that during non-haze
conditions contribution to termination from the net formation of PAN (~19%) becomes important; but
under haze conditions less than 2% of $RO_x$ termination comes from the net formation of PAN. In
comparision to the BEST-ONE campaign, during the clean periods (clean periods are defined as times
when $k$OH < 15 s$^{-1}$), the termination reactions of OH + $NO_x$, net-PAN and peroxy self-reaction
contributed ~ 55%, 8%, 30% respectively (Tan et al., 2018). During the polluted periods in the BEST-
ONE campaign, the termination reaction of OH + $NO_2$ increased to 80%, and the net-PAN formation

and peroxy self-reaction decreased to ~ 12% annd 6% respectively. The BEST-ONE campaign shows very similar trends to the APHH campaign, except the APHH campaign shows a higher contribution to

termination from OH + NO and OH + NO$_2$ even under cleaner periods. This is potentially due to the higher NO values observed during APHH (located in central Beijing ~6.50 km from Forbidden City) campaign compared to the BEST-ONE campaign. The work that took place at Peking University (PKU) (Ma et al., 2019) in Beijing (~11 km from the Forbidden City) shows a very smilar trend to the APHH campaign with 86% of the primary production of radicals produced from the photolysis of HONO

during the polluted periods . The PKU campaign also showed <1% production from O$^1$D + H$_2$O, whilst small contribtions from ozonlysis (6%) and photolysis of carbonyls (including HCHO, ~7%) during the polluted periods. Similar to the APHH campaign, the termination of radicals during the PKU campaign during the polluted periods was dominated by the OH + NO (55%) and OH + NO$_2$ (43%), whilst there was a small contribtuion (~2%) from the net-formation of PAN. The termination trend is very similar

to the APHH campaign.

|  | PUMA, Birmingham, UK | IMPACT, Tokyo, Japan | NCITT, Boulder, USA | PMTACS-NY, New York, USA | BEST-ONE, Suburban Beijing, China | APHH, Central Beijing, China | PKU, Central Beijing, China |
|---|---|---|---|---|---|---|---|
| Date | Jan – Feb 2000 | Jan – Feb, 2004 | Late Feb 2011 | Jan – Feb, 2001 | Jan – March 2016 | Nov -Dec, 2016 | Jan-Feb, 2017 |
| OH (cm$^{-3}$) | ~1.7 x 10$^6$ | ~1.6 x 10$^6$ | ~2.7 x 10$^6$ | ~ 1.4 x 10$^6$ | 3 x 10$^6$ | 3 x 10$^6$ | 1.4 x 10$^6$ |
| O$_3$ (ppbv) | 37 | 20 | 40 | 20 | 30 | 15 | 10 |
| $j$(O$^1$D) (s$^{-1}$) | ~1 x 10$^{-5}$ | ~2.8 x 10$^{-5}$ | ~1 x 10$^{-5}$ | ~5 x 10$^{-6}$ | 7 x 10$^{-6}$ | ~3 x 10$^{-6}$ | - |
| $j$(O$_3$) (%) | 0.6 | <1 | 14.7 | 1.1 | <1 | <1 | <1 |
| $j$(HONO) (%) | 36.2[1] | 19 | 80.4 | 65.5 | 46 | 83.3 | 86 |
| Ozonolysis (%) | 63.2 | 35 | 4.9 | 42.4 | 28 | 10.0 | 6 |
| $j$(Carbonyls) (%) | 22 | 23 | - | - | 9 | 4.5 | 7%[2] |
| $j$(HCHO) (%) | 6 | 10 | - | 6 | 9 | 1.1 | |
| Reference | Emmerson et al. (2005) | Kanaya et al. (2007) | Kim et al. (2014) | Ren et al. (2006) | Tan et al. (2018) | This work. | Ma et al. (2019) |

**Table 5.** Summary of some previous measurements of OH, HO$_2$ and RO$_2$ that have taken place during the winter, and a summary of the major primary radical sources during these campaigns. All values are the noon average for each campaign. [1] This should be considered a lower limit due to no HONO

measurements being made during the campaign. [2] Primary production from the sum of $j$(Carbonyls) and $j$(HCHO).



### 4.2 Dependence of radicals concentrations with NO$_x$

Figure 10 shows the ratio of measured-to-modelled OH, which is close to 1 at or below 10 ppbv of NO; similar to the BEST-ONE campaign. Above 6 ppbv of NO the model underpredicts the OH

concentration. As shown in Figure 10, at ~6 ppbv of NO; HO$_2$ and RO$_2$ are underpredicted by a factor of 5.4 and 18, respectively; similar peroxy radical under-predictions were reported from the BEST-ONE campaign (Tan et al., 2017; Tan et al., 2018), with HO$_2$ and RO$_2$ being underpredicted by a factor of 5 and 10 at 6 ppbv. Many previous urban campaigns have a more extensive data coverage at lower NO$_x$ values due to the smaller levels of NO$_x$ observed; however, no other campaign with *in situ*

measurements of OH has experienced NO values up to 250 ppbv as observed during APHH. Figure 10 shows that the measured-to-modelled ratio for OH, HO$_2$ and RO$_2$ increases with NO concentration; for OH the ratio initially increases and then plateaus above 30 ppbv. There have been some suggestions for the origin of the discrepancy that is observed between modelled and measured concentrations of radicals at high concentrations of NO. Dusanter et al., (2009) suggest that poor mixing of a point source

of NO with peroxy radicals across a site may cause some of the model to measurement discrepancy observed. Tan et al., (2017) suggest that there may be a missing source of peroxy radicals under high-NO$_x$ conditions. Alternatively, the measured-to-modelled discrepancy could be driven by unknown oxidation pathways of the larger, more complex, RO$_2$ species that are present in these urban environments, whose laboratory kinetics are under-studied.

When the MCM is constrained to the measurements of HO$_2$ (MCM-cHO$_2$), the model can replicate the OH measurements to ~10%, within the 26% error of the measurements, as shown in Figure 8. In addition, the MCM-base model can replicate the OH reactivity within 10 % (Figure 8), implying that almost all of the major sources and sinks of OH are captured. The underestimation of HO$_2$ by the model could be explained by the underestimation of RO$_2$ by the model, owing to an insufficient rate of

recycling of RO$_2$ to HO$_2$. Both the ability to replicate OH when the model is constrained to HO$_2$, along with OH reactivity being captured well by the model, suggests the presence of unknown RO$_2$ chemistry; either additional sources of RO$_2$ radicals under high levels of NO$_x$ or unknown chemistry/behaviour of RO$_2$ under high levels of NO$_x$. Indeed, many rate coefficients in the MCM for the more complex RO$_2$ species are based on structure activity relationships (SARs) determined from

studies of simpler RO$_2$ species (http://mcm.leeds.ac.uk/MCM/home, Jenkin et al. (2019)). During the APHH campaign, measurements of partially speciated RO$_2$ species were made: RO$_2$ simple (deriving from alkanes up to C$_3$) and RO$_2$ complex (deriving from alkanes > C$_4$, alkene and aromatics), see experimental section 2.2.1 for details on RO$_2$ speciation and (Whalley et al., 2013). The dependence of the concentration of speciated RO$_2$ measurements against [NO], as shown in Figure 10, highlights

that the concentration of complex RO$_2$ species has a steady decrease across the NO range, whilst the



concentration of simple RO₂ species starts to decrease rapidly above 2.5 ppbv, and can almost be reproduced by the model at NO concentrations above 100 ppbv. The chemistry of the simpler RO₂ species with NO should be well understood, owing to a more extensive laboratory database of the rate coefficients and product branching, so the model discrepancy for RO₂ species may be due to

inaccuracies within the MCM for the degredation of the more complex RO₂ species into these simple RO₂. The degradation pathways of the complex RO₂ species appear not to be well understood, and may be the reason why the real concentration of simple RO₂ species remain high even under high NOₓ conditions, whereas the modelled simple RO₂ concentration decreases at high NO.

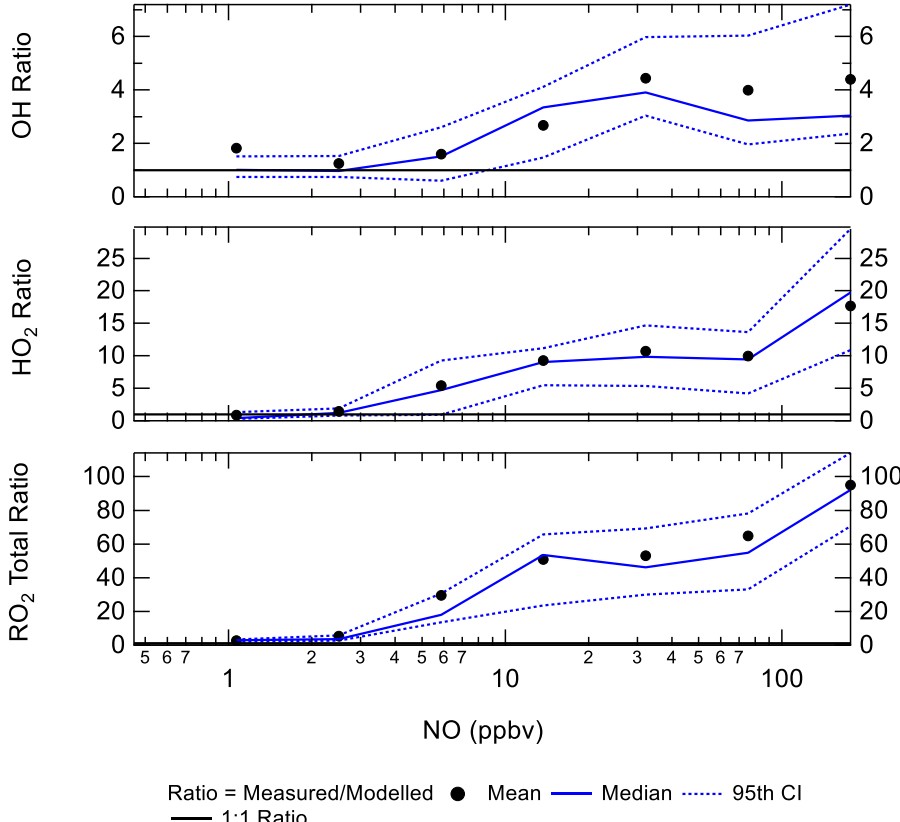

**Figure 10.** The ratio of measurement/model for OH (top), HO₂ (middle) and total RO₂ (bottom) across the range of NO concentrations experienced, for daytime values only (j(O¹D) > 1 x 10⁻⁶ s⁻¹). CI = Confidence Interval.

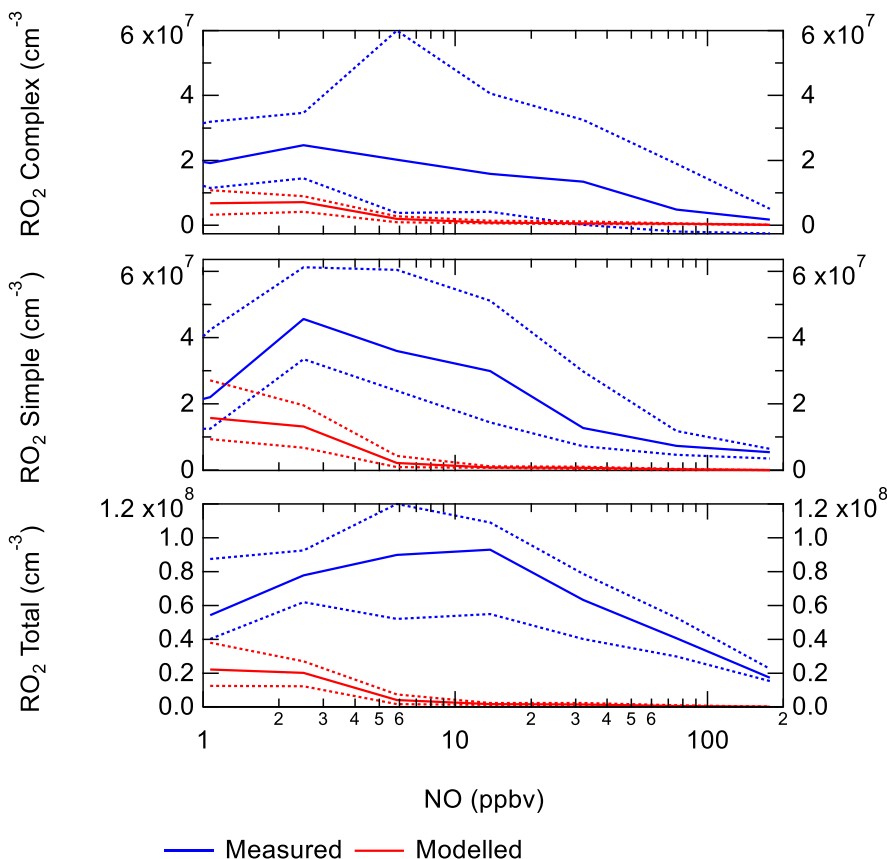

Measured ——— Modelled

**Figure 11.** a – Complex $RO_2$ measurements (blue) and complex $RO_2$ modelled (black) versus NO. b – Simple $RO_2$ measurements (blue) and simple $RO_2$ modelled (black) versus NO. c – Total $RO_2$ measurements (blue) and total $RO_2$ modelled (black) versus NO. The points used are for daytime values only ($j(O^1D) > 1 \times 10^{-6}$ s$^{-1}$). See text for definition of "simple" and "complex" $RO_2$.

The additional primary production of $RO_x$ (P'$RO_x$) radicals required to bridge the gap between measured and modelled total $RO_2$ was found to be on average $1.2 \times 10^8$ molecule cm$^{-3}$ s$^{-1}$ at noon (17 pbbv h$^{-1}$) as shown in Figure 12, calculated from Eq. 3 (Tan et al., 2018):

$$P'(RO_x) = k_{HO2+NO} [HO_2] [NO] + P(HO_2)_{prim} + P(RO_2)_{prim} + k_{VOC}[OH] \qquad \text{Eq. 3}$$
$$- L(HO_2)_{term} - L(RO_2)_{term}$$

where P(HO$_2$)$_{prim}$, P(RO$_2$)$_{prim}$, L(HO$_2$)$_{term}$ and L(RO$_2$)$_{term}$ are the rates of primary production of HO$_2$, primary production of RO$_2$, termination of HO$_2$ and termination of RO$_2$, respectively. The additional primary source of RO$_2$, P'(RO$_2$)$_{prim}$, is almost four times larger than the additional RO$_2$ source that was required to resolve the measured and modelled RO$_2$ during the BEST-ONE campaign (5 ppbv h$^{-1}$ during polluted periods, also calculated using Eq. 3), and is much larger compared to the noon-average



modelled primary production of $RO_x$ during the APHH campaign of 1.6 ppbv $h^{-1}$. It has been suggested previously in Tan et al. (2017) that the missing primary radical source originates from the photolysis of $ClNO_2$ and $Cl_2$ to generate Cl atoms, which can further oxidise VOCs to generate peroxy radicals. However, as no measurements of $ClNO_2$ or $Cl_2$ measurements were made during the campaign, this route cannot be quantified. However, Cl atom chemistry may only play a minor role, as the inclusion of $ClNO_2$ in a model during a summer campaign in Wangdu (60 km from Bejing) could only close 10 – 30% of the gap between the model and measurements (Tan et al., 2017).

Eq.3 has been used to calculate an additional primary source $(P'(RO_x))$ required to reconcile measured and modelled $RO_2$; on average this peaked at 1.05 x $10^8$ molecule $cm^{-3}$ $s^{-1}$. The calculated additional $RO_2$ $(P'(RO_x))$ source was included in the model (model run is called MCM-PRO2) as a single species 'A-I' that formed several $RO_2$ species at the required $RO_2$ production rate (i.e. $k*$[A-I] = missing primary production rate, $P'(RO_2)_{prim}$). Using the MCM nomenclature ([http://mcm.leeds.ac.uk/MCM/home](http://mcm.leeds.ac.uk/MCM/home)), the $RO_2$ species produced were HOCH2CH2O2, HYPROPO2, IBUTOLBO2, BUTDBO2, OXYBIPERO2, CH3O2 and BUT2OLO2, NBUTOLAO2, and the structures of these $RO_2$ species are shown in Table 6. The $RO_2$ species were chosen after a rate of production analysis (ROPA) analysis showed they were highest produced $RO_2$ species in the model.

The comparison between sum of $RO_2$ observed and sum of $RO_2$ modelled from the model run MCM-P'$RO_2$ demonstrates good agreement in general (Figure 13), although there is a slight overprediction of $RO_2$ in the afternoon and a slight underprediction of $RO_2$ in the morning. However, the MCM-PRO2 run overpredicts the observed OH and $HO_2$ by a factor of 1.6 and 2.4, respectively, with the large overprediction of $HO_2$ driving the overprediction of OH. To investigate whether the uptake of $HO_2$ onto the surface of aerosols could improve the agreement between measured and modelled $HO_2$, the MCM-PRO2 modelled was modified to include the uptake of $HO_2$ with the uptake coefficient set equal to 0.2, as suggested by Jacob (2000), in model run MCM-PRO2-SA. The measured average aerosol surface area peaked at an average of 6.38 x $10^{-6}$ $cm^2$ $cm^{-3}$. The comparison of MCM-PRO2-SA with both measurements and MCM-PRO2 (see Table 2 for details) is shown in Figure 13 and shows that the uptake of $HO_2$ only has a small impact (< 8%) on the modelled levels of OH, $HO_2$ and $RO_2$. The aerosol surface area used in the model may be a lower limit as it was calculated from an Scanning Mobility Particle Sizer (SMPS) that only measured aerosols ranging from 10 nm - 1000 nm. At the high levels of NO encountered, the lifetime of $HO_2$ is short, and the decrease in $HO_2$ in MCM-PRO2-SA owing to loss onto aerosols is not enough to reconcile measurements with the model and suggests that an additional primary source of $RO_2$ may not be the cause of the model underprediction of $RO_2$ species, as the inclusion of additional $RO_2$ production worsens the model's ability to predict OH and $HO_2$. If





there is missing RO₂ production, the rate of propagation of these species to HO₂ would need to be

        slower than currently assumed in the model to reconcile the observations of OH, HO₂ and RO₂.

        The small decrease in modelled HO₂ by heterogeneous uptake contrasts with the recent work from Li

        et al. (2019) that has shown, using GEOS-Chem, that the observed increasing ozone trend in North

        China Plain is caused by reduced uptake of HO₂ onto aerosol due to reduction in PM₂.₅ by ~40%

between 2013 – 2017.

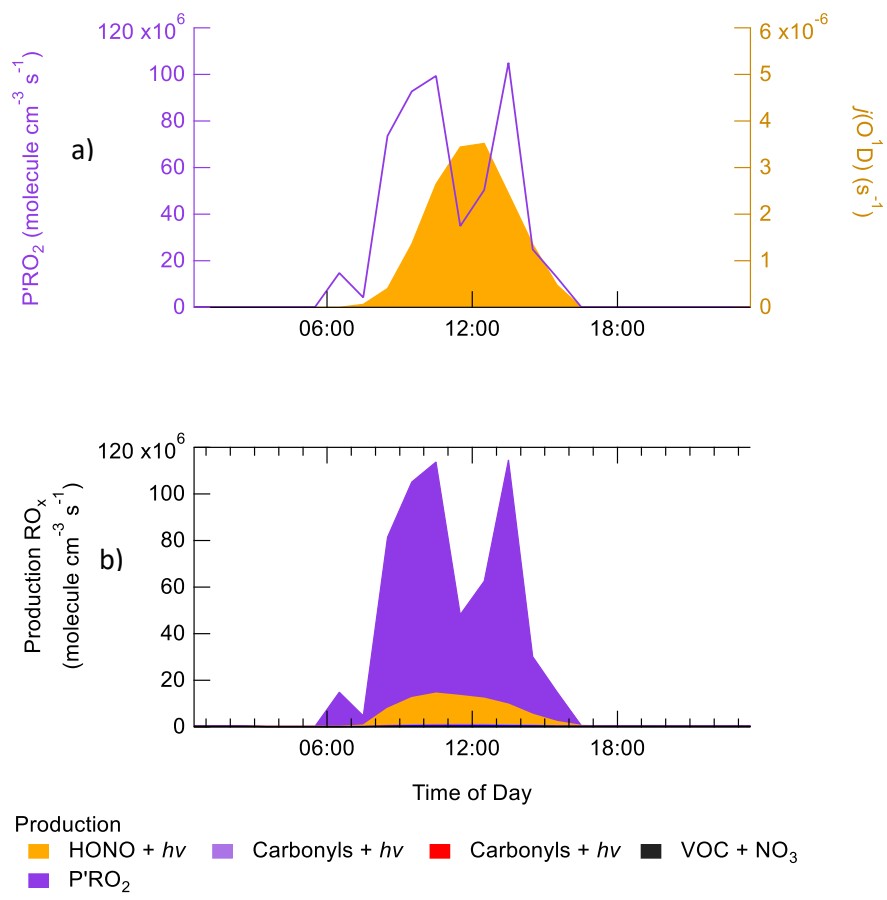

**Figure 12.** a- Average diel profile of the additional rate of primary production (P'RO$_x$) (blue) required
to reconcile the model with the measurements of total RO₂), and $j$(O$^1$D) average diel profile (yellow).
b- Breakdown of the rate of primary production of RO$_x$ showing the contribution made by this
additional rate of RO₂ production, P'(RO2) (shaded purple).


| MCM Name | Structure | MCM Name | Structure |
|---|---|---|---|
| HOCH2CH2O2 | | BUTDBO2 | |
| HYPROPO2 | | OXYBIPERO2 | |
| IBUTOLBO2 | | CH3O2 | |
| BUT2OLO2 | | NBUTOLAO2 | |

**Table 6.** The names and associated structures of the RO$_2$ species used to add additional primary production of RO$_2$ species into MCM-PRO2 and MCM-PRO2-SA. See http://mcm.leeds.ac.uk/MCMv3.3.1/home.htt for more details.

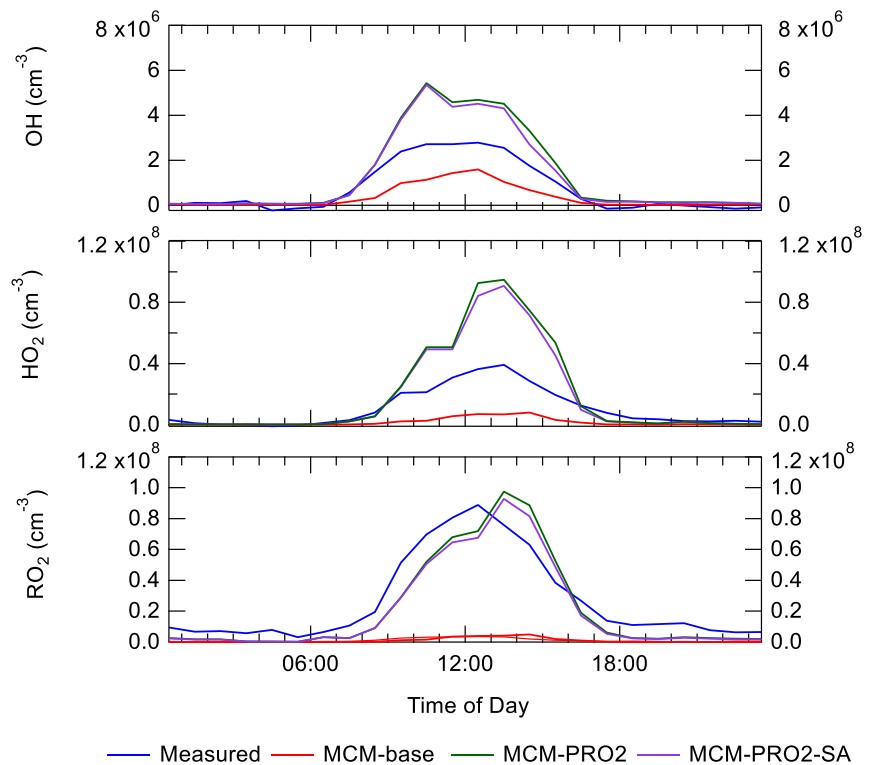

─── Measured ─── MCM-base ─── MCM-PRO2 ─── MCM-PRO2-SA

**Figure 13.** Average diel comparison of measurements of OH, HO₂ and sum of RO₂ with the MCM-base,
MCM-PRO2 and MCM-PRO2-SA box-model runs. The average diel is from the entire APHH winter
campaign. See text and Table 2 for definitions of each of the model runs.

**4.3 Chemistry of radicals under haze conditions and the rate of oxidation of NO₂ and SO₂ to form
nitrate and sulphate aerosol**

The observed concentrations of OH during the APHH campaign are much higher than those predicted

by global models (~0.4 x 10⁵ cm⁻³, for a 24 hr period average during summertime) in the north China

plane (NCP) (Lelieveld et al., 2016), and the OH concentration in and outside of haze events are

comparable, despite the lower light levels during these events (on average up to 50% less j(O¹D) during

the haze events) as shown in Figure 5. The levels of OH are partly sustained during haze events owing

to a significant increase in [HONO] in haze (see Figure 5), with HONO being a major source of OH,

despite the reduction in *j*(HONO) in haze. The average midday OH reactivity measurements in and out

of haze were 47 (s⁻¹) and 17 (s⁻¹), respectively, and since the OH concentrations are comparable in and

out of haze, this implies there is a larger turnover rate (defined as the product of [OH] and *k*(OH)), or

rate of chemical oxidation initiated by OH radicals, within haze, to balance this. The radical chain


length, ChL, is defined by the rate of radical propagation divided by the rate of radical production, and is given by Eq. 4:

$$\mathrm{ChL} = [\mathrm{OH}] \times k_{\mathrm{voc}}/P(\mathrm{RO_x}) \qquad \qquad \text{Eq. 4}$$

where $k_{\mathrm{VOC}}$ is the total OH reactivity with VOCs and $P(\mathrm{RO_x})$ is the primary production of $\mathrm{RO_x}$ radicals. As shown in Table 7 the average of ChL calculated using Eq. 4 during the APHH campaign was ~5.9. This large value indicates that radical propagation during the APHH campaign is very efficient; this

ChL is higher than calculated for previous winter campaigns that had OH radical and OH reactivity measurements available, together with VOCs. The large chain length comes from the product of large OH concentrations and high OH reactivity measurements.

| Campaign | OH $10^6$ cm$^{-3}$ | P(RO$_x$) (ppbv h$^{-1}$) | kOH (s$^{-1}$) | NO$_2$ (ppbv) | Chainlength ChL | Reference |
|---|---|---|---|---|---|---|
| PUMA, Birmingham | 1.7 | 2.8 | 30 | 9.3 | 2.1 | Emmerson et al. (2005) [a] |
| NY NYC, US | 1.4 | 1.4 | 27 | 15 | 3.3 | Ren et al. (2006) |
| IMPACT Tokyo | 1.5 | 1.4 | 23 | 12 | 3.1 | Kanaya et al. (2007) [a] |
| Boulder | 2.7 | 0.7 | 5 | 5 | 2.0 | Kim et al. (2014) |
| BEST-ONE, Suburban Beijing | 2.8 | 0.9 | 12 | 6 | 4.7 | Tan et al. (2017) |
| APHH, Central Beijing | 2.7 | 1.6 | 47 | 30 | 5.9 | This work. |

**Table 7.** Comparison of OH concentration, primary production of RO$_x$ radicals (P(RO$_x$)), OH reactivity (kOH), NO$_2$ concentration and chain length defined by Eqn (4) for various campaigns. The values are a noon-time average. Table modified from Tan et al. (2018). [a] OH reactivity is calculated only.

The average diel profiles of radical concentrations, both measured and calculated by the model, inside and outside of haze periods are presented in Figure 14: the maximum average OH concentration observed is almost the same in and out of haze (~2.7 x 10$^6$ molecule cm$^{-3}$) whilst the concentrations of the observed peroxy radicals decrease in haze. The model can replicate OH (within 20%) outside of haze but significantly underpredicts OH inside of haze events. The model also underpredicts HO$_2$ and

RO$_2$ during haze, but over-predicts HO$_2$ under the non-haze conditions. The OH reactivity is replicated well by the model both in haze and non-haze conditions. Figure 14 shows the OH concentration





observed both in and outside of haze events is significant and indictates that gas-phase oxidation is taking place, and hence the formation of secondary oxidation products, even within haze conditions. Secondary oxidation products, such as nitric acid and sulphuric acid, which partition to the aerosol

phase, are major contributors towards the formation of secondary particulate matter (Huang et al., 2014). The OH measurements enable calculation of the rate of $SO_2$ and $NO_2$ oxidation *via* reaction with OH, to form gas-phase phase $HNO_3$ and $H_2SO_4$.. Figure 16 shows that on average 1.5 ppbv/h and 0.03 ppbv/h of gas-phase $NO_2$ and $SO_2$ are oxidised to form acidic species, and that the oxidation increases in these haze periods caused by comparable OH concentration in and out of haze and, as shown in

Figure 5, an increase in local $NO_2$ and $SO_2$ concentrations. $NO_x$ can also be lost in the atmosphere by the formation of $N_2O_5$ (Evans, 2005) and subsequent hydrolysis, but this is uninportant in Beijing during winter due to the low levels of $O_3$. The reaction of OH + $SO_2$ in the gas-phase is the rate-determining step in the formation $SO_4^{-2}$, so the $H_2SO_4$ formed in the gas-phase will partition in the aerosol phase (Barth et al., 2000). $H_2SO_4$ is effectively a non-volatile gas at atmospheric temperatures,

and $H_2SO_4$ condensation onto pre-exsisiting particles is an irreversible kinetic process (Zaveri et al., 2008). Whilst $HNO_3$ is a semivolatile species and the gas-particle partitioning is highly sensitive to to meteorological conditions including: temperature, RH, particle size distribution, pH and particle composition. If the realtive humidity is lower than the deliquescence relative humidity ($RH_d$), then the $HNO_3$ that is formed in the gas phase reacts with $NH_3$ to form ammonium nitrate aersol ($NH_4NO_3$):

$$HNO_3(g) + NH_3(g) \rightleftharpoons NH_4NO_3(s) \qquad\qquad\qquad R1$$

If the ambient RH exceeds the $RH_d$ then $HNO_3$ and $NH_3$ dissolve into the aqueous phase (aq):

$$HNO_3(g) + NH_3(g) \rightleftharpoons NO_3^-(aq) + NH_4^+(aq) \qquad\qquad R2$$

To take into account the reversible process, knowledge of the $RH_d$ that marks the transition between the solid and the aqueous phase, and the equilibrium constant, $K_p$, for the two phase is required (Ackermann et al., 1998). The MADE module (modal aerosol dynamics model for europe) uses these thermodynamic parameters as given by (Mozurkewich, 1993), resulting in:

$$\ln\left(\frac{RH_d}{100}\right) = \frac{618.3}{T} - 2.551 \qquad\qquad\qquad \text{Eq. 5}$$

for $RH_d$ and:

$$\ln\left(K_p\right) = 118.87 - \frac{24084}{T} - 6.025\ln(T) \qquad\qquad \text{Eq. 6}$$

for $K_p$. Eq. *5* and Eq. *6* shows that nitrate formation is favoured thermodynamically at low temperatures and high relative humidties (Ge et al., 2017). Previous measurements of $SO_4^{-2}$ and $NO_3^-$




made in wintertime Beijing suggests that photochemstry is important in the formation of nitrate
aersol, but not the formation of sulphate (Ge et al., 2017; Sun et al., 2013).

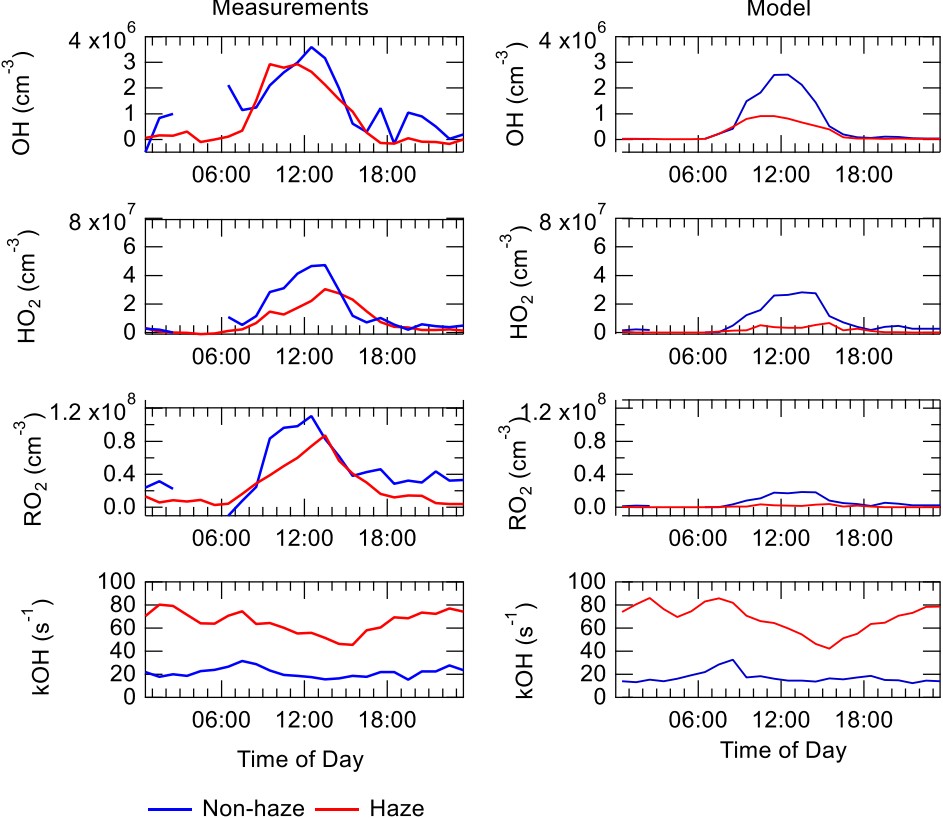


**Figure 14.** Average diel profiles for OH, HO₂, RO₂ and kOH for measurements (left) and model (right)
separated into haze (red) and non-haze (blue) periods.

The average diel profiles for the measurements of NO₃⁻ and SO₄⁻ made during the APHH separated into
haze and non-haze periods are shown in Figure 15. The average diurnal of NO₃⁻ shows a peak at

midday, suggesting photochemistry is important in its formation, whilst the SO₄⁻ diurnal shows an anti-
correlation with photolysis rates. As shown in Figure 15, the SO₄⁻ tracks the RH very well suggesting
that the dominant path for sulphate formation during winter-time in Beijing is through the aqueous
processing of SO₂. The shape of the average diurnal of NO₃⁻ and SO₄⁻ is consistent with studies made
by Sun et al. (2013) and Ge et al. (2017). Figure 16 also shows that the gas-phase oxidation of NO₂

increases under haze conditions, showing that nitrate formation is driven by photochemistry in haze
events despite the lower photolysis rates. Similar conclusions have been made in Lu et al. (2019) from
measurements during the BEST-ONE campaign; with SO₄⁻ aerosol predominantly driven by aqueous-





phase chemistry whilst the production of $NO_3^-$ aerosol from gas-phase oxidation of $NO_2$ with OH is important. The maximum production rate of $HNO_3$ observed during the BEST-ONE campaign is the
same as the one calculated for the APHH campaign (3 ppbv $hr^{-1}$). The BEST-ONE campaign assumed all the gas-phase $HNO_3$ formed partitioned into the aerosol-phase due to the high relative humidity observed during the campaign.

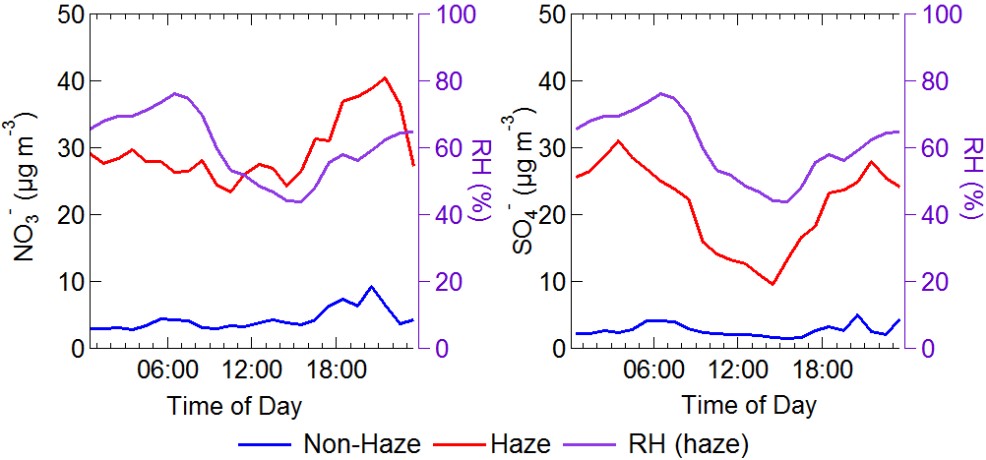

**Figure 15.** Average diel profiles of $NO_3^-$ and $SO_4^-$ made during the APHH winter campaign separated
into haze and non-haze conditions. And the relative humidity (RH) measured during haze periods. Haze = $PM_{2.5}$ > 75 µg $m^{-3}$.

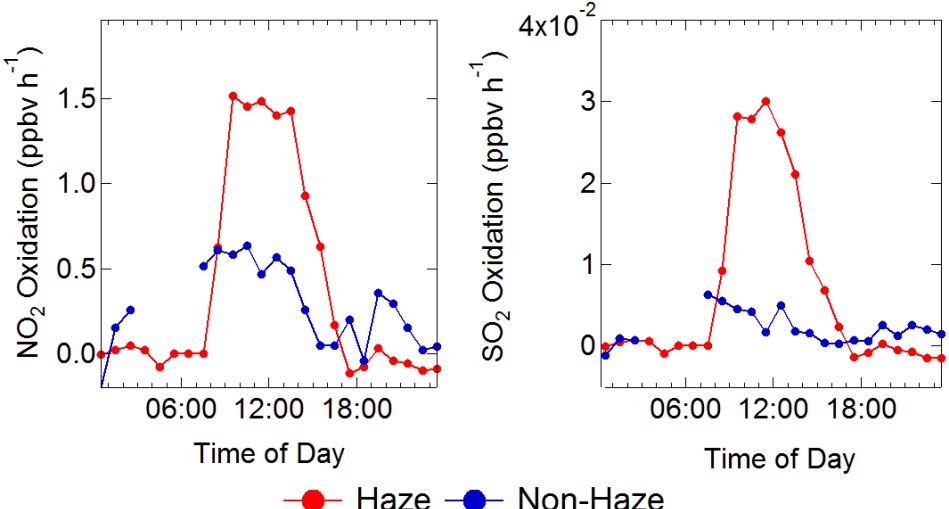

**Figure 16.** Average diel profiles of the rate of oxidation of $NO_2$ (left) and $SO_2$ (right) via reaction with OH in non-haze (blue) and haze (red) conditions.



### 4.4 Implications of model under-prediction of RO₂ radicals on the calculated rate of ozone production

Although ozone pollution is generally not considered a wintertime phenomenon in Beijing, the elevated levels of $RO_2$ observed under high $NO_x$ conditions suggests that ozone could be produced rapidly, but then is rapidly titrated to $NO_2$ by reaction with NO. As well as being an important greenhouse gas, $O_3$ has a negative impact on both human health and crop yields (Lin et al., 2018), and in China led to 74,200 premature deaths and a cost to the economy of 7.6 billion US$ in 2016 (Maji et al., 2019).

The $RO_2$ radicals are under-predicted in the model, especially under the higher $NO_x$ conditions, and as shown in Figure 17, this has an implication for the model's ability to predict the rate of *in situ* $O_3$ production. The rate of $O_3$ production is assumed to be equal to the net rate of $NO_2$ production Eq. 7:

$$P(O_3) = k_{HO_2+NO}[HO_2][NO] + k_{RO_2+NO}[RO_2][NO] - k_{OH+NO_2+M}[OH][NO_2][M] \qquad \text{Eq. 7}$$
$$- k_{HO_2+O_3}[HO_2][O_3] - P(RONO_2)$$

where $RO_2$ represents the sum of $RO_2$, and the last three terms allow for the reduction of ozone production owing to reactions that remove $NO_2$ or its precursors. The $P(RONO_2)$ term is the net rate of formation of organic nitrate, $RONO_2$, species, for example peroxy acetyl nitrates (PANs).

When the rate of $O_3$ production is calculated using the measured values of $HO_2$ and $RO_2$, there is a positive trend with increasing NO. However, when the modelled concentrations of $HO_2$ and $RO_2$ are used, there is a constant $P(O_3)$ across the whole NO range, leading to a large underestimation of $O_3$ production by the model at higher values of NO. At ~2.5 ppbv and ~177 ppbv of NO the model underestimates the $O_3$ production by 1.8 and 66, respectively. Figure 17 also shows that there is a high rate of *in situ* ozone production in Beijing in winter and, as shown in Table 8, the maximum rate of ozone production calculated from observed $HO_2$ and $RO_2$ is higher for Beijing winter than the corresponding values during the summer-time ClearfLo campaign in London. However, because of the very high NO in Beijing campaign, immediate titration of the $O_3$ formed results in very low ambient amounts, see Figure 5. As shown in Table 8, the average of the rate of ozone production calculated from observations of $HO_2$ and $RO_2$ between 08:00 and 17:00 during our APHH campaign (71 ppbv hr⁻¹, at 40 ppbv of NO) was higher than those calculated using observations during the BEST-ONE campaign (10 ppbv hr⁻¹, at 8 ppbv of NO) and calculated from the measured $HO_2$ and modelled $RO_2$ in the PKU campaign (43 ppbv hr⁻¹, at 39 ppbv of NO). An isopleth for ozone showing production as a function of $NO_x$ and VOC for the BEST-ONE campaign (Lu et al., 2019) showed that a reduction in $NO_x$ alone would lead to an increase in $O_3$ production, and an increase in the amount of secondary organic aerosol produced.



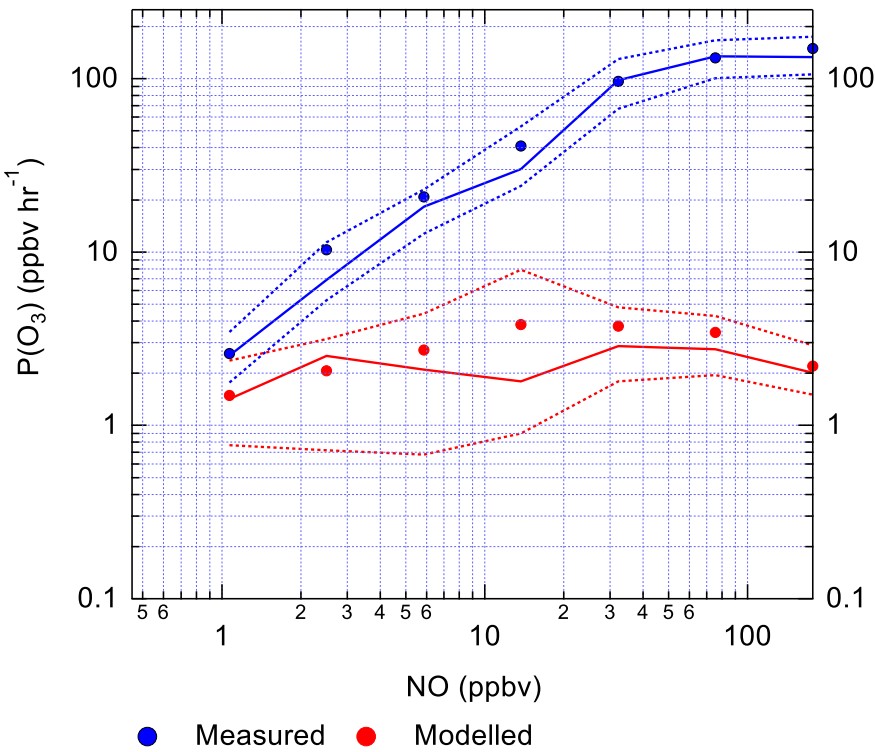

**Figure 17.** The calculated rate of *in situ* ozone production as a function of [NO] for Eq. 7 using modelled (red) and measured (blue) values of $HO_2$ and the sum of $RO_2$ radicals.

The top ten $RO_2$ species that react with NO to form $NO_2$ are shown in Figure 18, the top ten $RO_2$ only

contribute to 65.8% of the ozone formed whilst the other 34.2% is from different $RO_2$'s that individually contribute less than 1.5% each. It shows that simple $RO_2$ species ($CH_3O_2$ and $C_2H_5O_2$) contribute 26.8% of the total ozone production from $RO_2$ species.





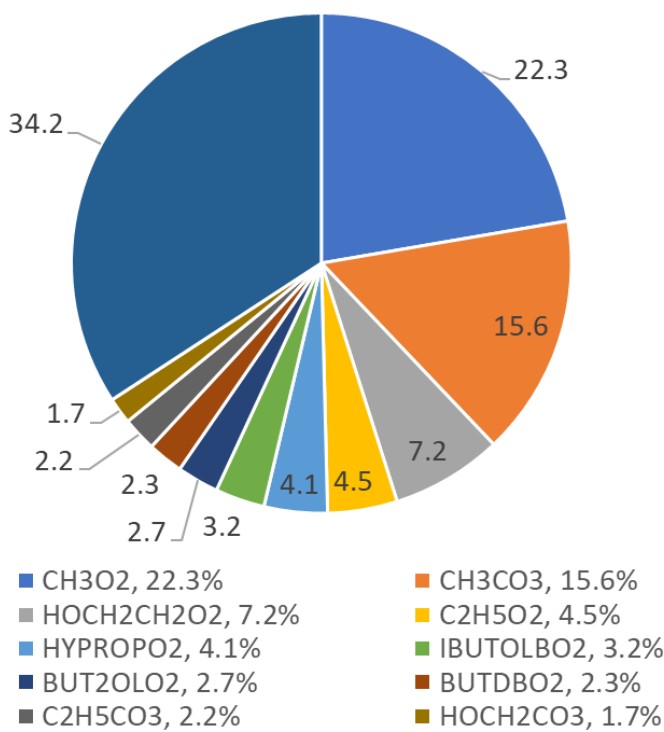

**Figure 18.** Pie chart showing the top ten $RO_2$ species that form ozone in the MCM-base model. These top ten $RO_2$ only contribute to a total of 65.8% of the ozone production, the rest coming from other $RO_2$ species (34.2%), each with less than a 1.5% contribution to the total production. The names for the $RO_2$ species are from the MCM, the related structures can be found http://mcm.leeds.ac.uk/MCM/.





| Campaign | Dates | NO | P(O₃) (ppbv hr⁻¹) | Notes | Reference |
|---|---|---|---|---|---|
| APHH | Nov – Dec 2016 | 40 | 71 | Rate average for the daytime periods between 08:00 and 17:00 | This work. |
| | | 177 | 132 | Maximum ozone production. | |
| BEST-ONE | Jan – Feb, 2016 | 8.0 | 10 | Rate average for the daytime periods between 08:00 and 17:00 | Tan et al. (2018) |
| PKU | Nov – Dec 2017 | 43 | 39 | Rate average for the daytime periods between 08:00 and 17:00 | Ma et al. (2019) |
| ClearfLo | July – Aug 2012 | 52 | 41 | Maximum ozone production. | Whalley et al. (2018) |

**Table 8.** The rate of *in situ* ozone production averaged between 08:00 – 17:00 for the APHH, BEST-ONE and PKU campaigns and the associated NO concentration. Also shown is the maximum rate of ozone production calculated from measured $HO_2$ and $RO_2$ during the APHH and ClearFLo campaigns.

**5. Summary**

The APHH AIRPRO campaign took place in central Beijing at the Institute for Atmospheric Physics (IAP)
in November and December 2016, with detailed measurements of OH, $HO_2$, sum of $RO_2$ and OH reactivity made using the FAGE technique. High radical concentrations were measured both inside and outside of haze events, despite the lower intensity of solar radiation and therefore photolysis rates in haze. The daily maxima for the radical species varied day-to-day from 1 to 8 x $10^6$ cm⁻³, 0.7 to 1.5 x $10^8$ cm⁻³ and 1 to 2.5 x $10^8$ cm⁻³ for OH, $HO_2$ and $RO_2$ respectively. Partial speciation of $RO_2$ was achieved,
with the sum of simple $RO_2$ deriving from <C₄ saturated VOCs reaching a daily maximum concentration between 0.2-1.3 x $10^8$ cm⁻³, and the complex $RO_2$ deriving from larger alkyl, unsaturated and aromatic VOCs reaching a daily maximum concentration between 0.2 and 0.6 x $10^8$ cm⁻³. The partially speciated $RO_2$ measurements showed on average almost 50:50 ratio between the two. The complex $RO_2$ species have higher mixing ratios under high NO (>40 ppbv) conditions whilst simple $RO_2$ have higher mixing
ratio at lower NO (<40 ppbv). The average daytime maximum of the radical species was 2.7.0 x $10^6$ cm⁻³, 0.39 x $10^8$ cm⁻³ and 0.88 x $10^8$ cm⁻³ for OH, $HO_2$ and total $RO_2$, respectively. The OH radical concentrations are higher than previous winter campaigns outside of China, and comparable to the BEST-ONE campaign that took place in suburban Beijing (60 km northeast of Beijing). The OH reactivity was very high, and showed a significant day to day variability from 10 s⁻¹ up to 150 s⁻¹ in the most
polluted periods. The major contribution to reactivity came from CO (17.3%), NO (24.9%), $NO_2$ (22.1%),



alkanes (3.0%), alkynes and alkenes (10.8%), carbonyls (5.7%), terpenes (3.7%) and model intermediates (6.77%). A steady state calculation for OH showed that the OH budget can be closed using measured $HO_2$, HONO and $k$(OH).

The primary production of new radicals by initiation reactions, as opposed to formation via propagation reactions, was dominated (>83%) by the photolysis of HONO, consistent with other winter campaigns. The rate of primary radical production from HONO was observed to increase during haze events, due to the large increase in HONO concentration, even though photolysis rates were considerably lower in haze. Radical termination was dominated by the reaction of OH with NO and $NO_2$, although under non-haze conditions, when $PM_{2.5} < 75$ μg m$^{-3}$, the contribution from net-PAN

formation became important (~19%).

The comparison of the measurements with a box-model utilising the detailed Master Chemical Mechanism generally showed an underestimation of OH, $HO_2$ and $RO_2$. The MCM was able to replicate OH and $HO_2$ concentrations quite well when [NO] was around 3 ppbv. The model underestimation occurred at [NO] > 2.5 ppbv for OH, $HO_2$ and $RO_2$. The underprediction of the radicals reached a

measured:modelled ratio of 3, 20 and 91 at 177 ppbv of NO. The under prediction of the peroxy radicals ($HO_2$ and $RO_2$) by the model leads to an underestimation of in situ $O_3$ production under high $NO_x$ conditions. When the MCM is constrained to the measured $HO_2$, the model can replicate measured OH, and the measured OH reactivity is captured well by the model. This suggests that under high $NO_x$ and haze conditions there is either an additional source of the peroxy radicals or unknown

recycling chemistry of $RO_2$ to $HO_2$. The OH concentrations inside and outside of haze events were very similar, on average $2.7 \times 10^6$ molecule cm$^{-3}$, which suggests that rapid gas-phase oxidation, generating secondary species such as secondary nitrate, sulphate and organic aerosol still occurs in haze events.

*Data availability*. Data presented in this study are available from the authors upon request
(l.k.whalley@leeds.ac.uk and d.e.heard@leeds.ac.uk).

*Author contributions.* ES, LW, RWM, CY and DH carried out the measurements; ES and LW developed the model and performed the calculations; JL, S, JH, RD, MS, JH, AL, LC, LK, WB, TV, YS, WX, PF, SY, LR, WA, CH and XW provided logistical support and supporting data to constrain the model; ES, LW and
DH prepared the manuscript; with contributions from all co-authors.

*Competing interests*. The authors declare that they have no conflict of interest.



**Acknowledgements** – We are grateful to the Natural Environment Research Council for funding via
the Newton Fund Atmospheric Pollution and Human Health in Chinese Megacity Directed
International Program (grant number NE/N006895/1) and the National Natural Science Foundation of
China (Grant No.41571130031). Eloise Slater and Freya Squires acknowledge NERC SPHERES PhD
studentships. We acknowledge the support from Zifa Wang and Jie Li from the Institute of Applied
Physics (IAP), Chinese Academy of Sciences for hosting the APHH-Beijing campaign. We thank
Liangfang Wei, Hong Ren, Qiaorong Xie, Wanyu Zhao, Linjie Li, Ping Li, Shengjie Hou and Qingqing
Wang from IAP, Kebin He and Xiaoting Cheng from Tsinghua University, and James Allan from the
University of Manchester for providing logistic and scientific support for the field campaigns. We
would also like to thank other participants in the APHH field campaign.

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
