# Peer review of "Elevated levels of OH observed in haze events during wintertime in central Beijing"

_Atmospheric Chemistry and Physics, 2020_

## Referee Comment (RC1) · Anonymous Referee #1 · 8 Jun 2020

Modeling and analysis of wintertime radical measurements (OH, HO2, RO2, and OH reactivity) in the highly polluted environment of Beijing offer unique conditions for testing our understanding of radical chemistry and its response to extreme values for precursor concentrations and radical recycling rates. The authors have undertaken a detailed analysis incorporating observational and model-based evidence that builds on a growing body of evidence that radical chemistry is not well understood under such conditions. In its current form, the paper could be shortened quite a bit and contains a bit of sloppiness that feels like a thesis is being repurposed into a manuscript. This work certainly needs to be published and the detailed comments offered below are intended to help improve the communication of these important results. One important omission is the lack of a short discussion on what should come next as this problem has been

observed across several studies and calls for a new approach if it is to be solved.

Detailed comments:

Figure 1: I fully appreciate the challenge in creating efficient and accurate diagrams such as Figure 1 to portray the chemistry of radical cycling. The species in green are described as primary routes for radical formation. I agree with the way that ozone is portrayed given that its role as an initiator as well as a product of the chemistry is shown. Both HONO and HCHO need to be better explained as they are also chemical products. Maybe a line from RO to CH2O acknowledging that CH2O is coming from oxidation (likely most of it) although some is also directly emitted. For HONO, its origin is still not well understood, so maybe this should be acknowledged in the caption by noting that it's abundance cannot be fully explained by formation via OH+NO. In the literature, CH2O and H2O2 are typically described as being responsible for "secondary" radical production rather than "primary" radical sources. While there may be a large primary source of CH2O in this particular environment, the same is not true for H2O2. One potential solution is to avoid the whole use of the word "primary" and simply say that the "green circles represent species that contribute to radical formation."

Is there a significance to the two additional yellow stars in Figure 2?

Typo on line 277: O2 should be O3

The mention of an AIRPRO project first comes up on line 160. It is mentioned again in the caption of Figure 3, then in Table 4, and again on line 473. Otherwise, all other references in the text, figures, and tables are to APHH. Only in the summary (line 754) do the authors finally say the "APHH AIRPRO campaign". Is AIRPRO an acronym? If so, it is undefined. Is it even necessary to mention AIRPRO? If so, it is crucial to make a clear distinction between what is meant by APHH versus AIRPRO.

On line 226, the authors note that the model was constrained by HCHO, but this species is not shown in Figure 4. This may be because it had minimal influence on radical

production, but it is also a key outcome from VOC oxidation. While I am not surprised that it is small compared to HONO, I am somewhat surprised that it doesn't seem to be important compared to photolysis of other carbonyl species and alkene ozonolysis. I would like to see HCHO added to Figure 4.

There is no comment about using PAN as a constraint, so was it predicted by the model? How well does it compare? Could it be a radical source at the surface?

Figure 4 would also be improved if a few more things were added. For instance, a couple of VOCs (an alkene and an aromatic would be good). Also, there was a ceilometer at the site. Could mixing height be added to the figure? I expect it would be quite relevant to some of the variability marking the haze periods. While this information does not alter the outcome of the paper, it provides valuable additional context.

Figure 4: Extra tick marks at irregular intervals on the CO axis appear to be an error.

Figure 5: The panel for J(O1D) shows a small blip after dark for the red curve. I assume that this is an error. If not, what does it signify?

Figure 5: Are the solid lines medians? Would it be better to call the dashed lines the interquartile range rather than confidence intervals? These are after all being used to exhibit real ambient variability in the two populations of data being compared.

Some information on mixing depth would also be helpful in the discussion of figure 5. For instance, how important is containment in explaining the high values during the haze periods in addition to wind direction?

Line 338: "OH reactivity is discussed further in Section 2.5" (this is a typo that needs to be corrected)

Lines 339-341: "Figure 6 shows the steady state calculation for OH between 2/12/2016 to 8/12/2016 where it is compared with the measured OH concentrations. These days were chosen as full data coverage for HONO, NO, j values, radical and k(OH) measurements were available." Referring back to figure 4, it does not appear that HONO

measurements are available from 2/12-5/12. Have I missed something? HONO is accounted for in calculations for each day in Figure 6, but this does not seem to track what I see in Figure 4. For instance, the lowest NOx and highest ozone occurs on 5/12, so why does HONO make its greatest contribution on that day? I can't make sense of it.

Lines 343-346: Further discussion of Figure 6 states that "The agreement highlights that the OH budget can be determined by field measurements of the parameters necessary to quantify its rate of production and loss, and is closed to within 10%, well within the 26% error on the OH measurements themselves." I would agree that this plays out in the aggregate, but there is always value in looking at gradients that occur in the time series, and there is a significant discrepancy on 4/12 that falls well outside the 26% error that at least deserves mention if not some investigation or deeper explanation. Even 3/23 exhibits a shift in agreement after the peak that might be able to provide insight. Why does HO2+NO drop so much faster than measured OH on that day? This period on 3/23 requires substantial additional OH sources to make sense.

Lines 364-366: Authors state, "As seen in Figure 7, the measured daily maximum for the radical species varied day-to-day over the range 1 to 8 x 10ˆ6 cm-3, 0.7 to 1.5 x 10ˆ8 cm-3 and 1 to 2.5 x 10ˆ8 cm-3 for OH, HO2 and sum of RO2 respectively." I am again being nitpicky, but precision in your language is important, and I again feel like I am not looking at the same figure that is being described. For instance, which day shows peak OH at 1x10ˆ6? The lowest I see is ∼2.5x10ˆ6. For HO2, every day after 2/12 shows peak values well below the stated 0.7x10ˆ8. Similarly, for RO2 I see several days peaking at values less that the stated 1x10ˆ8. These imprecisions lower confidence in the other values you mention regarding over and underprediction of OH, HO2, and RO2 that cannot be deduced as easily from the figure.

In the caption for Figure 7, it is stated that the lines in panel (d) are from the model, but this seems unlikely. Is this a typo?

Lines 407-410: The authors state, "The ability of the model to reproduce (to within

~10%) both the OH reactivity and the OH concentration when constrained to measured HO2 (in MCM-cHO2), but not to reproduce RO2 radicals (whether constrained or not to HO2) is suggestive of an incomplete representation of the chemistry of RO2 radicals in the winter Beijing environment." This is somewhat of a throwaway statement. Under the extreme NOx conditions, both OH and its lifetime have very little dependence on RO2. Given the dominance of HO2+NO as a source of OH (80-90%) it is somewhat of a foregone conclusion that the constrained version of the model corrects the OH discrepancy. It is simply the lack of RO2 in the model that indicates missing RO2 chemistry. The bigger problem is explaining the HO2, which is partly derived from RO2. Are there any other notable changes when HO2 is constrained?

The discussion leading off section 4 feels like a step backward. At this point, it has already been established that the OH abundance is fully consistent with the observed HO2, based on both the photostationary state equation and the MCM-cHO2 model calculations. The latter calculations further demonstrate that the improved representation of OH does almost nothing to close the gap with RO2 observations (figure 8). For this reason, defining ROx as OH+HO2+RO+RO2 does not provide any additional insight. Both production and loss is dominated by OH reactions, which is not where you are looking to solve the problem. If radical production is dominated by HONO photolysis to produce OH and getting OH correct in the model does nothing to rectify RO2, I don't understand how this helps. It is just another way of showing the same thing that you have already shown in Figures 6 and 8. Also, when OH+NO dominates radical loss in the haze period, it isn't really a termination, but more akin to a null cycle for radicals since it will photolyze to return to OH on a short timescale. If you removed this cycling, and only accounted for HONO from other sources, the figure would be more accurate. Nevertheless, the dominance of OH reactions prevents this figure from advancing beyond what has already been demonstrated.

The comparisons given in Table 5 are fine and do not require figure 4, but they are also somewhat of a distraction as you have already determined that the OH can be explained. As a reader, I am expecting you to advance more quickly to the clear questions regarding RO2 established at the end of section 3.

Line 462: "As summarized in Table 2" should be "Table 5"

Line 467: "campaign" is misspelled

Table 4 and 5: "NCITT" should be "NACHTT"

Line 539: "as shown in Figure 10" should be "Figure 11"

Line 541: "and can almost be reproduced by the model at NO concentrations above 100 ppbv." I do not think this a valuable statement as there is no expectation that the model is getting such an answer for the right reason. Instead, what you are seeing is that NOx reactions effectively suppress complex RO2 concentrations at only a few ppbv in the model, while it appears that in the observations such suppression does not occur until NOx is well above 100 ppbv.

Line 545: "degredation" is misspelled and should be "degradation"

Section 4.3: It is not clear to me why this section is necessary to the paper. Everything to this point has been about trying to understand the model discrepancies with radical chemistry, especially at high NOx. At this point, I would expect some discussion of what might be pursued in the future to reconcile the problem. The foray into what these oxidants are doing in terms of aerosol formation feels like it belongs in another paper. I would shorten what is already a lengthy manuscript and remove this section.

Section 4.4: This section focusing on ozone production only makes sense to include if it attempts to reconcile to calculated rate of production with what is observed. Ozone itself is on the order of only 1-30 ppbv and Ox fluctuations are on the order of 10-15 ppbv per day based on what is shown in Figure 5. Thus, a formation rate of 71 ppb/hr on average would need to be offset by an equally large NOx sink via NO2+OH. Also, with such low ozone, it would seem that radicals play an outsized role in NOx cycling between NO and NO2. Has there been any analysis of NO/NO2 and its consistency

with the observed ozone and radical abundances?

Line 765: "2.7.0" please fix this typo

---

## Referee Comment (RC2) · Anonymous Referee #2 · 8 Jun 2020

This paper focuses on the investigation of the OH, HO2 and RO2 radical chemistry at extremely high NO in Beijing, China. As it was observed by a previous study in the same area (Tan et al., 2018) the current "known" chemistry at high NO cannot reproduce the measured HO2 and RO2 radicals resulting in a large underestimation of the ozone production.

I agree with reviewer one on the possibility of shortening the paper which, at the current status, feels more as a description of the observation (with some model run) but does not really try and push for suggesting possible explanations for the finding or even looking in explanations given in the past (segregation for example or Cl2 chemistry) to check if they would help the situation in this campaign.

General comments

[Figure]

I would suggest trying and making better use of the complex and simple RO2 concentrations. Measurement of RO2 or scarce to start with and here several time the measurement of simple and complex RO2 separately is brought up but then the data is not really used. Even when mentioning that there seems to be a better agreement between the measurement of simple RO2 and model results at high NO (which, by the way, I do not agree with), the discussion stops there and there is no additional use of the data. Why not checking for example if the RO2 measurement is consistent with the VOC load? Does the contribution of simple and complex RO2 changes with time? During the day? From non-haze to haze periods? I think this type of analysis could maybe also help understanding a little bit more where the large discrepancy between measurement and model results arises from...

I am missing a small but useful description of all the measurements used within the model and which instrumentation (with accuracy and precision) was used for the different trace gases. It does not have to go too much in details but there is no mentioning of how NO, which is extremely important for the radicals chemistry, was measured...or O3 or anything. In addition to this, there is no description of how the OH reactivity was measured and how much of a deviation from the mono-exponential decay could be expected for values of NO reaching up to 250 ppbv. What is the accuracy of the kOH measured at high NO? Could this represent a lower limit? This should be discuss appropriately and it could add an additional explanation of why the model is largely underestimating the RO2 and HO2 concentrations (lack of some primary VOCs).

Specific comments:

Page2 line46: "...quality are of serious concern..."

Page2 line49: "... of the world fastest..."

Page2 line51: I would drop the number after the comma and round the percentages

Page2 line 59: NOx, SO2 and VOCs have not been defined

[Figure]

Chapter 2.1 More information on the specific of the campaign site would be beneficial. Was the site on the street? On a platform? On the roof of the building? What was the distance between different instruments? I understand there is a specific paper on the topic but just two lines with a little bit information would suffice.

Chapter 2.2.2 Here as well more details on the sensitivity towards the different RO2 is needed. The different concentrations of RO2 are used later on to justify some of the conclusions on the discrepancies between model and measurements so it is important to mention how well know is the separation in two classes of RO2 and which sensitivity is applied for which classes.

Page8 line212: Is there really no difference between the accuracy of OH, HO2 and RO2 accounting that HO2 requires conversion into OH and RO2 requires a minimum of 2 NO steps?

Page9 line 239: What is the concentration of H2 to 500 ppbv included in the model needed for?

Page 9 line241: What was the time resolution of the GC data?

Page 11 line290: Is the diel variation shown the mean or the median of the data?

Page 11 line300: O3 does not react with high levels of NO but with a high concentration of NO

Page 21 Section 4.1: I assume that here only the results from the model are shown but this is not clear from reading the text.

Page 24 Lines516-521: Has the possibility of segregation of air been investigated and ruled out or why this is mentioned here but there is no discussion on how this could have had an impact on this specific site? It could be worth discussing if this could help bringing measurements and model results in agreement.

Page 24 line539: Assuming that figure 10 is actually figure 11 (where in the caption of

the figure the model line is the red one (?)), I do not agree with the statement in the paper that the model can reproduce the simple RO2 measured for NO above 100 ppbv. Actually, there is overlap between the model and the measured RO2 95th percentile for the complex RO2. In all honesty, I am not sure this plot tells us much as the model equally predicts pretty much zero RO2 expected at NO above 10 ppbv for both type of RO2. Although I agree that the simple RO2 have been studied more carefully, what would be the difference in rate with NO to justify the observed concentration of RO2 or what type of different chemistry for the most complex RO2 would be needed? There is no discussion in this study about it and some suggestions of what is feasible are needed.

Page 27 line 570-573: What would be the concentration of CL2 and/or ClNO2 needed to justify such a production of RO2? This could tell us if it could be possible at all.

Page 28 line602-605: I think one needs to be a bit careful here as, as you pointed out, the conditions (NO in particular) are not comparable and this campaign is one extreme case where NO is so large that dominates the losses of HO2 in any case. It is also worth citing the study by (Tan et al., 2020) which came to a similar conclusion for smaller levels of NO.

Section 4.3: Although I agree with reviewer 1 that this session is not really needed as it is descriptive and should be substitute by a better analysis of what could bring measurement and model results in agreement, I think at page 31 line 650 the statement that the model over predict HO2 in non-haze events is wrong. From figure 14 the model clearly under predicts HO2 radical in non-haze events. Same for page 33 line 684 where I do not clearly see a midday peak for NO3-?

Reference

Tan, Z., Rohrer, F., Lu, K., Ma, X., Bohn, B., Broch, S., Dong, H., Fuchs, H., Gkatzelis, G. I., Hofzumahaus, A., Holland, F., Li, X., Liu, Y., Liu, Y., Novelli, A., Shao, M., Wang, H., Wu, Y., Zeng, L., Hu, M., Kiendler-Scharr, A., Wahner, A., and Zhang, Y.: Winter-

time photochemistry in Beijing: observations of ROx radical concentrations in the North China Plain during the BEST-ONE campaign, Atmos. Chem. Phys., 18, 12391-12411, doi:10.5194/acp-18-12391-2018, 2018.

Tan, Z., Hofzumahaus, A., Lu, K., Brown, S. S., Holland, F., Huey, L. G., Kiendler-Scharr, A., Li, X., Liu, X., Ma, N., Min, K.-E., Rohrer, F., Shao, M., Wahner, A., Wang, Y., Wiedensohler, A., Wu, Y., Wu, Z., Zeng, L., Zhang, Y., and Fuchs, H.: No Evidence for a Significant Impact of Heterogeneous Chemistry on Radical Concentrations in the North China Plain in Summer 2014, Environmental Science & Technology, 54, 5973-5979, doi:10.1021/acs.est.0c00525, 2020.

---

## Referee Comment (RC3) · Anonymous Referee #3 · 27 Jun 2020

This paper presents wintertime measurements of OH, HO2, and RO2 radicals and OH reactivity in Beijing in November and December 2016 as part of the Air Pollution and Human Health in Chinese Megacities" (APHH) campaign. The measurements are separated into "haze" and "non-haze" events based on PM2.5 concentrations, RH, and visibility. The radical concentrations and OH reactivity were modeled using a 0D model based on the Master Chemical Mechanism (MCM) constrained by measured concentrations of HONO, HCHO, O3, NOx, and VOCs, and included several modifications to the model scenarios.

Similar to other winter measurements, the authors find that the measured radical concentrations were higher than expected based on the reduced photolysis rates leading to reduced rates of traditional ROx production reactions. The authors find that the

base MCM model underpredicted the concentration of OH, HO2, and RO2 radicals, especially during haze events and under high mixing ratios of NOx. It was found that the steady-state OH radical budget can be closed with HO2, HONO, and k(OH) measurements, suggesting that the measured OH concentrations are consistent with the measured HO2 concentrations, and the underestimation of radicals is due to a missing source of RO2 radicals. The high radical concentrations and total OH reactivity measured during winter haze events suggests that radical propagation and chemical processing are rapid even during the winter. The underestimation of radicals by the model, and in particular the underestimation of peroxy radicals, suggests that models are significantly underestimating the rate of ozone production during winter haze events.

The results are of interest to the atmospheric chemistry community, and the paper is worthy of eventual publication in ACP. It is quite long, and there appear to be several inconsistencies throughout (see minor comments), and there are several sections/tables that probably could be moved to the Supplement. For example, the discussion of NO2 and SO2 oxidation in relation to aerosols in section 4.3, as well as Table 6 in section 4.2 could probably be moved to the Supplement. However, the paper would benefit from some additional experimental details and clarifications described below.

Main comments:

1) While the title and main conclusions of the paper refer to wintertime haze events, the main modeling of the results summarized in Figure 8 appears to include both haze and non-haze events, while the brief discussion in section 4.3 separates the model analysis to haze and non-haze events, with Figure 14 showing the base model agreement worse under haze events. While the model appears to underestimate the measured RO2 concentration similarly for both events, the agreement of the predicted OH and HO2 concentrations with the measurements is better for the non-haze events. It appears from Figure 7 that the number of haze and non-haze events were roughly equal. As a result, it is not clear whether some of the main conclusions of the paper would be

applicable to the haze events. It would be useful to illustrate in Figure 6, 8, and 13 how the different models in Table 1 are able to reproduce the radical measurements for haze and non-haze events. Is the estimation of the missing source different for the haze and non-haze events? Are the model results/conclusions different for the different events? While they may not be significant, any differences between the events should be discussed in more detail.

2) The authors should clarify their definition of OHwave and OHchem on pages 6-7. The current description suggests that OHchem is the on-line background measurement including interferences, while OHwave is the off-line background measurement. However, Figure 3 compares the measured OH concentration determined using chemical modulation (signal – OHchem background) with that determined by spectral modulation (signal – OHwave background), not a comparison of the background signal measured by both methods.

3) Related to this, the authors state that the spectral modulation measurements were also corrected for laser-generated OH from ozone photolysis + H2O (page 7). Based on the Woodward-Massey et al. (2020) paper, it appears that the interference was calculated based on laboratory measurements of the interference as a function of ozone, water and laser power. This should be clarified. Since this interference would be measured by chemical modulation, a comparison of the measured interference with that calculated would provide additional confidence in the OHChem measurement as well as the accuracy of the interference estimate.

4) There is little discussion of the HO2, HO2*, and RO2 experimental measurement conditions, except that it appears that the conditions were similar to that in the ClearfLo study. The paper would benefit from a brief discussion of the experimental conditions employed in this study. It appears that only a single NO flow was used in the HOx detection cell for these measurements, in contrast to the use of two NO flows used to measure HO2 and HO2* (RO2i) during ClearfLo (Whalley et al., 2018). Instead it appears that HO2* was measured using the ROxLIF detection cell. While it is stated

that the ROxLIF method is described "in detail below" (page 5), the paper again references Whalley et al. (2018) instead of providing details. Given the high concentrations of NOx in this study, how did the authors account for potential interferences from the decomposition of HO2NO2 and CH3O2NO2? More details on the experimental measurements are needed. In addition the authors should clarify how the simple RO2 and complex RO2 were derived from the measurements. It appears that complex RO2 was obtained from the difference between the HO2* ROxLIF measurements and the FAGE HO2 measurements, while the simple RO2 were obtained from the difference between the ROxLIF RO2 and HO2*measurements. Much of this information could go into the Supplement.

5) Similarly, there is no discussion of the experimental method used to measure total OH reactivity. From the information given in Figure 7, it appeared that the OH reactivity was calculated based on the measured OH sinks, but it is clear from Figure 8 that total OH reactivity was measured. Is the measured OH reactivity shown in Figure 7? A brief description of the measurement technique should be included. Given the high mixing ratios of NO that were observed, did interference from the HO2+NO reaction impact the OH reactivity measurements?

Minor points:

Abstract: There have been previous measurements of radicals at similar NO levels in Mexico City (Shirley et al., ACP, 2006; Dusanter et al., ACP, 2009).

The caption in Figure 3 states that the gray points represent an acquisition cycle of 6 min, but the legend states that they are 4 min averages.

While the VOC measurements used to constrain their model are given in Table 1, the paper would benefit from additional information on the instruments used to measure the other model constraints. Even though this information may be provided in a separate campaign paper, a table similar to that in Whalley et al. (2018) could be included in the Supplement.

It appears from Figure 4 that HONO measurements were not available between 2/12 and 5/12, but the steady-state calculations shown in Figure 6 include data between 2/12-8/12 and were chosen "as full data coverage for HONO, NO, j values, radical and k(OH) measurements were available." Was HONO available on all these days?

Page 16 and Table 4: The text and table state that the average OH maximum was 2.7 E6 cm-3, but a value of 3.03 E6 cm-3 is stated on page 18.

Page 19: I am not sure late February/March would be considered mid-summer in Boulder, but rather late winter/early spring.

Figure 9: The authors should clarify whether this is an experimental radical budget or one derived from the model. Given the importance of HONO to radical initiation, how sensitive was the model to the systematic differences in the HONO measurements as described in Crilley et al. (2019)?

Page 26: There appears to be a problem with the signs in Equation 3 (see the corresponding equation in Tan et al. (2018)).

Page 26, line 560: Here it is stated that the P'(ROx) is 1.2 E8 cm-3 s-1, but on page 27 line 575 states that it is 1.01 E8 cm-3 s-1.

---

## Author Response (AR1)

**Reviewer 1.**

We thank the reviewer for their careful reading of the manuscript. We address each of the comments in turn below, with the comments first given in bold, followed by the response in normal type, followed by any changes made to the manuscript.

**Figure 1: I fully appreciate the challenge in creating efficient and accurate diagrams such as Figure 1 to portray the chemistry of radical cycling. The species in green are described as primary routes for radical formation. I agree with the way that ozone is portrayed given that its role as an initiator as well as a product of the chemistry is shown. Both HONO and HCHO need to be better explained as they are also chemical products. Maybe a line from RO to CH2O acknowledging that CH2O is coming from oxidation (likely most of it) although some is also directly emitted. For HONO, its origin is still not well understood, so maybe this should be acknowledged in the caption by noting that it's abundance cannot be fully explained by formation via OH+NO. In the literature, CH2O and H2O2 are typically described as being responsible for "secondary" radical production rather than "primary" radical sources. While there may be a large primary source of CH2O in this particular environment, the same is not true for H2O2. One potential solution is to avoid the whole use of the word "primary" and simply say that the "green circles represent species that contribute to radical formation."**

A line between RO to HCHO has been added, and as suggested by the reviewer the caption has been changed so that the green circles represent species that contribute to radical formation.

Updated figure below:

[Figure]

Updated/Modified caption: Figure 1. "The tropospheric photochemical cycle, with the green circles representing species acting as routes for radical formation, the blue circles representing the radical species themselves and the red circles representing the formation of secondary pollutants. The cycle does not show any heterogeneous source (e.g. heterogeneous production of HONO) or loss processes for the radical species. It should be noted the measured HONO abundance cannot be explained by the reaction of OH + NO alone"

**Is there a significance to the two additional yellow stars in Figure 2?**

No. In order to avoid confusion these have been removed from Figure 2 Please see updated Figure 2 below:

[Figure]

**Typo on line 277: O2 should be O3**

This has been fixed.

**The mention of an AIRPRO project first comes up on line 160. It is mentioned again in the caption of Figure 3, then in Table 4, and again on line 473. Otherwise, all other references in the text, figures, and tables are to APHH. Only in the summary (line 754) do the authors finally say the "APHH AIRPRO campaign". Is AIRPRO an acronym? If so, it is undefined. Is it even necessary to mention AIRPRO? If so, it is crucial to make a clear distinction between what is meant by APHH versus AIRPRO.**

We have removed AIRPRO from the paper so that the campaign is just called the APHH campaign.

**On line 226, the authors note that the model was constrained by HCHO, but this species is not shown in Figure 4. This may be because it had minimal influence on radical production, but it is also a key outcome from VOC oxidation. While I am not surprised that it is small compared to HONO, I am somewhat surprised that it doesn't seem to be important compared to photolysis of other carbonyl species and alkene ozonolysis. I would like to see HCHO added to Figure 4.**

HCHO has been added to Figure 4. See updated figure and caption below:

[Figure]

**Figure 4.** Time-series of $j(O^1D)$, relative humidity (RH), temperature (Temp), CO, $SO_2$, $O_3$, $NO_x$, HONO, boundary layer (BL), $PM_{2.5}$, HCHO, butane and toluene from the 8th of November to 10th December 2016 at Institute of Atmospheric Physics (IAP), Beijing.

As shown in Figure 9 the photolysis of HCHO contribute ~2% to the primary formation of $RO_x$.

**There is no comment about using PAN as a constraint, so was it predicted by the model? How well does it compare? Could it be a radical source at the surface?**

PAN was not measured during the winter APHH campaign (only during the summer campaign) thus no comparison could be made between modelled and measured values. Indeed, it may be a radical source at the surface, however, the analysis of production and termination of radicals shown in Figure 9 shows a net-formation of PAN – suggesting that PAN is acting as an overall net $RO_2$ sink.

**Figure 4 would also be improved if a few more things were added. For instance, a couple of VOCs (an alkene and an aromatic would be good). Also, there was a ceilometer at the site. Could mixing height be added to the figure? I expect it would be quite relevant to some of the variability marking the haze periods. While this information does not alter the outcome of the paper, it provides valuable additional context.**

HCHO, butane, toluene and boundary layer height have all been added to Figure 4 (see response above for updated version of the Figure 4 and caption), and a short discussion about this has been added.

Modified text related to Figure 4:

"The median diel variation in $j(O^1D)$, relative humidity (RH), temperature, CO, $SO_2$, $O_3$, NO,$NO_2$, HONO, $PM_{2.5}$, boundary layer height (BL), HCHO, butane and toluene is shown in Figure 4."

"The VOC concentration (HCHO, toluene and butane) track pollution events and each other very well; the mole fraction of the VOCs varied between 0.2 - 11.3 ppbv."

The average diel profile of boundary layer height has also been added to Figure 5, replacing the $SO_2$ panel, both for inside and outside of haze, along with a short discussion.

Updated version of Figure 5 with the updated caption:

[Figure]

**Figure 5.** Comparison of the median average diel variation for $j(O^1D)$ (s$^{-1}$), NO (ppbv), O$_3$ (ppbv), CO (ppbv), O$_x$ (ppbv), NO$_2$ (ppbv), HONO (ppbv) and boundary layer height (m) inside and outside haze events; denoted by solid red and blue lines, respectively. The dashed lines represent the interquartile range for the respective species and pollution period.

Modified text relating to Figure 5:

"The boundary layer height (BLH) shows a similar diurnal variation inside and outside of haze, although the maximum BLH in haze is shifted to 14:30 compared to 12:30 outside of haze. The maximum and minimum BLH is similar inside and outside of haze and shows that containment is not the only driving force for pollution periods."

**Figure 4: Extra tick marks at irregular intervals on the CO axis appear to be an error.**

These have been corrected on Figure 4 (see response above which shows an updated Figure 4).

**Figure 5: The panel for J(O1D) shows a small blip after dark for the red curve. I assume that this is an error. If not, what does it signify?**

We apologise, this was an error and has been corrected in Figure 5 (see above).

**Figure 5: Are the solid lines medians? Would it be better to call the dashed lines the interquartile range rather than confidence intervals? These are after all being used to exhibit real ambient variability in the two populations of data being compared.**

The solid lines are indeed the median values and we have changed the caption from confidence intervals to interquartile range.

Modified text: "Comparison of the median average diel variation for $j(O^1D)$ ($s^{-1}$), NO (ppbv), $O_3$ (ppbv), CO (ppbv), $O_x$ (ppbv), $NO_2$ (ppbv), HONO (ppbv) and boundary layer height (m) in and outside haze events; denoted by solid red and blue lines, respectively. The dashed lines represent the interquartile range for the respective species and pollution period."

**Some information on mixing depth would also be helpful in the discussion of figure 5. For instance, how important is containment in explaining the high values during the haze periods in addition to wind direction?**

A panel showing the boundary layer height inside and outside of the haze has been added to Figure 5 replacing the $SO_2$ panel (see response above), and shows that containment is not a large factor for explaining the high values observed during haze periods.

Modified text: "The boundary layer height (BLH) shows a similar diurnal variation inside and outside of haze, although the maximum BLH in haze is shifted to 14:30 compared to 12:30 outside of haze. The maximum and minimum BLH is similar inside and outside of haze and shows that containment is not the only driving force for pollution periods."

**Line 338: "OH reactivity is discussed further in Section 2.5" (this is a typo that needs to be corrected)**

This has been fixed in the revised MS.

**Lines 339-341: "Figure 6 shows the steady state calculation for OH between 2/12/2016 to 8/12/2016 where it is compared with the measured OH concentrations. These days were chosen as full data coverage for HONO, NO, j values, radical and k(OH) measurements were available." Referring back to figure 4, it does not appear that HONO measurements are available from 2/12-5/12. Have I missed something? HONO is accounted for in calculations for each day in Figure 6, but this does not seem to track what I see in Figure 4. For instance, the lowest NOx and highest ozone occurs on 5/12, so why does HONO make its greatest contribution on that day? I can't make sense of it.**

The HONO dataset shown in Figure 4 was from one HONO instrument only and the HONO used in the steady-state calculation was the HONO concentration recommended by Crilley et al. (2019) based on measurements by several instruments during the campaign, and represents a more complete dataset. The HONO shown in Figure 4 has now been updated to those recommended by Crilley et al. (2019), and are the values that have been used in the steady-state calculation and MCM model. Low $NO_x$ would lead to reduction in recycling from $HO_2$ + NO, which is the largest source of OH production, and

hence on 5/12 at the lowest NOx, this makes HONO the largest contributor to the rate of OH production. Figure 4 has been updated with the correct HONO dataset, see response above for the updated version of Figure 4 along with the updated caption.

Crilley, L. R., Kramer, L. J., Ouyang, B., Duan, J., Zhang, W., Tong, S., Ge, M., Tang, K., Qin, M., Xie, P., Shaw, M. D., Lewis, A. C., Mehra, A., Bannan, T. J., Worrall, S. D., Priestley, M., Bacak, A., Coe, H., Allan, J., Percival, C. J., Popoola, O. A. M., Jones, R. L., and Bloss, W. J.: Intercomparison of nitrous acid (HONO) measurement techniques in a megacity (Beijing), Atmos. Meas. Tech., 12, 6449–6463, https://doi.org/10.5194/amt-12-6449-2019, 2019.

**Lines 343-346: Further discussion of Figure 6 states that "The agreement highlights that the OH budget can be determined by field measurements of the parameters necessary to quantify its rate of production and loss, and is closed to within 10%, well within the 26% error on the OH measurements themselves." I would agree that this plays out in the aggregate, but there is always value in looking at gradients that occur in the time series, and there is a significant discrepancy on 4/12 that falls well outside the 26% error that at least deserves mention if not some investigation or deeper explanation. Even 3/23 exhibits a shift in agreement after the peak that might be able to provide insight. Why does HO2+NO drop so much faster than measured OH on that day? This period on 3/23 requires substantial additional OH sources to make sense.**

The steady-state Figure 6 has been revised and has been generated using kOH values from optimised fitting of the OH decays (consideration of the start and end of the decay fit), as used in the MCM modelling comparison. See below for updated version of Figure 6. The diel profile has been separated into haze and non-haze periods as recommended by reviewer 3.

[Figure]

**Figure 1.** Average diel profile for observed and steady state calculated OH concentrations for: (a) non-haze, and(b) haze periods. Panel (c) shows a comparison time-series for the steady state calculation of OH and measured OH. The OH generated by $O^1D+H_2O$, although included in the key, is too small to be visible.

Figure S7 below shows that if the measured or modelled kOH is used on the 03/12 the PSS and measured OH agree within error. Although we saw no evidence of OH recycling in the kOH decay curves (no bimolecular behaviour – see response to reviewer 2), the measured kOH was lower than the modelled on the 04/12 and the use of the measured kOH may have caused the PSS to over-estimate OH on this day. However, using the modelled kOH in the PSS calculation does reproduce the PSS calculated OH using the measured kOH and the PSS stills overpredicts the OH by a factor of ~2.4. The large overprediction by the PSS suggests the differences between the PSS and measured OH on the 04/12/2016 stems from measurement problems and could be derived from issues with the OH, $HO_2$, HONO or NO measurements on this day. On the 3/12, the $HO_2$ + NO drop is driven by a decrease in NO, however the PSS and measured OH do agree within the error on the OH measurements.

The text in the MS has been modified: "Although on the 04/12/2016 the PSS overpredicts the measured OH by a factor of ~2.5, the differences between the PSS and measured OH could be due to a variety of reason including errors in OH, $HO_2$, NO, kOH and HONO measurements and NO segregation across the site. A further discussion for the PSS for the 04/12 can be found in the supplementary section S1.6."

The information added to the Supplementary Information is as follows:

"**S1.6 In-depth comparison of measured OH and OH calculated from the PSS on the 04/12 using measured and modelled OH reactivity.**

On the 04/12/2016 the PSS calculation for OH is overpredicted by ~2.5 and the modelled OH reactivity is higher than the measured OH reactivity by an average of ~14 $s^{-1}$. The modelled OH reactivity was used in the PSS calculation for OH and a comparison between the PSS calculation using measured and modelled kOH and measured OH is shown in Figure S7. Figure S7 shows that whilst using the modelled OH reactivity does reduce the calculated PSS OH, the PSS using modelled kOH still overpredicts the measured OH by a factor of ~2.4. The large overprediction by the PSS suggests the differences between the PSS and measured OH on the 04/12/2016 stems from measurement problems and could be derived from issues with the OH, $HO_2$, HONO or NO measurements on this day.

[Figure]

**Figure S1.** Comparison of measured OH (with errors, blue bars) with OH calculated from a photostationary steady-state (PSS) calculation using measured OH reactivity. The contributions towards OH production from HONO + hv (green) and $HO_2$ + NO (red) are shown, as well as the OH calculated using the PSS but with modelled OH reactivity (black)."

**Lines 364-366: Authors state, "As seen in Figure 7, the measured daily maximum for the radical species varied day-to-day over the range 1 to 8 x 10ˆ6 cm-3, 0.7 to 1.5 x 10ˆ8 cm-3 and 1 to 2.5 x 10ˆ8 cm-3 for OH, HO2 and sum of RO2 respectively." I am again being nitpicky, but precision in your language is important, and I again feel like I am not looking at the same figure that is being described. For instance, which day shows peak OH at 1x10ˆ6? The lowest I see is ~2.5x10ˆ6. For HO2, every day after 2/12 shows peak values well below the stated 0.7x10ˆ8. Similarly, for RO2 I see several days peaking at values less that the stated 1x10ˆ8. These imprecisions lower confidence in the other values you mention regarding over and underprediction of OH, HO2, and RO2 that cannot be deduced as easily from the figure**

These inconsistencies have been corrected and the precision in the wording tightened up, and the other values have been checked for accuracy. Originally, we used some values from the first measurement period during 17/12 – 19/12 but then discounted these as no measurements were available at midday for OH. For $HO_2$, we apologise, the power was given incorrectly, the last day peak is at $0.7 \times 10^7$. The other values have been checked and are correct.

New range of concentrations: OH – $2.3 – 8 \times 10^6$ cm$^{-3}$, $HO_2$ – $0.07 – 1.5 \times 10^8$, $RO_2$ – $0.8 – 2 \times 10^8$

Modified wording, "As seen in Figure 7, the measured daily maximum for the radical species varied day-to-day over the range 2.5 to 8 x $10^6$ cm$^{-3}$, 0.07 to 1.5 x $10^8$ cm$^{-3}$ and 0.8 to 2 x $10^8$ cm$^{-3}$ for OH, $HO_2$ and sum of $RO_2$ respectively."

**In the caption for Figure 7, it is stated that the lines in panel (d) are from the model, but this seems unlikely. Is this a typo?**

Yes. This has now been corrected in the caption.

**Lines 407-410: The authors state, "The ability of the model to reproduce (to within ~10%) both the OH reactivity and the OH concentration when constrained to measured HO2 (in MCM-cHO2), but not to reproduce RO2 radicals (whether constrained or not to HO2) is suggestive of an incomplete representation of the chemistry of RO2 radicals in the winter Beijing environment." This is somewhat of a throwaway statement. Under the extreme NOx conditions, both OH and its lifetime have very little dependence on RO2. Given the dominance of HO2+NO as a source of OH (80-90%) it is somewhat of a foregone conclusion that the constrained version of the model corrects the OH discrepancy. It is simply the lack of RO2 in the model that indicates missing RO2 chemistry. The bigger problem is explaining the HO2, which is partly derived from RO2. Are there any other notable changes when HO2 is constrained?**

We feel this is still an important statement to make as it also highlights that the OH and $HO_2$ measurements are self-consistent. MCM-cHO2 also increases the $RO_2$ concentration by a factor of ~3.5 compared to MCM-base, but the $RO_2$ is still underpredicted by a factor ~7. We have added statement to the paper to make these points as follows.

Modified text: "MCM-cHO2 also increases the $RO_2$ concentration by ~3.5 compared to MCM-base, but the $RO_2$ is still underpredicted by a factor ~7."

We agree with the referee that this highlights missing $RO_2$ chemistry and this is discussed later in the paper.

**At this point, it has already been established that the OH abundance is fully consistent with the observed HO2, based on both the photostationary state equation and the MCM-cHO2 model calculations. The latter calculations further demonstrate that the improved representation of OH does almost nothing to close the gap with RO2 observations (figure 8). For this reason, defining ROx as OH+HO2+RO+RO2 does not provide any additional insight. Both production and loss is dominated by OH reactions, which is not where you are looking to solve the problem. If radical production is dominated by HONO photolysis to produce OH and getting OH correct in the model does nothing to rectify RO2, I don't understand how this helps. It is just another way of showing the same thing that you have already shown in Figures 6 and 8. Also, when OH+NO dominates radical loss in the haze period, it isn't really a termination, but more akin to a null cycle for radicals since it will photolyze to return to OH on a short timescale. If you removed this cycling, and only accounted for HONO from other sources, the figure would be more accurate. Nevertheless, the dominance of OH reactions prevents this figure from advancing beyond what has already been demonstrated.**

Figure 9 highlights that most of the $RO_2$ in the model derives from OH sources, and highlights the need for additional primary $RO_2$ sources, this has clarified this in the paper using the statement below:

Modified text, line 508-509: "Figure 9. Shows that almost all of the $RO_2$ species in the model are derived from OH sources highlighting the need for additional primary $RO_2$ sources in the model."

We have also removed the cycling between OH + NO and HONO photolysis as suggested by the referee so now termination is not shown through the OH + NO pathway, and Figure 9 now only accounts for the HONO from other sources. The percentages for termination and production in the paper and Table 5 (which have been removed to the Supplementary Information) have also been updated.

[revised manuscript text omitted]

**The comparisons given in Table 5 are fine and do not require figure 4, but they are also somewhat of a distraction as you have already determined that the OH can be explained. As a reader, I am expecting you to advance more quickly to the clear questions regarding RO2 established at the end of section 3.**

We agree, and so this table and the comparison/discussion with previous campaigns (other than the discussion in the text for BEST-ONE and PKU campaigns) have been moved into the supplementary material. See response above.

**Line 462: "As summarized in Table 2" should be "Table 5"**

This line is no longer in the paper as it has been moved into the supplementary material, see response above and is now "As summarised in S4 Table 1".

**Line 467: "campaign" is misspelled**

This has been corrected.

**Table 4 and 5: "NCITT" should be "NACHTT"**

Corrected in both tables.

**Line 539: "as shown in Figure 10" should be "Figure 11"**

This has been corrected.

**Line 541: "and can almost be reproduced by the model at NO concentrations above 100 ppbv." I do not think this a valuable statement as there is no expectation that the model is getting such an answer for the right reason. Instead, what you are seeing is that NOx reactions effectively suppress complex RO2 concentrations at only a few ppbv in the model, while it appears that in the observations such suppression does not occur until NOx is well above 100 ppbv**

We agree and this statement has now been removed from the paper.

**Line 545: "degredation" is misspelled and should be "degradation"**

This has been corrected.

**Section 4.3: It is not clear to me why this section is necessary to the paper. Everything to this point has been about trying to understand the model discrepancies with radical chemistry, especially at high NOx. At this point, I would expect some discussion of what might be pursued in the future to reconcile the problem. The foray into what these oxidants are doing in terms of aerosol formation feels like it belongs in another paper. I would shorten what is already a lengthy manuscript and remove this section.**

We still feel that it is an important outcome of higher than expected concentrations of oxidants, namely that the rate of secondary aerosol production is also higher than expected. However, we agree about trying to shorten the manuscript and so have moved this section to the supplementary information, with just a sentence to signpost this in the main paper.

Signposting text in main text: "A discussion on the impact of similar OH concentration inside and outside of haze on the oxidation of SO2 and NO2 can be found in the supplementary section S1.3. "

Section added to the supplementary information shown below:

**"S1.3 $NO_2$ and $SO_2$ oxidation during haze events**

Secondary oxidation products, such as nitric acid and sulphuric acid, which partition to the aerosol phase, are major contributors towards the formation of secondary particulate matter (Huang et al., 2014). The OH measurements enable calculation of the rate of $SO_2$ and $NO_2$ oxidation *via* reaction with OH, to form gas-phase phase $HNO_3$ and $H_2SO_4$. Figure 4 shows that on average 1.5 ppbv/h and 0.03 ppbv/h of gas-phase $NO_2$ and $SO_2$ are oxidised to form acidic species, and that the oxidation increases in these haze periods caused by comparable OH concentration in and out of haze and, as shown in Figure 4, an increase in local $NO_2$ and $SO_2$ concentrations. $NO_x$ can also be lost in the atmosphere by the formation of $N_2O_5$ (Evans, 2005) and subsequent hydrolysis, but this is uninportant in Beijing during winter due to the low levels of $O_3$. The reaction of OH + $SO_2$ in the gas-phase is the rate-determining step in the formation $SO_4^{-2}$, so the $H_2SO_4$ formed in the gas-phase will partition in the aerosol phase (Barth et al., 2000). $H_2SO_4$ is effectively a non-volatile gas at atmospheric temperatures, and $H_2SO_4$ condensation onto pre-exsisiting particles is an irreversible kinetic process (Zaveri et al., 2008). Whilst $HNO_3$ is a semivolatile species and the gas-particle partitioning is highly sensitive to to meteorological conditions including: temperature, RH, particle size distribution, pH and particle composition. If the realtive humidity is lower than the deliquescence relative humidity ($RH_d$), then the $HNO_3$ that is formed in the gas phase reacts with $NH_3$ to form ammonium nitrate aersol ($NH_4NO_3$):

$$HNO_3(g) + NH_3(g) \rightleftharpoons NH_4NO_3(s) \qquad\qquad\qquad S\ R1$$

$$\text{HNO}_3(\text{g}) + \text{NH}_3(\text{g}) \rightleftharpoons \text{NO}_3^-(\text{aq}) + \text{NH}_4^+(\text{aq})$$  S R2

If the ambient RH exceeds the $RH_d$ then $HNO_3$ and $NH_3$ dissolve into the aqueous phase (aq):
To take into account the reversible process, knowledge of the $RH_d$ that marks the transition between the solid and the aqueous phase, and the equilibrium constant, $K_p$, for the two phase is required (Ackermann et al., 1998). The MADE module (modal aerosol dynamics model for europe) uses these thermodynamic parameters as given by (Mozurkewich, 1993), resulting in:

$$\text{In}\left(\frac{RH_d}{100}\right) = \frac{618.3}{T} - 2.551$$  S E1

for $RH_d$ and:

$$\text{In}\left(K_p\right) = 118.87 - \frac{24084}{T} - 6.025\ln(T)$$  S E2

for $K_p$. SE1 and SE2 shows that nitrate formation is favoured thermodynamically at low temperatures and high relative humidties (Ge et al., 2017). Previous measurements of $SO_4^{-2}$ and $NO_3^-$ made in wintertime Beijing suggests that photochemstry is important in the formation of nitrate aersol, but not the formation of sulphate (Ge et al., 2017; Sun et al., 2013).
Figure S4 also shows that the gas-phase oxidation of $NO_2$ increases under haze conditions, showing that nitrate formation is driven by photochemistry in haze events despite the lower photolysis rates. Similar conclusions have been made in Lu et al. (2019) from measurements during the BEST-ONE campaign; with $SO_4^-$ aerosol predominantly driven by aqueous-phase chemistry whilst the production of $NO_3^-$ aerosol from gas-phase oxidation of $NO_2$ with OH is important. The maximum production rate of $HNO_3$ observed during the BEST-ONE campaign is the same as the one calculated for the APHH campaign (3 ppbv $hr^{-1}$). The BEST-ONE campaign assumed all the gas-phase $HNO_3$ formed partitioned into the aerosol-phase due to the high relative humidity observed during the campaign.

[Figure]

**Figure S2** Average diel profiles of the rate of oxidation of $NO_2$ (left) and $SO_2$ (right) via reaction with OH in non-haze (blue) and haze (red) conditions.

"

**Section 4.4: This section focusing on ozone production only makes sense to include if it attempts to reconcile to calculated rate of production with what is observed. Ozone itself is on the order of only 1-30 ppbv and Ox fluctuations are on the order of 10-15 ppbv per day based on what is shown in Figure 5. Thus, a formation rate of 71 ppb/hr on average would need to be offset by an equally large NOx sink via NO2+OH. Also, with such low ozone, it would seem that radicals play an outsized role in NOx cycling between NO and NO2. Has there been any analysis of NO/NO2 and its consistency with the observed ozone and radical abundances?**

We feel that the inclusion of the ozone production rate is important as it highlights how this missing $RO_2$ chemistry impacts the models ability to predict *in situ* ozone formation.

The Leighton ratio calculated from the measured ozone, NO and $NO_2$ concentrations from the campaign is generally below 1 indicating that the ratio of $NO/NO_2$ are not in steady-state during the winter campaign and so it is not possible to estimate a peroxy radical concentration from this.

Also, a comparison between measured $O_3$ and ozone production rate is difficult as the changes in measured ozone will be controlled by transport and dilution which the box model does not take into account. In the case of this work the ozone concentration is likely reduced by mixing with $O_3$ depleted air – such as air from the nearby large roads in Beijing. Here were are showing that the instantaneous rate of ozone production is underestimated by the model compared with measured levels of $HO_2$ and $RO_2$ are used.

**Line 765: "2.7.0" please fix this typo**

Fixed

**Reviewer 2.**

We thank the reviewer for their careful reading of the manuscript. We address each of the comments in turn below, with the comments first given in bold, followed by the response in normal type, followed by any changes made to the manuscript.

**I agree with reviewer one on the possibility of shortening the paper which, at the current status, feels more as a description of the observation (with some model run) but does not really try and push for suggesting possible explanations for the finding or even looking in explanations given in the past (segregation for example or Cl2 chemistry) to check if they would help the situation in this campaign.**

As outlined in the responses to Reviewer 1, we have now shortened the manuscript by either shortening or completely removing Tables, Figures and Sections, and moving these to the Supplementary Material.

Unfortunately, there were no $ClNO_2$ measurements during the winter campaign, and hence it was not possible to calculate a time series for Cl atoms formed from photolysis of $ClNO_2$ and to assess any additional $RO_2$ radicals generated. Using the model run where an additional $RO_2$ source was added to reconcile the measurements and the model, a rough calculation has shown that the $ClNO_2$ concentration would have to be of the order of ~5800 ppbv in order to close the gap between modelled and measured $RO_2$. Previous measurements of $ClNO_2$ in suburban Beijing has shown a peak of ~2.9 ppbv (Wang et al. 2017) which is ~3 orders of magnitude smaller than the $ClNO_2$ concentration required, suggesting other additional primary sources are needed in the model besides Cl chemistry.

Added statement to paper about chlorine chemistry (page 28, line 615 – 619): "Although the $ClNO_2$ concentration required to bridge the gap between model and measurements would be ~5800 ppbv on average (see supplementary section S1.8 for details). Previous measurements in China in suburban Beijing have shown $ClNO_2$ peaking at 2.9 ppbv (Wang et al. 2017), however, and suggests other additional primary source are needed in the model besides Cl chemistry."

This section has been added into the supplememtary material:

"

**S1.8 $ClNO_2$ and Cl concentration required to bridge the gap between measured and modelled total $RO_2$**

Unfortunately, there were no $ClNO_2$ measurements during the winter campaign, and hence it was not possible to calculate a time series for Cl atoms formed from photolysis of $ClNO_2$ and to assess any additional $RO_2$ radicals generated. Using the model run where additional $RO_2$ source was added to reconcile the measurements and the model a rough calculation has shown that the $ClNO_2$ concentration would have to be on average ~5800 ppbv in order to close the gap between modelled and measured $RO_2$. Figure S10 shows the average diel of the calculated $ClNO_2$ and Cl concentration with peak at $1.4 \times 10^4$ ppbv and $1.6 \times 10^6$ molecule $cm^{-3}$, respectively. The $ClNO_2$ and Cl concentration have been calculated using SE3 – SE5:

$$\text{P}'\text{RO}_2 = k_{\text{VOC+Cl}}[\text{VOC}][\text{Cl}] \qquad \text{S E3}$$

$$[\text{Cl}] = \frac{\text{P}'\text{RO}_2}{k_{\text{VOC+Cl}}[\text{VOC}][\text{Cl}]} \qquad \text{S E4}$$

$$[\text{ClNO}_2] = \frac{k_{\text{VOC+Cl}}[\text{VOC}][\text{Cl}]}{j\text{ClNO}_2} \qquad \text{S E5}$$

where $k_{\text{VOC+Cl}}$ is a generic rate constant to represents the reaction of all VOCs with Cl which in this case is 4 x $10^{-12}$ molecule$^{-1}$ cm$^3$ s$^{-1}$, [VOC] is the sum of the measured VOC concentration for the campaign and P'RO$_2$ is the calculated additional RO$_2$ used in MCM-PRO2 (see main paper section 4.2 for more details). The ClNO$_2$ required to bridge the gap between measured and modelled of RO$_2$ is ~3 orders of magnitude greater than the peak ClNO$_2$ concentration measured in suburban Beijing (2.9 ppbv) by Wang et al. (2018) suggesting that other additional primary source are needed in the model besides Cl chemistry .

[Figure]

**Figure S10** Average diel of the ClNO$_2$ and Cl atom concentration required to bridge the gap between measured and modelled RO$_2$. The ClNO$_2$ and Cl concentrations have been calculated from the additional primary source of RO$_2$ added to the MCM-PRO2 model run, see section 4.2 in the main paper for more details.

"

Wang, X., Wang, H., Xue, L., Wang, T., Wang, L., Gu, R., Wang, W., Tham, Y.J., Wang, Z., Yang, L. and Chen, J., 2017. Observations of N2O5 and ClNO2 at a polluted urban surface site in North China: High N2O5 uptake coefficients and low ClNO2 product yields. *Atmospheric environment*, *156*, pp.125-134.

Regarding segregation, there were several instruments for NO measurements located at different positions around the field-site and there were no obvious differences between the measurements, and so we feel that NO segregation between the instruments cannot account for the differences between the measured and modelled RO$_2$. We have made a statement in the text (page 25, line 563-564) regarding this: ". There were several instruments for NO measurements located around the site and no differences in concentrations were observed, hence no evidence of any obvious segregation"

**I would suggest trying and making better use of the complex and simple RO2 concentrations. Measurement of RO2 or scarce to start with and here several time the measurement of simple and complex RO2 separately is brought up but then the data is not really used. Even when mentioning**

**that there seems to be a better agreement between the measurement of simple RO2 and model results at high NO (which, by the way, I do not agree with), the discussion stops there and there is no additional use of the data. Why not checking for example if the RO2 measurement is consistent with the VOC load?**

We have now included an additional analysis of the complex and simple $RO_2$ concentrations, and their agreement with the model (see the next comment below). Regarding whether $RO_2$ simple and $RO_2$ complex are consistent with the VOC load, the increase is kOH contribution for VOCs from non-haze to haze periods has been assessed. It shows that the increased contribution to kOH ($s^{-1}$) from VOCs going from non-haze to haze is a factor of: ~10 for aromatics, ~8 for alkenes and alkynes , ~6 for alkanes, ~9 for alcohols and ~2 for aldehydes. The large increase in relative contribution to kOH from aromatics, alkenes and alkynes is consistent with the observation of higher complex $RO_2$ (compared to simple $RO_2$) during haze periods compared to non-haze periods.

The statement "The increased contribution to kOH ($s^{-1}$) from VOCs going from non-haze to haze conditions is a factor of: ~10 for aromatics, ~8 for alkenes and alkynes , ~6 for alkanes, ~9 for alcohols and ~2 for aldehydes. The large increase in the relative contribution to kOH from aromatics, alkenes and alkynes is consistent with the observation of higher complex $RO_2$ (compared to simple $RO_2$) during haze periods compared to non-haze periods." has been added to the paper.

The statements: "and can almost be reproduced by the model at NO concentrations above 100 ppbv." Has been removed from the paper as suggested by reviewer 1.

**Does the contribution of simple and complex RO2 changes with time? During the day? From non-haze to haze periods? I think this type of analysis could maybe also help understanding a little bit more where the large discrepancy between measurement and model results arises from.**

The average diel profile of both measured and modelled complex and simple $RO_2$ inside and outside of haze has been added to Figure 14, and we have added the following text to the paper. "The measured complex $RO_2$ radical species peak at similar concentrations inside (4.3 x $10^7$ molecule $cm^{-3}$) and outside (4.6 x $10^7$ molecule $cm^{-3}$) of haze.  Interestingly, unlike the complex $RO_2$, the simple $RO_2$ concentration peaks at a lower concentration inside of haze (3.4 x $10^7$ molecule $cm^{-3}$) compared with outside of haze (5.5 x $10^7$ molecule $cm^{-3}$).  The complex $RO_2$ is undepredicted by the model by a factor of ~48 and ~12 inside and outside of haze, respectively, whilst the simple $RO_2$ is undepredicted by a factor of  ~66 and ~5.7 inside and outside of haze, respectively. The sharp increase for the underprediction of both simple and complex $RO_2$ inside haze events highlights the need of a large additional primary source of both simple and complex $RO_2$".

[Figure]

**Figure 14.** Average diel profiles for measured and modelled OH, HO₂, total RO₂, complex RO₂ (RO₂ comp), simple RO₂ (RO₂ simp) and kOH separated into haze (right) and non-haze (left) periods.

The average diurnal profile of measured and modelled simple and complex RO₂ have been added to Figure.8., and the updated Figure 8. Is shown below:

[Figure]

**Figure 8.** Campaign averaged diel profile of OH (a), HO$_2$ (b), sum of RO$_2$ (c), complex RO$_2$ (d), simple RO$_2$ (e) for measurements (blue) and box-model calculations: MCM-base (red) and MCM-cHO$_2$ (green). See text for descriptions of each model scenario. (f) – OH reactivity (s$^{-1}$) for measurements (black line) and model (stacked plot) with the contribution to reactivity from different measured species and modelled intermediates shown in the key.

Along with a small discussion on the variability during the day. The new text is as follows:.

"The complex and simple RO$_2$ show a very similar diurnal profile both peaking at 12:30 at a concentration of 4.4 x 10$^7$ molecule cm$^{-3}$ and 4.5 x 10$^7$ molecule cm$^{-3}$, respectively. The model underpredicts the simple and complex RO$_2$ at 12:30 by a factor of 30 and 22, respectively. The large underprediction of both simple and complex RO$_2$ highlights the needs for additional primary sources

forming both simple and complex species in the model. Section 4.2 explores the impact of additional primary source of $RO_2$ added into the model on OH and $HO_2$

**I am missing a small but useful description of all the measurements used within the model and which instrumentation (with accuracy and precision) was used for the different trace gases. It does not have to go too much in details but there is no mentioning of how NO, which is extremely important for the radicals chemistry, was measured. . .or O3 or anything.**

We have added a Table (Table 2) which describes the methods used for some of the key species which are used to constrain the model. For many of the other species used to constrain the model, details are given in Shi et al 2018, and we have made a clear reference to that paper.

Modified wording "The accuracy and precision of trace gas species can be found in Table 2, details on the HONO measurements used in the modelling scenarios can be found in Crilley et al.(2019). Details for other measurements can be found in Shi et al.(2018)"

The following table has been added to the manuscript:

| Instrument | Technique | $2\sigma$ Uncertainty / % | $2\sigma$ Precision/ ppbv |
|---|---|---|---|
| $O_3$, TEi49i | UV absorption | 4.04 | 0.28[1] |
| NO, TEi42i-TL | Chemiluminescence via reaction with $O_3$ | 4.58 | 0.03[1] |
| $SO_2$, TEi43i | UV fluorescence | 3.12 | 0.03[1] |
| $NO_2$, CAPS, T500U | Cavity enhanced absorption spectroscopy | 5.72 | 0.04[1] |
| HONO | LOPAP x2, BBCEAS x 2, ToF-CIMS and SIFT-MS | 9 – 22% | 0.025 – 0.130 |

**Table 2.** Instruments and techniques used to measure key model constraints. $2\sigma$ uncertainties for the measured trace gas species used in the modelling scenarios are quoted. [1]Precision is given for 15-minute averaging time. For details of the HONO measurements please see Crilley et al.(2019).

**In addition to this, there is no description of how the OH reactivity was measured and how much of a deviation from the mono-exponential decay could be expected for values of NO reaching up to 250 ppbv. What is the accuracy of the kOH measured at high NO? Could this represent a lower limit? This should be discuss appropriately and it could add an additional explanation of why the model is largely underestimating the RO2 and HO2 concentrations (lack of some primary VOCs).**

The kOH decays show no biexponential behaviour suggesting that recycling from $HO_2$ + NO was not observed and all decays were fitted with a single exponential decay. Details of the OH reactivity instrument have been added to the instrumental details section, and relevant citations are given. The total uncertainty in the ambient measurements of OH reactivity is ~ 6% (Stone et al. 2016). The new text describing the method is as follows:

"OH reactivity measurements were made using the laser flash photolysis pump-probe technique and the instrument is described in detail in Stone et al. (2016). Ambient air was drawn into the reaction cell (85 cm in length, 5 cm in diameter) at 12 SLM. Humidified ultra-high purity air (Messer, Air Grade Zero 2) passed a low-pressure Hg lamp at 0.5 SLM to generate ~ 50 ppbv of $O_3$ which was mixed with the ambient air. The $O_3$ was photolyzed at 266 nm to generate a uniform OH concentration across the reaction cell. The change in the OH radical concentration from pseudo-first-order loss with species present in ambient air was monitored by sampling the air from the reaction cell into a FAGE detection cell at ~1.5 Torr. The 308 nm probe laser (same as the FAGE laser describe above) was passed across the gas flow in the FAGE cell to excite OH radicals, and then detected the fluorescence signal at ~ 308

nm detected by a gated channel photomultiplier tube. The OH decay profile owing to reactions with species in ambient air was detected in real time. The decay profile was averaged for 5-mins and fitted with a first-order rate equation to find the rate coefficient describing the loss of OH ($k_{loss}$), with $k_{OH}$ determined by subtracting the physical loss of OH ($k_{phys}$). The OH reactivity data were fitted with a mono-exponential decay function as no bi-exponential behaviour was observed, even at the highest NO concentrations, and hence there was no evidence for recycling from $HO_2$ + NO impacting on the retrieved values The total uncertainty in the ambient measurements of OH reactivity is ~ 6% (Stone et al. 2016). "

**Page2 line46: ". . .quality are of serious concern. . ."**

This has been fixed

**Page2 line49: ". . . of the world fastest. . ."**

This has been fixed.

**Page2 line51: I would drop the number after the comma and round the percentages**

We agree and this has been done.

**Page2 line 59: NOx, SO2 and VOCs have not been defined**

We have now defined these, and added the following text:

"The reaction of OH with primary pollutant emissions (particularly $NO_x$ ($NO+NO_2$), $SO_2$ and VOCs (volatile organic carbon)) can form secondary pollutants such as $HNO_3$, $H_2SO_4$ and secondary oxygenated organic compounds (OVOCs)."

**Chapter 2.1 More information on the specific of the campaign site would be beneficial. Was the site on the street? On a platform? On the roof of the building? What was the distance between different instruments? I understand there is a specific paper on the topic but just two lines with a little bit information would suffice.**

We have added a brief description of the field site as follows:

"The instruments were housed in containers and located on the ground at the IAP site on a grassed area, the distance between the Leeds and York container (VOC and trace gas measurements) was ~3 m."

**Chapter 2.2.2 Here as well more details on the sensitivity towards the different RO2 is needed. The different concentrations of RO2 are used later on to justify some of the conclusions on the discrepancies between model and measurements so it is important to mention how well know is the separation in two classes of RO2 and which sensitivity is applied for which classes.**

A more detailed description of the ROxLIF instrument has been added which explains how the two different classes of $RO_2$ are measured and discusses what the sensitivity towards different $RO_2$ is and how this is determined.

The new text is as follows:

"The ROxLIF flow reactor (83 cm in length, 6.4 cm in diameter) was coupled to the second FAGE detection cell to allow for detection of $RO_2$ (total, complex and simple) using the method outlined by Fuchs et al. (2008). The flow reactor was held at ~30 Torr and drew ~7.5 SLM through a 1 mm pinhole ID (in-diameter). The flow reactor was operated in two mode: in the first ($HO_x$ mode) 125 sccm of CO (Messer, 10% in $N_2$) was mixed with ambient air close to the pinhole to convert OH to $HO_2$. In the second ($RO_x$ mode), 25 sccm of NO in $N_2$ (Messer, 500 ppmv) was also added to the CO flow to convert $RO_2$ into OH. The CO present during $RO_x$ mode rapidly converts the OH formed into $HO_2$. The air from

the RO$_x$LIF flow reactor was drawn (5 SLM) into the FAGE fluorescence cell (held at ~1.5 Torr) and NO (Messer, 99.9%) was injected into the fluorescence cell to convert HO$_2$ to OH. In HO$_x$ mode a measure of OH + HO$_2$ + cRO$_2$ (complex RO$_2$) was obtained; whilst RO$_x$ measured OH + HO$_2$ + ΣRO$_2$. sRO$_2$ (simple RO$_2$) concentration was determined by subtracting the concentration of cRO$_2$, HO$_2$ and OH from RO$_x$.

In previous laboratory experiments the sensitivity of the instrument to a range of different RO$_2$ was investigated and can be found in Whalley et al.(2018). Similar sensitivities were determined for a range of RO$_2$ species that were tested and agreed well with model-determined sensitivities. For comparison of the modelled RO$_2$ to the observed RO$_2$-total, RO$_2$-complex and RO$_2$-simple, the RO$_x$LIF instrument sensitivity towards each RO$_2$ species in the model was determined by running a model first under the RO$_x$LIF reactor and then the RO$_x$LIF FAGE cell conditions (NO concentrations and residence times) to determine the conversion efficiency of each modelled RO$_2$ species to HO$_2$. "

**Page8 line212: Is there really no difference between the accuracy of OH, HO2 and RO2 accounting that HO2 requires conversion into OH and RO2 requires a minimum of 2 NO steps?**

Although detection of OH is direct, detection of HO$_2$ is via conversion to OH via addition of NO, and RO$_2$ is via conversion to HO$_2$, and then the HO$_2$ is converted to OH in the FAGE (requiring two steps as the reviewer points out), because the instruments are calibrated separately using known concentrations of OH, HO$_2$ and RO$_2$, the accuracy of the measurement is the same as this depends on the calibration accuracy. The latter is controlled mainly by the accuracy in determining the product of the lamp intensity, the water vapour (in air) and photolysis time (which makes OH and HO$_2$), which is determined using chemical actinometry. Other factors such as absorption cross-sections, rate coefficients and quantum yields to make OH and HO$_2$, and the conversion efficiency of OH to the relevant RO$_2$ (which is quantitative) have very low uncertainties. In addition, the flow of NO is very reproducible.

**Page9 line 239: What is the concentration of H2 to 500 ppbv included in the model needed for?**

H$_2$ can react with OH and thus constitutes part of the OH reactivity, although a very minor contribution, and is also a source of HO$_2$. However, the inclusion of H$_2$ does not change the modelled reactivity or HO$_2$ (< 0.1%) much but is included in the model for completeness, as is normally the case in field studies of radicals and comparison with models.

**Page 9 line241: What was the time resolution of the GC data?**

The time resolution for the GC data was 1 hr and has been interpolated at 15 min intervals for the model. A sentence about this has been added to the paper as follows:

" The time resolution for the GC-FID data was 1 hr and has been interpolated to 15-min for the model input."

**Page 11 line290: Is the diel variation shown the mean or the median of the data?**

It is the median and this has been added to the caption of Figure 5.

"Comparison of the median average diel variation for j(O$^1$D) (s$^{-1}$), NO (ppbv), O$_3$ (ppbv), CO (ppbv), O$_x$ (ppbv), NO$_2$ (ppbv), HONO (ppbv) and boundary layer height (m) inside and outside haze events; denoted by solid red and blue lines, respectively. The dashed lines represent the interquartile range for the respective species and pollution period."

**Page 11 line300: O3 does not react with high levels of NO but with a high concentration of NO**

Thanks, this has been fixed.

**Page 21 Section 4.1: I assume that here only the results from the model are shown but this is not clear from reading the text.**

Yes, only the results from the model are shown here, and text to make this clear has been added to caption of Figure 9.

"Figure 9. Rates of primary production (top panel) and termination (bottom panel) for $RO_x$ radicals (defined as OH + $HO_2$ + RO + $RO_2$) calculated for MCM-base model separated into haze (right) and non-haze (left) periods. The definition of haze is when $PM_{2.5}$ exceeds 75 $\mu m^{-3}$. The production from: $O^1D$ + $H_2O$ and VOC + $NO3$ and the termination reactions: $RO_2$ + $HO_2$, $HO_2$ + $HO_2$, $HO_2$ + $NO_2$, are shown in the key, although many are not visible and contributed <1% of the total production and termination."

**Page 24 Lines516-521: Has the possibility of segregation of air been investigated and ruled out or why this is mentioned here but there is no discussion on how this could have had an impact on this specific site? It could be worth discussing if this could help bringing measurements and model results in agreement.**

As noted in the response to reviewer 1, various NO measurements were made at ground level around the site via multiple instruments and which might have pointed to any segregation of NO owing to local point source. A sentence has been added to paper as follows:

"There were several instruments for NO measurements located around the site and no differences in concentrations were observed, hence no evidence of any obvious segregation."

**Page 24 line539: Assuming that figure 10 is actually figure 11 (where in the caption of the figure the model line is the red one (?)), I do not agree with the statement in the paper that the model can reproduce the simple RO2 measured for NO above 100 ppbv. Actually, there is overlap between the model and the measured RO2 95th percentile for the complex RO2. In all honesty, I am not sure this plot tells us much as the model equally predicts pretty much zero RO2 expected at NO above 10 ppbv for both type of RO2. Although I agree that the simple RO2 have been studied more carefully, what would be the difference in rate with NO to justify the observed concentration of RO2 or what type of different chemistry for the most complex RO2 would be needed? There is no discussion in this study about it and some suggestions of what is feasible are needed.**

We apologise, Figure 10 is indeed Figure 11 and this has been amended in the text.

Please see the response to Reviewer 1 regarding the model behaviour for the various types of $RO_2$ at high NO, where there is an amended statement that the model could reproduce simple $RO_2$ at high NO.

This plot (Figure. 11) is important as it shows that the missing source of $RO_2$ must form both complex and simple $RO_2$, as the underprediction of both increases with increasing NO.

The effect on decreasing the kRO2 + NO has been investigated and shows that decreasing the rate constant by a factor ~ 10 cannot reconcile the modelled $RO_2$ with the measured at high NO (still underpredicted by a factor of 10). Also, whilst the modelled $RO_2$ is improved by decreasing the rate constant, the increased $RO_2$ in the model is not recycled into $HO_2$ and OH and the model underpredictions for these radical species remains. A discussion for these results has been added into the supplementary in section S1.7, and is as follows:

"**S1.7 The effects of the kRO2 + NO rate constant on the modelled radical species**

Other than $CH_3O_2$ and $C_2H_5O_2$, rate constants for the reaction of many other $RO_2$ + NO is based on structure activity relationships (SARs) in the MCM and is lumped to kRO2NO and kAPNO (http://mcm.leeds.ac.uk/MCM/). The lumped rate constants kRO2NO and kAPNO were both decreased by a factor of 2 and 10 to investigate the effects on modelled OH, $HO_2$ and $RO_2$. The model where the rate constant for $RO_2$ + NO was decreased by a factor of 2 is titled MCM-kRO2-2, whilst the model where the rate constant was decreased by a factor of 10 is titled MCM-kRO2NO-10.

The comparison of measured values with modelled values (MCM-base, MCM-kRO2-2 and MCM-kRO2-10) is shown in Figure S8. Figure S8 shows that on certain days (e.g. 19/11, 5/12 and 9/12) when the model (MCM-base) could not reproduce the measured values of $RO_2$ the discrepancy between the measurements and the MCM-kRO2NO-10 model is almost reconciled. On these days the MCM-kRO2NO-10 does not really change the OH or $HO_2$ concentration from the base model. On all days the MCM-base underpredicts the $RO_2$ concentration, and MCM-kRO2NO-10 does decrease the gap between measurements and modelled, compared to MCM-base. MCM-kRO2NO-2 does not significantly increase the total $RO_2$ concentration from MCM-base, unlike MCM-kRO2NO-10. Since changing the rates of $RO_2$ + NO will be very dependent on the NO concentration, the ratio of measured:modelled radical concentration has been binned against the log of NO for MCM-base, MCM-kRO2NO-2 and MCM-kRO2NO-10 in Figure S9. Figure S9 shows similar results to the timeseries where at the lower concentration of NO (19/11, 5/12 and 9/12) the MCM-kRO2NO-10 can reproduce the $RO_2$ concentration. The results at higher [NO] show that decreasing the rate of $RO_2$ + NO improves the agreement between measured:modelled $RO_2$, especially for MCM-kRO2NO-10, but the observed $RO_2$ concentration is still underpredicted beyond 30 ppbv.

[Figure]

**Figure S8.** (a) Time-series comparison of measured values of OH with modelled OH concentrations from MCM-base, MCM-kRO23NO-2 and MCM-kRO2-10. (b) Time-series comparison of measured values of $HO_2$ with modelled $HO_2$ concentrations from MCM-base, MCM-kRO23NO-2 and MCM-kRO2-10. (c) Time-series comparison of measured values of total $RO_2$ with modelled total $RO_2$ concentrations from MCM-base, MCM-kRO23NO-2 and MCM-kRO2-10. The data sets are 15-minutes averaged.

The fact that the OH and $HO_2$ modelled concentrations do not change significantly for the models with reduced $RO_2$ + NO rate constant highlights that the enhanced $RO_2$ radicals (in MCM-kRO2-10) are not recycling into $HO_2$ or OH, even though the agreement for the $RO_2$ concentration is improved for these models (MCM-kRO2NO-2 and MCM-krO2NO-10). The lack of $RO_2$ recycling highlights that the $RO_2$ and RO radicals are terminating rather than propagating in the model.

This work highlights alternative chemistry and solutions must be applied for the two different NO regimes observed during the Beijing wintertime campaign. At high [NO] (above 10 ppbv) further reductions in the $RO_2$+NO rate constant would be required to reconcile the model with observations. However, at NO mixing ratios below 10 ppbv, further reductions in the $RO_2$+NO rate constant would lead to the model overpredicting the $RO_2$ concentration.

[Figure]

**Figure S9.** The ratio of measurement/model for OH (a), $HO_2$ (b) and $RO_2$ (c) across various NO concentrations for daytime values only ($j(O^1D) > 1 \times 10^{-6}$ s$^{-1}$). Light blue represents for results from MCM-kRO2NO-2, dark blue represents results from MCM-base and red represents results from MCM-kRO2NO-10.

"

A reference in the paper to this section has been added "The effect on reducing the $RO_2$ has been investigated and is shown in S1.7 in the supplementary material. The results show that reducing the rate constant by a factor ~10 does improved the modelled to measurements agreement by a factor of 8.3 for total $RO_2$. However, $RO_2$ is still underpredicted by a factor of ~12 at the highest NO. Also the increased $RO_2$ in the model does not recycle into $HO_2$ or OH efficiently. This work highlights that uncertainties in the rate constant for $RO_2$ + NO for different $RO_2$ cannot be the only explanation for the underprediction of $RO_2$ in the model."

**Page 27 line 570-573: What would be the concentration of CL2 and/or ClNO2 needed to justify such a production of RO2? This could tell us if it could be possible at all.**

Unfortunately, there were no $ClNO_2$ measurements during the winter campaign, and hence it was not possible to calculate a time series for Cl atoms formed from photolysis of $ClNO_2$ and to assess any

additional RO$_2$ radicals generated. Using the model run where an additional RO$_2$ source was added to reconcile the measurements and the model a rough calculation has shown that the ClNO$_2$ concentration would have to be of the order of ~5800 ppbv in order to close the gap between modelled and measured RO$_2$. Previous measurements of ClNO$_2$ in suburban Beijing has shown a peak of ~2.9 ppbv (Wang et al. 2017) which is ~3 orders of magnitude smaller than the ClNO$_2$ concentration required, suggesting other additional primary sources are needed in the model besides Cl chemistry.

Added statement to paper about chlorine chemistry (page 28, line 615 – 619): "Although the ClNO$_2$ concentration required to bridge the gap between model and measurements would be ~5800 ppbv on average (see supplementary section S1.8 for details). Previous measurements in China in suburban Beijing have shown ClNO$_2$ peaking at 2.9 ppbv (Wang et al. 2017), however, and suggests other additional primary source are needed in the model besides Cl chemistry."

This section has been added into the supplememtary material:

"

**S1.8 ClNO$_2$ and Cl concentration required to bridge the gap between measured and modelled total RO$_2$**

Unfortunately, there were no ClNO$_2$ measurements during the winter campaign, and hence it was not possible to calculate a time series for Cl atoms formed from photolysis of ClNO$_2$ and to assess any additional RO$_2$ radicals generated. Using the model run where additional RO$_2$ source was added to reconcile the measurements and the model a rough calculation has shown that the ClNO$_2$ concentration would have to be on average ~5800 ppbv in order to close the gap between modelled and measured RO$_2$. Figure S10 shows the average diel of the calculated ClNO$_2$ and Cl concentration with peak at 1.4 x 10$^4$ ppbv and 1.6 x 10$^6$ molecule cm$^{-3}$, respectively. The ClNO$_2$ and Cl concentration have been calculated using SE3 – SE5:

$$\text{P}'\text{RO}_2 = k_{\text{VOC+Cl}}[\text{VOC}][\text{Cl}] \qquad \text{S E3}$$

$$[\text{Cl}] = \frac{\text{P}'\text{RO}_2}{k_{\text{VOC+Cl}}[\text{VOC}][\text{Cl}]} \qquad \text{S E4}$$

$$[\text{ClNO}_2] = \frac{k_{\text{VOC+Cl}}[\text{VOC}][\text{Cl}]}{j\text{ClNO}_2} \qquad \text{S E5}$$

where $k_{\text{VOC+Cl}}$ is a generic rate constant that represents the reaction of all VOCs with Cl which in this case is 4 x 10$^{-12}$ molecule$^{-1}$ cm$^3$ s$^{-1}$, [VOC] is the sum of the measured VOC concentration for the campaign and P'RO$_2$ is the calculated additional RO$_2$ used in MCM-PRO2 (see main paper section 4.2 for more details). The ClNO$_2$ required to bridge the gap between measured and modelled of RO$_2$ is ~3 orders of magnitude greater than the peak ClNO$_2$ concentration measured in suburban Beijing (2.9 ppbv) by Wang et al. (2018) suggesting that other additional primary source are needed in the model besides Cl chemistry.

[Figure]

**Figure S10** Average diel of the ClNO$_2$ and Cl atom concentration required to bridge the gap between measured and modelled RO$_2$. The ClNO$_2$ and Cl concentrations have been calculated from the additional primary source of RO$_2$ added to the MCM-PRO2 model run, see section 4.2 in the main paper for more details.

"

Wang, X., Wang, H., Xue, L., Wang, T., Wang, L., Gu, R., Wang, W., Tham, Y.J., Wang, Z., Yang, L. and Chen, J., 2017. Observations of N2O5 and ClNO2 at a polluted urban surface site in North China: High N2O5 uptake coefficients and low ClNO2 product yields. *Atmospheric environment*, *156*, pp.125-134.

**Reviewer 3.**

We thank the reviewer for their careful reading of the manuscript. We address each of the comments in turn below, with the comments first given in bold, followed by the response in normal type, followed by any changes made to the manuscript.

**While the title and main conclusions of the paper refer to wintertime haze events, the main modeling of the results summarized in Figure 8 appears to include both haze and non-haze events, while the brief discussion in section 4.3 separates the model analysis to haze and non-haze events, with Figure 14 showing the base model agreement worse under haze events. While the model appears to underestimate the measured RO2 concentration similarly for both events, the agreement of the predicted OH and HO2 concentrations with the measurements is better for the non-haze events. It appears from Figure 7 that the number of haze and non-haze events were roughly equal. As a result, it is not clear whether some of the main conclusions of the paper would be applicable to the haze events. It would be useful to illustrate in Figure 6, 8, and 13 how the different models in Table 1 are able to reproduce the radical measurements for haze and non-haze events. Is the estimation of the missing source different for the haze and non-haze events? Are the model results/conclusions different for the different events? While they may not be significant, any differences between the events should be discussed in more detail.**

The reviewer makes a good point and we have now updated some figures to include separate comparisons for haze and non-haze events and introduced a more detailed discussion. In Figure 8 we do include diel profiles for both haze and non-haze events, and we have now updated Figure 14 to include modelling results from MCM-base, MCM-cHO2 as well as the measured values (including speciated $RO_2$ concentrations) for both haze and non-haze events, and we have included an additional discussion on the differences in model performance for the haze and non-haze periods for each of these species. The modified text is as follows:

"The measured complex $RO_2$ radical species peak at similar concentration inside ($4.3 \times 10^7$ molecule $cm^{-3}$) and outside ($4.6 \times 10^7$ molecule $cm^{-3}$) of haze. Unlike the complex $RO_2$, the simple $RO_2$ concentration peaks at a lower concentration inside of haze ($3.4 \times 10^7$ molecule $cm^{-3}$) compared with outside ($5.5 \times 10^7$ molecule $cm^{-3}$). The complex $RO_2$ is undepredicted by a factor of ~48 and ~12 inside and outside of haze, respectively. Whilst the simple $RO_2$ is undepredicted by a factor of ~66 and ~5.7 inside and outside of haze, respectively. The sharp increase for the underprediction of both simple and complex $RO_2$ inside of haze highlights the need of a large additional source of both simple and complex $RO_2$, especially under haze conditions. The increased contribution to kOH ($s^{-1}$) from VOCs from non-haze to haze conditions is a factor of: ~10 for aromatics, ~8 for alkenes and alkynes, ~6 for alkanes, ~9 for alcohols and ~2 for aldehydes. The large increase in relative contribution to kOH from aromatics, alkenes and alkynes is consistent with observation of higher complex $RO_2$ (compared to simple $RO_2$) during haze periods compared to non-haze periods. " and the updated Figure 14 is shown below:

[Figure]

**Figure 14.** Average diel profiles for measured and modelled OH, $HO_2$, total $RO_2$, complex $RO_2$ ($RO_2$ comp), simple $RO_2$ ($RO_2$ simp) and kOH separated into haze (right) and non-haze (left) periods.

Figure 6, which shows the results of the photo-stationary state (PSS) expression for OH together with measurements of OH, has also been updated to include separation into haze and non-haze events. The PSS has been separated into haze and non-haze events and shows that during haze events the PSS captures the OH concentration, although the PSS does overpredict the OH concentration by ~1.35 between 09:30 – 14:30 in haze events. The overprediction by the PSS in haze events is highly influenced by the overprediction on the 04/12/2016. Whilst under non-haze conditions the PSS captures the OH concentration very well throughout the day. The production of from HONO increases in non-haze (~19%) compared with haze events (~7%). The updated text is as follows:

"The PSS has been separated into haze and non-haze events and shows that during haze events the PSS captures the OH concentration, although the PSS does overpredict the OH concentration by ~1.35 between 09:30 – 14:30 in haze events. However, the overprediction by the PSS in haze events is highly influenced by the overprediction on the 04/12/2016. Whilst under non-haze conditions the PSS captures the OH concentration very well throughout the day. The production of from HONO increases in non-haze (~19%) compared with haze events (~7%). " and the updated Figure 6 is shown below:

[Figure]

**Figure 6.** Average diel profile for observed and steady state calculated OH concentrations for: (a) non-haze, and(b) haze periods. Panel (c) shows a comparison time-series for the steady state calculation of OH and measured OH. The OH generated by $O^1D+H_2O$, although included in the key, is too small to be visible.

Figure 12 which shows the additional primary production required to bridge the gap between measured and modelled $RO_2$ and Figure 13 have been merged in an effort to shorten the manuscript. The merged graph (now Figure.12) has been separated into haze and non-haze events. The P'$RO_2$ is higher in the updated version of Figure 12 (see below) as the original Figure 12 had not been filtered for when measured data was available.

Figure 13 is now separated into haze and non-haze events too. A discussion has been added for the new graph.

Modified text :" The additional primary production of $RO_x$ (P'$RO_x$) radicals required to bridge the gap between measured and modelled total $RO_2$ was found to peak at an average of 3.5 x$10^8$ molecule cm$^{-3}$ s$^{-1}$ at 08:30 non-haze events. Under haze conditions, the gap between measured and modelled total $RO_2$ was found to peak at an average of 4 x $10^8$ molecule cm$^{-3}$ s$^{-1}$ at 13:30 as shown in Figure 12, calculated from Eq. 3 (Tan et al., 2018):

$$P'(RO_x) = k_{HO2+NO}\,[HO_2]\,[NO\ ] - P(HO_2)_{prim} - P(RO_2)_{prim} - k_{VOC}[OH] \qquad \text{Eq. 3}$$
$$+ L(HO_2)_{term} + L(RO_2)_{term}$$

where P(HO$_2$)$_{prim}$, P(RO$_2$)$_{prim}$, L(HO$_2$)$_{term}$ and L(RO$_2$)$_{term}$ are the rates of primary production of HO$_2$, primary production of RO$_2$, termination of HO$_2$ and termination of RO$_2$, respectively. The overall (haze and non-haze) additional primary production peak of ~44 ppbv hr$^{-1}$ (at 10:30 ) is almost nine times

larger than the additional $RO_2$ source that was required to resolve the measured and modelled $RO_2$ during the BEST-ONE campaign (5 ppbv h$^{-1}$ during polluted periods, also calculated using Eq. 3), and is much larger compared to the known noon-average modelled primary production of $RO_x$ during the APHH campaign of 1.7 ppbv hr$^{-1}$. The additional primary production required in non-haze rises sharply in the morning peaking at 08:30 (3.5 x 10$^8$ molecule cm$^{-3}$) and then decreases rapidly; whilst the additional source needed in haze events peaks at 4 x 10$^8$ molecule cm$^{-3}$ s$^{-1}$. The additional primary source required during haze events through-out the day is ~7 times higher than that during non-haze events."

Modified text: "However, the MCM-PRO2 run overpredicts the observed $HO_2$ during haze and non-haze events by a factor of 3.4 and 2.5, respectively, with the large overprediction of $HO_2$ in haze and non-haze events driving the overprediction of OH by a factor of 2.2 and 2.5. This highlights that the additional primary $RO_2$ source may be an $RO_2$ species that does not readily propagate to $HO_2$, this has also been discussed in Whalley et al. (2020). "

Modified text: "The comparison of MCM-PRO2-SA with both measurements and MCM-PRO2 (see Table 2 for details) is shown in Figure 12 and shows that the uptake of $HO_2$ only has a small impact <6% and <14% on the modelled levels of OH, $HO_2$ and $RO_2$ during haze and non-haze events, respectively."

Updated Figure 13.

[Figure]

**Figure 13.** Average diel comparison of measurements of P'RO$_2$, OH, HO$_2$ and sum of RO$_2$ with the MCM-base, MCM-PRO2 and MCM-PRO2-SA box-model runs inside (e – h) and outside (a – d) of haze events. The average diel is from the entire APHH winter campaign. See text and Table 2 for definitions of each of the model runs.

Lisa K Whalley, Eloise J Slater, Robert Woodward-Massey, Chunxiang Ye, James D Lee, Freya Squires, James R Hopkins, Rachel E Dunmore, Marvin Shaw, Jacqueline F Hamilton, Alastair C Lewis, Archit Mehra, Stephen D Worrall, Asan Bacak, Thomas J Bannan, Hugh Coe, Bin Ouyang, Roderic L Jones, Leigh R Crilley, Louisa J Kramer, William J Bloss, Tuan Vu, Simone Kotthaus, Sue Grimmond, Yele Sun, Weiqi Xu, Siyao Yue, Lujie Ren, W. Joe F Acton, C. Nicholas Hewitt, Xinming Wang, Pingqing Fu, and Dwayne E Heard : Evaluating the sensitivity of radical chemistry and ozone formation to ambient VOCs and NOx in Beijing, *Atmos. Chem. Phys Disc*, 2020.

**2) The authors should clarify their definition of OHwave and OHchem on pages 6-7. The current description suggests that OHchem is the on-line background measurement including interferences, while OHwave is the off-line background measurement. However, Figure 3 compares the measured OH concentration determined using chemical modulation (signal – OHchem background) with that determined by spectral modulation (signal – OHwave background), not a comparison of the background signal measured by both methods.**

The definition of OHwave and OHchem and the discussion surrounding Figure 3 has been tightened up in the paper, and the modified sections of text are as follows:

Modified text

"OHchem is the online OH signal – OHchem background and OHwave is the OH online signal – Ohwave background."

**3) Related to this, the authors state that the spectral modulation measurements were also corrected for laser-generated OH from ozone photolysis + H2O (page 7). Based on the Woodward-Massey et al. (2020) paper, it appears that the interference was calculated based on laboratory measurements of the interference as a function of ozone, water and laser power. This should be clarified. Since this interference would be measured by chemical modulation, a comparison of the measured interference with that calculated would provide additional confidence in the OHChem measurement as well as the accuracy of the interference estimate.**

OHwave data were indeed corrected for the known interference from O$_3$ + H$_2$O, with further details available from Woodward-Massey et al. (2020). The O$_3$ + H$_2$O interference calculated was very small (median ~8.5 x 10$^3$ molecule cm$^{-3}$) due to the low concentrations of H$_2$O and O$_3$.

Modified text: " OHwave data were corrected for the known interference from O$_3$ + H$_2$O, see Woodward-Massey et al. (2020) for further details. The O$_3$ + H$_2$O interference calculated was very small (median ~8.5 x 10$^3$ molecule cm$^{-3}$) due to the low concentration of H$_2$O and O$_3$. All figures and calculation from now on have used OHwave as it is the most extensive time-series (12 days compared to 5 days)."

Woodward-Massey, R., Slater, E. J., Alen, J., Ingham, T., Cryer, D. R., Stimpson, L. M., Ye, C., Seakins, P. W., Whalley, L. K., and Heard, D. E.: Implementation of a chemical background method for atmospheric OH measurements by laser-induced fluorescence: characterisation and observations from the UK and China, Atmos. Meas. Tech., 13, 3119–3146, https://doi.org/10.5194/amt-13-3119-2020, 2020.

**4) There is little discussion of the HO2, HO2*, and RO2 experimental measurement conditions, except that it appears that the conditions were similar to that in the ClearfLo study. The paper would**

**benefit from a brief discussion of the experimental conditions employed in this study. It appears that only a single NO flow was used in the HOx detection cell for these measurements, in contrast to the use of two NO flows used to measure HO2 and HO2\* (RO2i) during ClearfLo (Whalley et al., 2018). Instead it appears that HO2\* was measured using the ROxLIF detection cell. While it is stated that the ROxLIF method is described "in detail below" (page 5), the paper again references Whalley et al. (2018) instead of providing details. Given the high concentrations of NOx in this study, how did the authors account for potential interferences from the decomposition of HO2NO2 and CH3O2NO2? More details on the experimental measurements are needed. In addition the authors should clarify how the simple RO2 and complex RO2 were derived from the measurements. It appears that complex RO2 was obtained from the difference between the HO2\* ROxLIF measurements and the FAGE HO2 measurements, while the simple RO2 were obtained from the difference between the ROxLIF RO2 and HO2\*measurements. Much of this information could go into the Supplement.**

- Description of the $RO_x$LIF instrument and the running conditions has been added to the paper. The text is as follows:

"The ROxLIF flow reactor (83 cm in length, 6.4 cm in diameter) was coupled to the second FAGE detection cell to allow for detection of $RO_2$ (total, complex and simple) using the method outlined by Fuchs et al. (2008). The flow reactor was held at ~30 Torr and drew ~7.5 SLM through a 1 mm pinhole ID (in-diameter).  The flow reactor was operated in two mode: in the first ($HO_x$ mode) 125 sccm of CO (Messer, 10% in $N_2$) was mixed with ambient air close to the pinhole to convert OH to $HO_2$. In the second ($RO_x$ mode), 25 sccm of NO in $N_2$ (Messer, 500 ppmv) was also added to the CO flow to convert $RO_2$ into OH. The CO present during $RO_x$ mode rapidly converts the OH formed into $HO_2$. The air from the $RO_x$LIF flow reactor was drawn (5 SLM) into the FAGE fluorescence cell (held at ~1.5 Torr) and NO (Messer, 99.9%) was injected into the fluorescence cell to convert $HO_2$ to OH.  In $HO_x$ mode a measure of OH + $HO_2$ + $cRO_2$ was obtained; whilst $RO_x$ measured OH + $HO_2$ + $\Sigma RO_2$. $sRO_2$ concentration was determined by subtracting the concentration of $cRO_2$, $HO_2$ and OH from $RO_x$.

In previous laboratory experiments the sensitivity of the instrument to a range of different $RO_2$ was investigated and can be found in Whalley et al.(2018). Similar sensitivities were determined for a range of $RO_2$ species that were tested and agreed well with model-determined sensitivities. For comparison of the modelled $RO_2$ to the observed $RO_2$-total, $RO_2$-complex and $RO_2$-simple, the $RO_x$LIF instrument sensitivity towards each $RO_2$ species in the model was determined by running a model first under the $RO_x$LIF reactor and then the $RO_x$LIF FAGE cell conditions (NO concentrations and residence times) to determine the conversion efficiency of each modelled $RO_2$ species to $HO_2$. "- The values of $RO_2$ (simple, complex and total) in the paper have not been corrected for the decomposition of $HO_2NO_2$ and $CH_3O_2NO_2$ but an estimation has been added to the supplementary material and shows the correction from the decomposition of $HO_2NO_2$ and $CH_3O_2NO_2$ is ~6 %, ~8 % and 4 % for total, complex and simple $RO_2$, respectively.

Signposting text has been added in the main paper to the supplementary material discussion for $HO_2NO_2$ and $CH_3O_2NO_2$ decomposition. Modified text: "The potential interference in the $RO_2$ measurements from $HO_2NO_2$ and $CH_3O_2NO_2$ has been explored in the supplementary material in section S1.4, however the data presented through-out the paper are the uncorrected data since the correction is small (correction from the decomposition of $HO_2NO_2$ and $CH_3O_2NO_2$ is ~6 %, ~8 % and 4 % for total, complex and simple $RO_2$, respectively.)"

Information added to the supplementary material:

**"S1.4 Estimating the contribution of $HO_2NO_2$ and $CH_3O_2NO_2$ to the $RO_2$ signal**

In the main paper we do not apply a correction for a possible contribution of pernitric acid (PNA, $HO_2NO_2$) and methyl peroxy nitric acid (MPNA, $CH_3O_2NO_2$). The MPNA decomposition will contribute

to the simple $RO_2$ and total $RO_2$ whilst the PNA contributes to the complex and total $RO_2$ measurements. The concentration of $HO_2NO_2$ and $CH_3O_2NO_2$ was modelled using the MCM-base model, then in agreement with the work by Fuchs et al.(2008) 0.43 % and 9 % of the $HO_2NO_2$ and $CH_3O_2NO_2$ is calculated to decompose and contribute to the $RO_2$ signal. The rate of decomposition in the Julich and Leeds $RO_xLIF$ reactors is expected since the design and residence time (~1 second) are similar. The comparison of the measured total, simple and complex $RO_2$ with the corrected values is shown in Figure S5. Figure S5 shows that the correction from the decomposition of $HO_2NO_2$ and $CH_3O_2NO_2$ is ~6 %, ~8 % and 4 % for total, complex and simple $RO_2$, respectively.

[Figure]

**Figure S5** a) Timeseries comparison for measured total $RO_2$ (blue) and total $RO_2$ corrected (black) for the decomposition from $HO_2NO_2$ and $CH_3O_2NO_2$. b) Timeseries comparison for measured complex $RO_2$ (blue) and complex $RO_2$ corrected (black) for the decomposition from $HO_2NO_2$. c) Timeseries comparison for measured simple $RO_2$ (blue) and simple $RO_2$ corrected (black) for the decomposition from $CH_3O_2NO_2$.

"

Fuchs, H., Holland, F. and Hofzumahaus, A., 2008. Measurement of tropospheric $RO_2$ and $HO_2$ radicals by a laser-induced fluorescence instrument. *Review of Scientific Instruments*, *79*(8), p.084104.

Whalley, L. K., Stone, D., Dunmore, R., Hamilton, J., Hopkins, J. R., Lee, J. D., Lewis, A. C., Williams, P., Kleffmann, J., and Laufs, S.: Understanding in situ ozone production in the summertime through radical observations and modelling studies during the Clean air for London project (ClearfLo), Atmospheric Chemistry and Physics, 18, 2547-2571, 2018.

**5) Similarly, there is no discussion of the experimental method used to measure total OH reactivity. From the information given in Figure 7, it appeared that the OH reactivity was calculated based on**

**the measured OH sinks, but it is clear from Figure 8 that total OH reactivity was measured. Is the measured OH reactivity shown in Figure 7?**

Measured OH reactivity is shown in Figure 7 and represented by the black line, and the caption has been updated to make this clear:

Updated Figure 7. caption :" Time-series of OH, b) $HO_2$, c) total $RO_2$, d) partly-speciated $RO_2$ and e) Measured (black) and modelled (stacked plot) OH Reactivity. For (a)-(c), the raw measurements (6-min data acquisition cycle) are blue open circles with 15 min average represented by the solid blue line. The 15 min model output in a-c is represented by the red line for OH, $HO_2$ and $RO_2$. The partly-speciated $RO_2$ is separated into simple (gold open circles) and complex (purple open circles). The individual contributions of the model to the OH reactivity is given below the graph. The grey shaded areas show the haze periods when $PM_{2.5} > 75 \, \mu g \, m^{-3}$."

A section has also been added describing the OH reactivity method, modified text is shown below copied from response to reviewer 2: "OH reactivity measurements were made using the laser flash photolysis pump-probe technique and the instrument is described in detail in Stone et al. (2016). Ambient air was drawn into the reaction cell (85 cm in length, 5 cm in diameter) at 12 SLM. Humidified ultra-high purity air (Messer, Air Grade Zero 2) passed a low-pressure Hg lamp at 0.5 SLM to generated ~ 50 ppbv of $O_3$ which was mixed with the ambient air. The $O_3$ was photolyzed at 266 nm to generate a uniform OH concentration across the reaction cell. The change in the OH radical concentration from pseudo-first-order loss with species present in ambient air was monitored by sampling the air from the reaction cell into a FAGE detection cell at ~1.5 Torr. The 308 nm probe laser (same as the FAGE laser describe above) was passed across the gas flow in the FAGE cell to excite OH radicals, and then detected the fluorescence signal at ~ 308 nm detected by a gated channel photomultiplier tube. The OH decay profile owing to reactions with species in ambient air was detected in real time. The decay profile was averaged for 5-mins and fitted with a first-order rate equation to find the rate coefficient describing the loss of OH ($k_{loss}$), with $k_{OH}$ determined by subtracting the physical loss of OH ($k_{phys}$). The OH reactivity data were fitted with a mono-exponentially decaying function as no bi-exponential behaviour was observed, even at the highest NO concentrations, and hence there is no evidence for recycling from $HO_2$ + NO impacting on the retrieved values The total uncertainty in the ambient measurements of OH reactivity is ~ 6% (Stone et al. 2016). "

Stone, D., Whalley, L.K., Ingham, T., Edwards, P., Cryer, D.R., Brumby, C.A., Seakins, P.W. and Heard, D.E., 2016. Measurement of OH reactivity by laser flash photolysis coupled with laser-induced fluorescence spectroscopy. *Atmospheric Measurement Techniques*, pp.2827-2844.

**brief description of the measurement technique should be included. Given the high mixing ratios of NO that were observed, did interference from the HO2+NO reaction impact the OH reactivity measurements?**

The kOH decays show no biexponential behaviour suggesting that recycling from $HO_2$ + NO was not observed and all decays were fitted with a single exponential decay. Details of the OH reactivity instrument have been added to the instrumental details section, and relevant citations are given. The total uncertainty in the ambient measurements of OH reactivity is ~ 6% (Stone et al. 2016). The new text describing the method is as follows:

"OH reactivity measurements were made using the laser flash photolysis pump-probe technique and the instrument is described in detail in Stone et al. (2016). Ambient air was drawn into the reaction cell (85 cm in length, 5 cm in diameter) at 12 SLM. Humidified ultra-high purity air (Messer, Air Grade Zero 2) passed a low-pressure Hg lamp at 0.5 SLM to generated ~ 50 ppbv of $O_3$ which was mixed with the ambient air. The $O_3$ was photolyzed at 266 nm to generate a uniform OH concentration across the reaction cell. The change in the OH radical concentration from pseudo-first-order loss with species present in ambient air was monitored by sampling the air from the reaction cell into a FAGE detection

cell at ~1.5 Torr. The 308 nm probe laser (same as the FAGE laser describe above) was passed across the gas flow in the FAGE cell to excite OH radicals, and then detected the fluorescence signal at ~ 308 nm detected by a gated channel photomultiplier tube. The OH decay profile owing to reactions with species in ambient air was detected in real time. The decay profile was averaged for 5-mins and fitted with a first-order rate equation to find the rate coefficient describing the loss of OH ($k_{loss}$), with $k_{OH}$ determined by subtracting the physical loss of OH ($k_{phys}$). The OH reactivity data were fitted with a mono-exponentially decaying function as no bi-exponential behaviour was observed, even at the highest NO concentrations, and hence there is no evidence for recycling from $HO_2$ + NO impacting on the retrieved values The total uncertainty in the ambient measurements of OH reactivity is ~ 6% (Stone et al. 2016). ."

**Abstract: There have been previous measurements of radicals at similar NO levels in Mexico City (Shirley et al., ACP, 2006; Dusanter et al., ACP, 2009).**

We thank for the reviewer for pointing this out. The abstract has been corrected as follows:

"Wintertime *in situ* measurements of OH, $HO_2$ and $RO_2$ radicals and OH reactivity were made in central Beijing during November and December 2016. Exceptionally elevated NO was observed on occasions, up to ~250 ppbv."

**The caption in Figure 3 states that the gray points represent an acquisition cycle of 6 min, but the legend states that they are 4 min averages.**

The average stated in the legend is for the OH measurement period only, while the overall data acquisition is for the whole measurement period (including 2 minutes of $HO_2$ measurements).

The caption has been updated, and now reads as follows:

"Overall intercomparison of OHwave and OHchem observations from the winter 2016 APHH campaign. Grey markers represent raw data (6 min acquisition cycle, 4 minutes and 2 minutes for the OH and $HO_2$ measurements), with 1 h averages (±2 standard error, SE) in red. The thick red line is the orthogonal distance regression (ODR) fit to the hourly data, with its 95% confidence interval (CI) bands given by the thin red lines; fit errors given at the 2σ level. For comparison, 1:1 agreement is denoted by the blue dashed line. OHwave data were corrected for the known interference from $O_3$ + $H_2O$. Taken from (Woodward-Massey et al., 2020) where further details can be found."

**While the VOC measurements used to constrain their model are given in Table 1, the paper would benefit from additional information on the instruments used to measure the other model constraints. Even though this information may be provided in a separate campaign paper, a table similar to that in Whalley et al. (2018) could be included in the Supplement.**

This has been covered in response to reviewer 2, and the response to reviewer 2 is copied below:

We have added a Table (Table 2) which describes the methods used for some of the key species which are used to constrain the model. For many of the other species used to constrain the model, details are given in Shi et al 2018, and we have made a clear reference to that paper.

Modified wording "The accuracy and precision of trace gas species can be found in Table 2, details on the HONO measurements used in the modelling scenarios can be found in Crilley et al.(2019). Details for other measurements can be found in Shi et al.(2018)"

The following table has been added to the manuscript:

| Instrument | Technique | 2σ Uncertainty / % | 2σ Precision/ ppbv |
|---|---|---|---|
| O₃, TEi49i | UV absorption | 4.04 | 0.28[1] |
| NO, TEi42i-TL | Chemiluminescence via reaction with O₃ | 4.58 | 0.03[1] |
| SO₂, TEi43i | UV fluorescence | 3.12 | 0.03[1] |
| NO₂, CAPS, T500U | Cavity enhanced absorption spectroscopy | 5.72 | 0.04[1] |
| HONO | LOPAP x2, BBCEAS x 2, ToF-CIMS and SIFT-MS | 9 – 22% | 0.025 – 0.130 |

**Table 2.** Instruments and techniques used to measure key model constraints. 2σ uncertainties for the measured trace gas species used in the modelling scenarios are quoted. [1]Precision is given for 15-minute averaging time. For details of the HONO measurements please see Crilley et al.(2019).

**It appears from Figure 4 that HONO measurements were not available between 2/12 and 5/12, but the steady-state calculations shown in Figure 6 include data between 2/12-8/12 and were chosen "as full data coverage for HONO, NO, j values, radical and k(OH) measurements were available." Was HONO available on all these days?**

This has been covered in response to reviewer 1, the response to reviewer 1 is copied below:

The HONO dataset shown in Figure 4 was from one HONO instrument only and the HONO used in the steady-state calculation was the HONO recommended by Crilley et al. (2019) based on measurements by several instruments during the campaign, and represents a more complete dataset. The HONO shown in Figure 4 has now been updated to those recommended by Crilley et al. (2019), and are the values that have been used in the steady-state calculation and MCM model. Low $NO_x$ would lead to reduction in recycling from $HO_2 + NO$, which is the largest source of OH production, and hence on 5/12 at the lowest NOx, this makes HONO the largest contributor to the rate of OH production. Figure 4 has been updated with the correct HONO dataset, see response above for the updated version of Figure 4 along with the updated caption.

Crilley, L. R., Kramer, L. J., Ouyang, B., Duan, J., Zhang, W., Tong, S., Ge, M., Tang, K., Qin, M., Xie, P., Shaw, M. D., Lewis, A. C., Mehra, A., Bannan, T. J., Worrall, S. D., Priestley, M., Bacak, A., Coe, H., Allan, J., Percival, C. J., Popoola, O. A. M., Jones, R. L., and Bloss, W. J.: Intercomparison of nitrous acid (HONO) measurement techniques in a megacity (Beijing), Atmos. Meas. Tech., 12, 6449–6463, https://doi.org/10.5194/amt-12-6449-2019, 2019.

See updated figure and caption below:

[Figure]

**Figure 4.** Time-series of $j(O^1D)$, relative humidity (RH), temperature (Temp), CO, $SO_2$, $O_3$, $NO_x$, HONO, boundary layer (BL), $PM_{2.5}$, HCHO, butane and toluene from the 8th of November to 10th December 2016 at Institute of Atmospheric Physics (IAP), Beijing.

**Page 16 and Table 4: The text and table state that the average OH maximum was 2.7 E6 cm-3, but a value of 3.03 E6 cm-3 is stated on page 18.**

We apologise for the inconsistency. The correct value is $2.7 \times 10^6$ molecule $cm^{-3}$, and this has been corrected on page 18.

**Page 19: I am not sure late February/March would be considered mid-summer in Boulder, but rather late winter/early spring.**

Indeed, that is true. We have used the words 'closer to spring' instead.

**Figure 9: The authors should clarify whether this is an experimental radical budget or one derived from the model.**

The caption now explicitly states that the radical budget is calculated from the model.

New caption for Figure 9. "Rates of primary production (top panel) and termination (bottom panel) for $RO_x$ radicals (defined as OH + $HO_2$ + RO + $RO_2$) calculated for MCM-base model separated into haze (right) and non-haze (left) periods. The definition of haze is when $PM_{2.5}$ exceeds 75 $\mu m^{-3}$. The production from: $O^1D + H_2O$, VOC + $NO_3$, carbonyls + $hv$ and the termination reactions: $RO_2 + HO_2$, $HO_2$ + $HO_2$, $HO_2 + NO_2$, although shown in the key, are not visible and contributed <1% of the total prodcution and termination."

**Given the importance of HONO to radical initiation, how sensitive was the model to the systematic differences in the HONO measurements as described in Crilley et al. (2019)?**

Rather than show the impact of different HONO concentrations using the MCM model, we have demonstrated the impact of different HONO measurements using a sensitivity analysis of the PSS calculation for OH. The model was not used since the effect would be small as the underprediction of the radicals derives from the $RO_2$ chemistry which does not lead to any terms in the PSS equation. The sensitivity of the results of the PSS calculation towards the HONO concentration has been included in

the supplementary material, and shows the PSS can be perturbed up to 17% when the HONO measurement is increased/decreased by 40%.

Some new text has been added to the supplementary material as follows:

**"S1.5 Exploring the sensitivity of the photostationary steady-state OH calculation to the HONO concentration.**

The HONO concentration used to constrained both the model and the photostationary steady-state calculation was the suggested value by Crilley et al.(2019). During the campaign there was several HONO measurement present and, although the measurements agreed on temporal trends and variability ($r^2$>0.97), the absolute concentration diverged between 12 – 39%, the value suggested by Crilley et al. (2019) was the mean of the measurements. Since HONO is a primary source of OH the impact of the variable HONO concentration has been explored by increasing and decreasing the HONO by 40%, the results are shown in Figure S6. Figure S6 shows that the variation observed in the HONO measurements can increase/decrease the PSS up to 17% which is smaller than the error on the measured OH of ~26%.

[Figure]

**Figure S6** Top – Percentage change in the OH calculated from the PSS when the HONO is varied by 40%. Bottom – Comparison of the measured OH and the OH calculated from the PSS using the mean suggested value by Crilley et al. (2019).

"

and a sentence has been added to the main paper regarding the conclusions of this sensitivity analysis, as follows:

"The different HONO measurements present during the APHH campaign varied up-to ~40%, the sensitivity of the PSS on measured HONO is shown in the supplementary material section S1.5."

**Page 26: There appears to be a problem with the signs in Equation 3 (see the corresponding equation in Tan et al. (2018))**

This has now been fixed.

**Page 26, line 560: Here it is stated that the P'(ROx) is 1.2 E8 cm-3 s-1, but on page 27 line 575 states that it is 1.01 E8 cm-3 s-1.**

We apologise for the inconsistency. The P'RO$_2$ is higher in the updated version of Figure 12 (see above) as the original Figure 12 had not been filtered for when measured data was available only.

The correct value is 3 x 10$^8$ cm$^{-3}$ s$^{-1}$ and has been corrected in both instances.

[revised manuscript text omitted]

**S1.4 Estimating the contribution of $HO_2NO_2$ and $CH_3O_2NO_2$ to the $RO_2$ signal**

In the main paper we do not apply a correction for a possible contribution of pernitric acid (PNA, $HO_2NO_2$) and methyl peroxy nitric acid (MPNA, $CH_3O_2NO_2$). The MPNA decomposition will contribute to the simple $RO_2$ and total $RO_2$ whilst the PNA contributes to the complex and total $RO_2$ measurements. The concentration of $HO_2NO_2$ and $CH_3O_2NO_2$ was modelled using the MCM-base model, then in agreement with the work by Fuchs et al.(2008) 0.43 % and 9 % of the $HO_2NO_2$ and $CH_3O_2NO_2$ is calculated to decompose and contribute to the $RO_2$ signal. The rate of decomposition in the Julich and Leeds $RO_x$LIF reactors is expected since the design and residence time (~1 second) are similar. The comparison of the measured total, simple and complex $RO_2$ with the corrected values is shown in Figure S5. Figure S5 shows that the correction from the decomposition of $HO_2NO_2$ and $CH_3O_2NO_2$ is ~6 %, ~8 % and 4 % for total, complex and simple $RO_2$, respectively.

[Figure]

**Figure S2.** a) Timeseries comparison for measured total RO$_2$ (blue) and total RO$_2$ corrected (black) for the decomposition from HO$_2$NO$_2$ and CH$_3$O$_2$NO$_2$. b) Timeseries comparison for measured complex RO$_2$ (blue) and complex RO$_2$ corrected (black) for the decomposition from HO$_2$NO$_2$. c) Timeseries comparison for measured simple RO$_2$ (blue) and simple RO$_2$ corrected (black) for the decomposition from CH$_3$O$_2$NO$_2$.

**S1.5 Exploring the sensitivity of the photostationary steady-state OH calculation to the HONO concentration.**

The HONO concentration used to constrained both the model and the photostationary steady-state calculation was the suggested value by Crilley et al.(2019). During the campaign there was several HONO measurement present and, although the measurements agreed on temporal trends and variability ($r^2$>0.97), the absolute concentration diverged between 12 – 39%, the value suggested by Crilley et al. (2019) was the mean of the measurements. Since HONO is a primary source of OH the impact of the variable HONO concentration has been explored by increasing and decreasing the HONO by 40%, the results are shown in Figure S6. Figure S6 shows that the variation observed in the HONO measurements can increase/decrease the PSS up to 17% which is smaller than the error on the measured OH of ~26%.

[Figure]

**Figure S3.** Top – Percentage change in the OH calculated from the PSS when the HONO is varied by 40%. Bottom – Comparison of the measured OH and the OH calculated from the PSS using the mean suggested value by Crilley et al. (2019).

**S1.6 In-depth comparison of measured OH and OH calculated from the PSS on the 04/12 using measured and modelled OH reactivity.**

On the 04/12/2016 the PSS calculation for OH is overpredicted by ~2.5 and the modelled OH reactivity is higher than the measured OH reactivity by an average of ~14 s$^{-1}$. The modelled OH reactivity was used in the PSS calculation for OH and a comparison between the PSS calculation using measured and modelled kOH and measured OH is shown in Figure S7. Figure S7 shows that whilst using the modelled OH reactivity does reduce the calculated PSS OH, the PSS using modelled kOH still overpredicts the measured OH by a factor of ~2.4. The large overprediction by the PSS suggests the differences between the PSS and measured OH on the 04/12/2016 stems from measurement problems and could be derived from issues with the OH, HO$_2$, HONO or NO measurements on this day.

[Figure]

**Figure S4.** Comparison of measured OH (with errors, blue bars) with OH calculated from a photostationary steady-state (PSS) calculation using measured OH reactivity. The contributions towards OH production from HONO + hv (green) and $HO_2$ + NO (red) are shown, as well as the OH calculated using the PSS but with modelled OH reactivity (black)."

**S1.7 The effects of the kRO2 + NO rate constant on the modelled radical species**

Other than $CH_3O_2$ and $C_2H_5O_2$, rate constants for the reaction of many other $RO_2$ + NO is based on structure activity relationships (SARs) in the MCM and is lumped to kRO2NO and kAPNO (http://mcm.leeds.ac.uk/MCM/). The lumped rate constants kRO2NO and kAPNO were both decreased by a factor of 2 and 10 to investigate the effects on modelled OH, $HO_2$ and $RO_2$. The model where the rate constant for $RO_2$ + NO was decreased by a factor of 2 is titled MCM-kRO2-2, whilst the model where the rate constant was decreased by a factor of 10 is titled MCM-kRO2NO-10.

The comparison of measured values with modelled values (MCM-base, MCM-kRO2-2 and MCM-kRO2-10) is shown in Figure S8. Figure S8 shows that on certain days (e.g. 19/11, 5/12 and 9/12) when the model (MCM-base) could not reproduce the measured values of $RO_2$ the discrepancy between the measurements and the MCM-kRO2NO-10 model is almost reconciled. On these days the MCM-kRO2NO-10 does not really change the OH or $HO_2$ concentration from the base model. Onall days the MCM-base underpredicts the $RO_2$ concentration, and MCM-kRO2NO-10 does decrease the gap between measurements and modelled, compared to MCM-base. MCM-kRO2NO-2 does not significantly increase the total $RO_2$ concentration from MCM-base, unlike MCM-kRO2NO-10. Since

changing the rates of $RO_2$ + NO will be very dependent on the NO concentration, the ratio of measured:modelled radical concentration has been binned against the log of NO for MCM-base, MCM-kRO2NO-2 and MCM-kRO2NO-10 in Figure S9. Figure S9 shows similar results to the timeseries where at the lower concentration of NO (19/11, 5/12 and 9/12) the MCM-kRO2NO-10 can reproduce the $RO_2$ concentration. The results at higher [NO] show that decreasing the rate of $RO_2$ + NO improves the agreement between measured:modelled $RO_2$, especially for MCM-kRO2NO-10, but the observed $RO_2$ concentration is still underpredicted beyond 30 ppbv.

[Figure]

**Figure S5.** (a) Time-series comparison of measured values of OH with modelled OH concentrations from MCM-base, MCM-kRO23NO-2 and MCM-kRO2-10. (b) Time-series comparison of measured values of $HO_2$ with modelled $HO_2$ concentrations from MCM-base, MCM-kRO23NO-2 and MCM-kRO2-10. (c) Time-series comparison of measured values of total $RO_2$ with modelled total $RO_2$ concentrations from MCM-base, MCM-kRO23NO-2 and MCM-kRO2-10. The data sets are 15-minutes averaged.

The fact that the OH and $HO_2$ modelled concentrations do not change significantly for the models with reduced $RO_2$ + NO rate constant highlights that the enhanced $RO_2$ radicals (in MCM-kRO2-10) are not recycling into $HO_2$ or OH, even though the agreement for the $RO_2$ concentration is improved for these models (MCM-kRO2NO-2 and MCM-krO2NO-10). The lack of $RO_2$ recycling highlights that the $RO_2$ and RO radicals are terminating rather than propagating in the model.

This work highlights alternative chemistry and solutions must be applied for the two different NO regimes observed during the Beijing wintertime campaign. At high [NO] (above 10 ppbv) further

reductions in the $RO_2$+NO rate constant would be required to reconcile the model with observations. However, at NO mixing ratios below 10 ppbv, further reductions in the $RO_2$+NO rate constant would lead to the model overpredicting the $RO_2$ concentration.

[Figure]

**Figure S6.** The ratio of measurement/model for OH (a), $HO_2$ (b) and $RO_2$ (c) across various NO concentrations for daytime values only ($j(O^1D) > 1 \times 10^{-6}$ s$^{-1}$). Light blue represents for results from MCM-kRO2NO-2, dark blue represents results from MCM-base and red represents results from MCM-kRO2NO-10.

**S1.8 $ClNO_2$ and Cl concentration required to bridge the gap between measured and modelled total $RO_2$**

Unfortunately, there were no $ClNO_2$ measurements during the winter campaign, and hence it was not possible to calculate a time series for Cl atoms formed from photolysis of $ClNO_2$ and to assess any additional $RO_2$ radicals generated. Using the model run where additional $RO_2$ source was added to reconcile the measurements and the model a rough calculation has shown that the $ClNO_2$ concentration would have to be on average ~5800 ppbv in order to close the gap between modelled and measured $RO_2$. Figure S10 shows the average diel of the calculated $ClNO_2$ and Cl concentration with peak at $1.4 \times 10^4$ ppbv and $1.6 \times 10^6$ molecule cm$^{-3}$, respectively. The $ClNO_2$ and Cl concentration have been calculated using SE3 – SE5:

$$P'RO_2 = k_{VOC+Cl}[VOC][Cl] \qquad\qquad \text{S E3}$$

$$[Cl] = \frac{P'RO_2}{k_{VOC+Cl}[VOC][Cl]} \qquad\qquad \text{S E4}$$

$$[ClNO_2] = \frac{k_{VOC+Cl}[VOC][Cl]}{jClNO_2} \qquad\qquad \text{S E5}$$

where $k_{VOC+Cl}$ is a generic rate constant to represents the reaction of all VOCs with Cl which in this case is 4 x $10^{-12}$ molecule$^{-1}$ cm$^3$ s$^{-1}$, [VOC] is the sum of the measured VOC concentration for the campaign and P'RO$_2$ is the calculated additional RO$_2$ used in MCM-PRO2 (see main paper section 4.2 for more details). The ClNO$_2$ required to bridge the gap between measured and modelled of RO$_2$ is ~3 orders of magnitude greater than the peak ClNO$_2$ concentration measured in suburban Beijing (2.9 ppbv) by Wang et al. (2018) suggesting that other additional primary source are needed in the model besides Cl chemistry .

[Figure]

**Figure S7.** Average diel of the ClNO$_2$ and Cl atom concentration required to bridge the gap between measured and modelled RO$_2$. The ClNO$_2$ and Cl concentrations have been calculated from the additional primary source of RO$_2$ added to the MCM-PRO2 model run, see section 4.2 in the main paper for more details.

**References**

[revised manuscript text omitted]